# A pangenome and pantranscriptome of hexaploid oat

Raz Avni[1,41], Nadia Kamal[2,3,41], Lidija Bitz[4,41], Eric N. Jellen[5,41], Wubishet A. Bekele[6,41], Tefera T. Angessa[7], Petri Auvinen[8], Oliver Bitz[4], Brian Boyle[9], Francisco J. Canales[10], Craig H. Carlson[11], Brett Chapman[7], Harmeet Singh Chawla[12], Yutang Chen[13], Dario Copetti[14], Samara Correia de Lemos[2], Viet Dang[7], Steven R. Eichten[15], Kathy Esvelt Klos[16], Amit M. Fenn[3], Anne Fiebig[1], Yong-Bi Fu[17], Heidrun Gundlach[3], Rajeev Gupta[11], Georg Haberer[3], Tianhua He[7], Matthias H. Herrmann[18], Axel Himmelbach[1], Catherine J. Howarth[19], Haifei Hu[7], Julio Isidro y Sánchez[20], Asuka Itaya[6], Jean-Luc Jannink[21,22], Yong Jia[7], Rajvinder Kaur[23], Manuela Knauft[1], Tim Langdon[19], Thomas Lux[3], Sofia Marmon[24], Vanda Marosi[2,3], Klaus F. X. Mayer[2,3], Steve Michel[25], Raja Sekhar Nandety[11], Kirby T. Nilsen[12,26], Edyta Paczos-Grzęda[27], Asher Pasha[28], Elena Prats[10], Nicholas J. Provart[28], Adriana Ravagnani[19], Robert W. Reid[29], Jessica A. Schlueter[29], Alan H. Schulman[4,8,30], Taner Z. Sen[25,31], Jaswinder Singh[23], Mehtab Singh[23], Nick Sirijovski[24,40], Nils Stein[1,32], Bruno Studer[13], Sirja Viitala[4], Shauna Vronces[28], Sean Walkowiak[33], Penghao Wang[34], Amanda J. Waters[35], Charlene P. Wight[6], Weikai Yan[6], Eric Yao[25], Xiao-Qi Zhang[7], Gaofeng Zhou[7], Zhou Zhou[23], Nicholas A. Tinker[6✉], Jason D. Fiedler[11✉], Chengdao Li[7,36,37,38✉], Peter J. Maughan[5✉], Manuel Spannagl[3,36✉] & Martin Mascher[1,36,39✉]

Oat grain is a traditional human food that is rich in dietary fibre and contributes to improved human health[1,2]. Interest in the crop has surged in recent years owing to its use as the basis for plant-based milk analogues[3]. Oat is an allohexaploid with a large, repeat-rich genome that was shaped by subgenome exchanges over evolutionary timescales[4]. In contrast to many other cereal species, genomic research in oat is still at an early stage, and surveys of structural genome diversity and gene expression variability are scarce. Here we present annotated chromosome-scale sequence assemblies of 33 wild and domesticated oat lines, along with an atlas of gene expression across 6 tissues of different developmental stages in 23 of these lines. We construct an atlas of gene-expression diversity across subgenomes, accessions and tissues. Gene loss in the hexaploid is accompanied by compensatory upregulation of the remaining homeologues, but this process is constrained by subgenome divergence. Chromosomal rearrangements have substantially affected recent oat breeding. A large pericentric inversion associated with early flowering explains distorted segregation on chromosome 7D and a homeologous sequence exchange between chromosomes 2A and 2C in a semi-dwarf mutant has risen to prominence in Australian elite varieties. The oat pangenome will promote the adoption of genomic approaches to understanding the evolution and adaptation of domesticated oats and will accelerate their improvement.

Oat (*Avena sativa*, 2n = 6x = 42, $A_sA_sC_sC_sD_sD_s$ genome[5]) is the world's seventh most widely grown cereal crop[6]. It is appreciated for its high content of dietary fibre, which has been shown to have substantial benefits for human health[1,2]. In 2022–23, more than 25 million metric tonnes were produced worldwide. Genetically improved cultivars have the potential to make oat cultivation more productive and sustainable, but much of this potential remains unrealized, and the first oat reference sequences have been published only in the past few years[4,7,8]. The complexity of the oat genome is partly to blame for the slow progress. Oat is an allohexaploid species with the subgenomes A, C and D, each between 3 Gb and 4 Gb in size[4]. In contrast to bread wheat, which arose as a hexaploid only about 12,000 years ago[7,9], oat's conspecific wild progenitor *Avena sterilis*[10,11],

a wild grass that is common in western Asia and the Mediterranean basin, has been a hexaploid for at least 500,000 years[12]. Inheritance in oat is disomic; that is, chromosomes from different subgenomes (homeologues) do not generally recombine. Even so, the presence[7,12] of three subgenomes in the same nucleus has afforded opportunities for rare homeologous exchanges to reshuffle the oat genome[4]. In the first analyses of a chromosome-scale oat genome sequence[4], chromosomes were assigned to the A, C and D subgenomes according to which diploid progenitor their pericentromeres were descended from. However, this was an incomplete approximation of oat genomic ancestry. For example, genes that could be traced back to C genomes now reside on chromosomes whose pericentromeres match those of A and D genome species[4].

---

Moreover, all but one intergenomic interchange occurred between the C and D subgenomes in the evolution of the tetraploid progenitor *Avena insularis* ($C_iC_iD_iD_i$), which may well have existed for several million years before hexaploidization and the addition of genome A from *Avena longiglumis*[4]. Now that contiguous genome sequences can be assembled even for complex, repeat-rich plant genomes[13], the polyploid and mosaic ancestry of oat should no longer be seen as a challenge, but rather as an opportunity for pangenomic analyses. In this context, oat provides a model system for questions such as how genic presence–absence variation (PAV) affects gene expression in a polyploid and whether structural variation in an old polyploid, especially sequence exchange between subgenomes, affects breeding. To tackle these and other questions, we studied gene-expression diversity and structural variation in an annotated pangenome of cultivated oats and allied taxa.

## An annotated pangenome of hexaploid oat

We assembled and annotated the genomes of 33 diverse oat lines (Supplementary Table 1). Henceforth, we refer to these lines as the PanOat panel. This panel comprises (i) commercially successful elite varieties from major oat-growing regions; (ii) plant genetic resources with interesting properties; (iii) two accessions of wild *A. sterilis*; (iv) *Avena occidentalis*; (v) the closest extant relatives of oat's diploid and tetraploid progenitors, *A. longiglumis* and *A. insularis;* (vi) Amagalon, a synthetic hexaploid; and (vii) two distant diploid *Avena* species, *Avena eriantha* and *Avena atlantica*. The PanOat lines cover most of the genetic diversity space of the crop[14], as represented in a principal component analysis (PCA) of 9,111 diverse wild and domesticated gene-bank accessions, and global breeding germplasm (Fig. 1a). Genome sequences for six members of the PanOat panel were published previously[4,8,15]. The genomes of three lines—Gehl, AAC Nicolas and *Avena byzantina* PI258586—were sequenced with Illumina short reads. The remaining 24 genome sequences were assembled from accurate long reads generated on the PacBio HiFi platform (Supplementary Table 1). All contig-level assemblies were scaffolded with chromosome conformation capture (Hi-C) sequencing data. On average, 99.97% of the assembled sequences were assigned to precise chromosomal locations (Supplementary Table 1). We annotated genes on these assemblies using a multi-tiered approach[16]. To do so, we sequenced the transcriptomes of 6 different tissues and developmental stages in 23 PanOat lines (Fig. 2a, Extended Data Figs. 1 and 2a,b and Supplementary Table 2) using Illumina RNA sequencing (RNA-seq). In addition, we sequenced pooled samples using PacBio Iso-Seq. For two additional lines (*A. sterilis* TN4 and *A. byzantina* PI258586), we used only RNA-seq data. These data, as well as evidence from protein homology and ab initio predictions, were used to predict gene models, which were then projected onto the eight PanOat assemblies without native transcriptome data. All gene models were assigned to the high- and low-confidence categories[17] on the basis of their homology to genes in other plants and the presence of domains commonly found in transposable elements (TEs). For a hexaploid oat genome, we predicted between 107,847 and 136,836 genes (Extended Data Fig. 3a), of which 60.5% on average were expressed (Supplementary Table 2). We used benchmarking universal single-copy orthologs (BUSCO)[18] to assess the completeness of our gene annotations. Out of the 4,896 single-copy near-universal orthologues in the Poales BUSCO dataset (poales_odb10), an average of 4,839 (98.8%) were complete in hexaploid oats. Specifically, 3,680 (75.2%) were triplicated, 666 (13.6%) were duplicated and 166 (3.4%) occurred in a single copy (Supplementary Table 3 and Extended Data Fig. 3b). Completeness was slightly lower in short-read than in long-read assemblies.

## A catalogue of gene-based PAV

The gene content of a pangenome can be divided into core, shell and cloud compartments, which consist of genes present in all, many or a single line, respectively[19]. To study genic PAV, we constructed an orthologous framework from our gene annotations using OrthoFinder[20]. To define the three genic categories—core, shell and cloud—in the 30 members of the PanOat panel that are part of this framework (Supplementary Table 3), we used the following thresholds: core genes are present in 30 genomes; shell genes in 2–29 and cloud genes in a single genome. Our orthologous framework had 102,076 hierarchical orthologous groups (HOGs). The core genome comprised 12,671 HOGs (943,786 genes) that contained at least one orthologous gene from all 30 lines. In total, we found 32.7% of the genes in the core, 66.2% in the shell and 1.1% in the cloud genome (Extended Data Fig. 3c,d and Supplementary Table 3). By definition, cloud and shell genes are not present in one or more genomes; we observed PAVs with varying contributions across different genomes (Extended Data Fig. 3c,d). The core genome was enriched for genes involved in essential physiological processes such as flower formation, nutrient uptake and cell-wall organization (Supplementary Table 4). By contrast, the shell genome was enriched for genes related to defence mechanisms and activities, as well as seed storage processes. It included genes encoding various transcription-factor families such as MYB, WRKY, NAC, AP2/ERF and MADS/AGL. The cloud genome was notably enriched for phosphoinositide signalling, which has a role in plant defence[21], and for P-type ATPases, which are crucial for ion transport, pH regulation, nutrient uptake, metal detoxification and maintaining electrochemical gradients essential for plant growth and stress responses[22,23]. These findings are consistent with reports in other plant pangenomes[24,25]. Core genes tended to be more highly expressed than those in the cloud and shell compartments (Extended Data Fig. 3e). In all polyploid oat genomes and tissues, the mean expression of genes assigned to the C genome was significantly lower than that of their A and D counterparts, with a Fisher's exact test $P$ value of $5.46 \times 10^{-45}$ (Supplementary Table 5 and Extended Data Fig. 4a). This confirms trends reported previously in the first genome analysis of a single oat variety[4].

## Diversity in gene-expression dynamics

Our replicated expression data allowed us to investigate the complexity of the oat pantranscriptome. Variation occurs at multiple, intersecting levels: copy numbers of genes vary between subgenomes and across lines, and expression can differ between tissues, subgenomes and lines. To quantify transcript abundance, we mapped the RNA-seq reads of 23 oat lines to their respective genomes. Transcript levels across lines were rendered comparable by our orthologous framework. The resultant gene-expression matrices are available at the Bio-Analytic Resource[26] (https://bar.utoronto.ca/-asher/efp_oat/cgi-bin/efpWeb.cgi). Owing to the plethora of possible patterns, we first focused on a set of 5,965 genes that occurred in single copies in each of the 3 subgenomes in 20 hexaploid *A. sativa* genomes in our panel (A:C:D 1:1:1 configuration in each), termed '60-lets'. The expression patterns of these 'triads' in each genome were classified into one of seven categories on the basis of the Euclidean distance to seven ideal expression level profiles: A-, C- or D-dominant or A-, C- or D-suppressed, in which one gene is predominantly expressed or suppressed, and a balanced category, in which A, C and D genes are equally expressed[4,27] (Extended Data Fig. 4b). Almost half (49.4%) of the 60-lets had the same classification into one of these categories across lines and were termed 'stable triads' (Fig. 2b,c). In most cases (94% on average, with variations across tissues), the expression of stable triads was balanced among subgenomes (Fig. 2b and Supplementary Table 6). These stable balanced triads were, for example, enriched for essential cellular functions such as vesicle trafficking, ribosome biogenesis and protein biosynthesis and modification (Supplementary Table 7). In an average of 3% of the stable triads, expression was unbalanced, with the bias appearing most often in the C-genome orthologue (Fig. 2b,c and Supplementary Table 6).

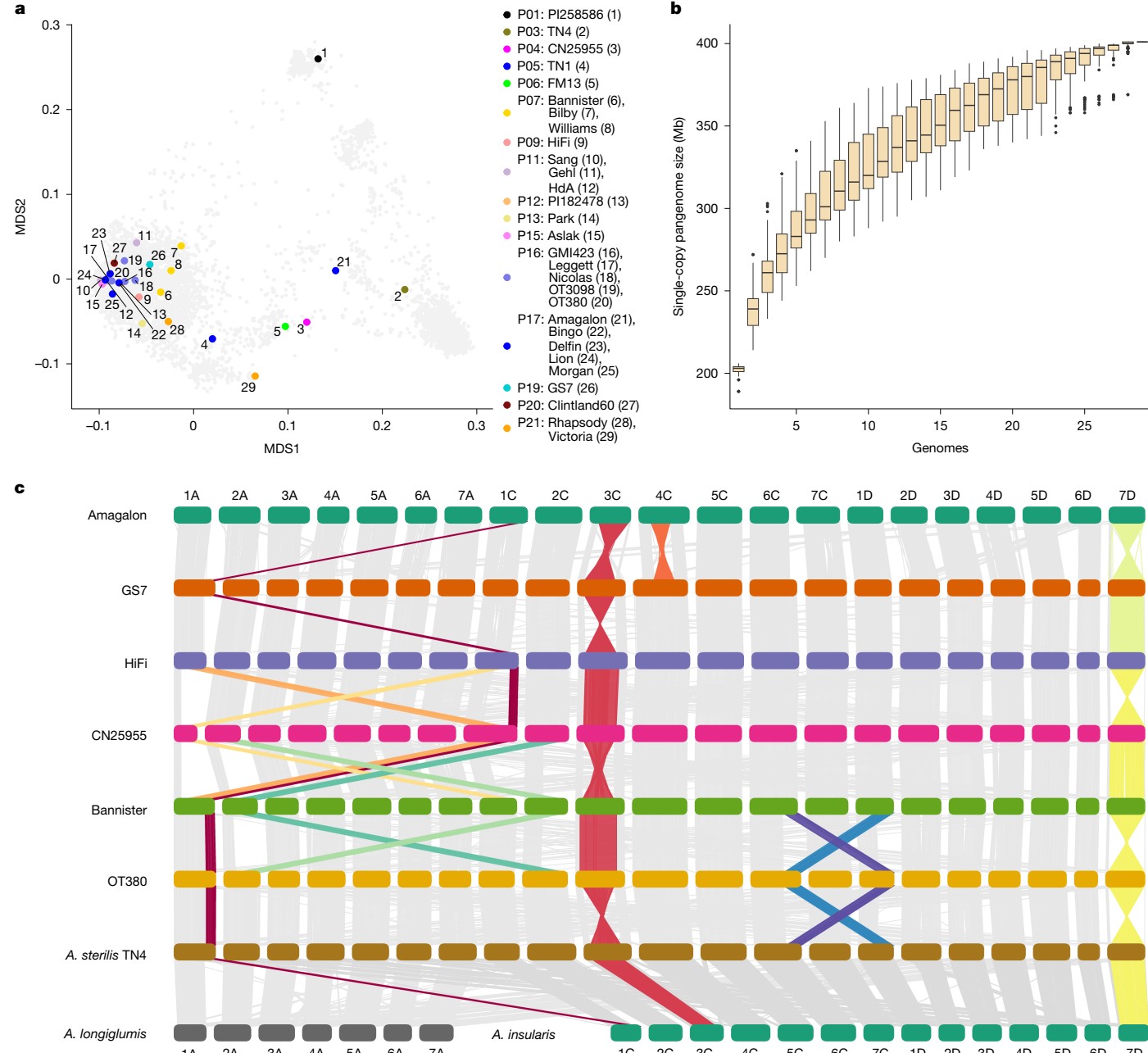

**Fig. 1 | Oat pangenome composition and structural variation.**
**a**, Multidimensional scaling (MDS) plot of 9,111 hexaploid oats including *A. sativa*, *A. byzantina* and *A. sterilis* (light-grey points, ref. 14) overlaid with the position of PanOat assemblies (numbering refers to the PanOat assemblies and colours refer to the population assignment (P1–P21; ref. 14). CN25955, *A. occidentalis* CN25955; HdA, Hâtives des Alpes; Morgan, AC Morgan; Nicolas, AAC Nicolas; PI258586, *A. byzantina* PI258586. **b**, Hexaploid oat pangenome complexity estimated by single-copy *k*-mers. The curves trace the growth of non-redundant single-copy sequences as sample size increases. Error bars are derived from 100 ordered permutations each. The central line represents the median; the box bounds are the interquartile range (IQR), from the first quartile to the third quartile; and the whiskers extend to the most extreme values within 1.5 × IQR from the quartiles. **c**, NGenomeSyn[41] synteny plot between representative PanOat assemblies, including the diploid and tetraploid genome donors (*A. longiglumis* (A genome) and *A. insularis* (CD genomes)), showing major SVs. The grey bars represent syntenic regions between all assemblies, and the coloured bars mark the SVs in the different assemblies.

Expression in roots, embryos and panicles tended to be more conserved across lines, whereas expression in leaf, internode and caryopsis tissues was more variable (Fig. 2d and Supplementary Table 6). For example, HD-ZIP I and HD-ZIP II transcription factors, which were enriched among stable genes in embryo tissue, are crucial regulators in plant embryogenesis[28,29], and stable genes in root tissue were enriched for genes involved in root formation (Supplementary Table 8). This variability was further highlighted by calculating Cramér's V matrices using the expression level categories, revealing notable differences between the lines (Fig. 2e). Part of this variability could be attributed to factors such as geographical distribution, because gene expression might reflect adaptations to environmental conditions. To determine how DNA sequence and expression diversity are related, we compared the matrices derived from expression data with a genetic distance matrix derived from single-nucleotide polymorphism (SNP) data. The clustering analysis performed using these matrices showed broad agreement with genomic distance measures (Mantel test, *P* < 0.001), with consistently positive correlations across all tissues, suggesting

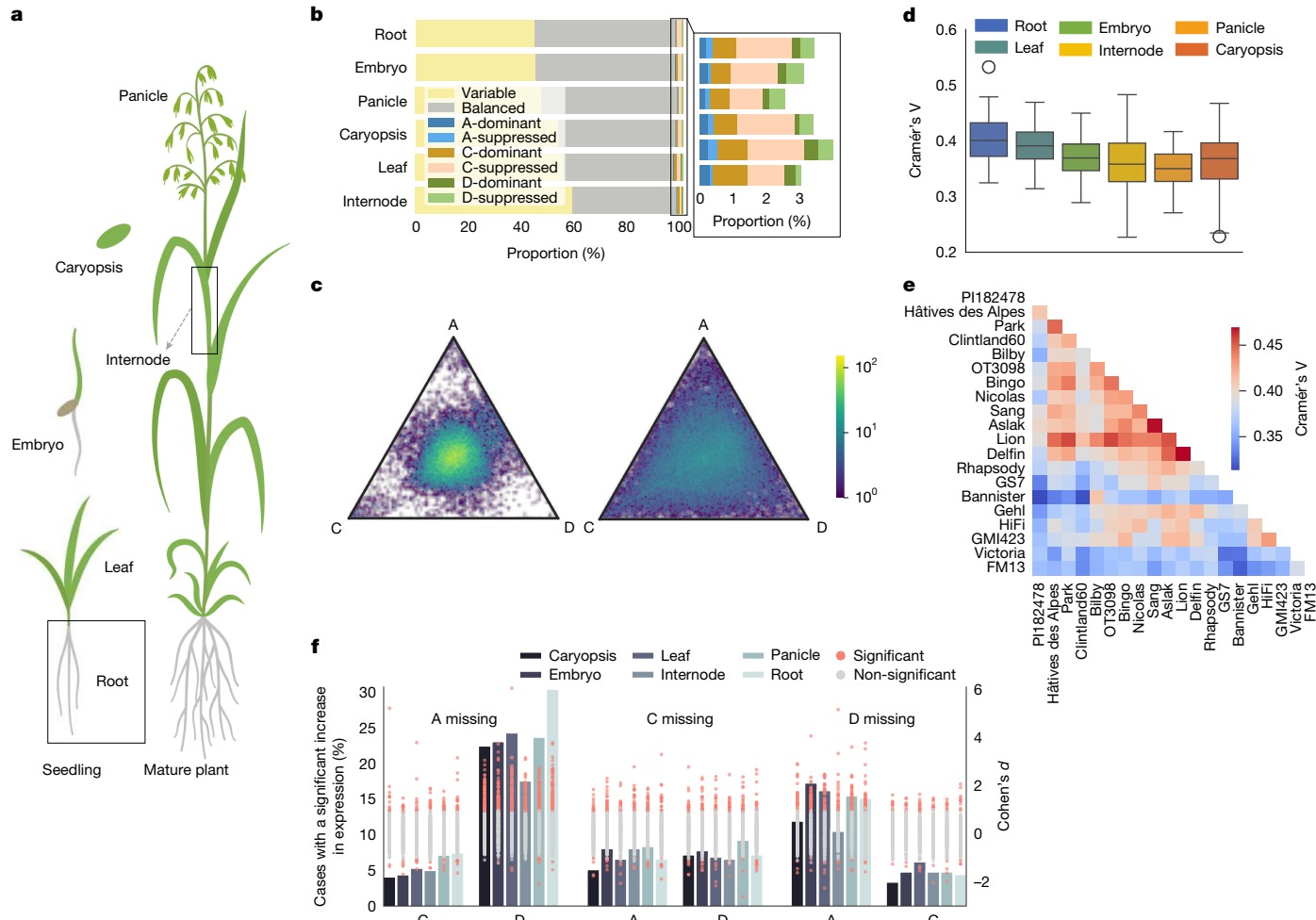

**Fig. 2 | An oat pantranscriptome. a**, Illustration of the six harvested tissues for transcriptome sequencing. **b**, Proportions of stable and variable genes across 20 *A. sativa* lines in 6 tissues. For stable genes, the expression categories are classified into A-, C- and D-dominant, A-, C- and D-suppressed, and balanced. **c**, Ternary plot of stable (left, 46.5% of the triads) and variable (right, 53.5% of the triads) triads in caryopsis tissue. Colour intensity in the hexbin plots represents data density on a logarithmic scale, highlighting the distribution of gene-expression proportions across the three subgenomes (A, C and D). **d**, Pairwise distribution of Cramér's V values between tissues, indicating the degree of similarity in gene-expression patterns. The central line represents the median; the box bounds are the 25th and 75th percentiles (IQR); the whiskers extend to the most extreme data within 1.5 × IQR of the box; and the points

beyond the whiskers are outliers. Thus, minima and maxima refer to the non-outlier range. The *y* axis shows Cramér's V; *n* = 380 pairwise line–line comparisons per tissue (20 lines; all ordered pairs excluding self-comparisons); each Cramér's V computed across 5,964 genes. **e**, Distance matrix based on expression level categories in 20 *A. sativa* lines using Cramér's V, which is exemplified by leaf tissue. **f**, Dynamics of gene expression, comparing complete triads to those missing a homeologue from one subgenome. The grey bars represent the percentage of cases in which the absence of one homeologue leads to a significant expression increase in the remaining two subgenomes. Cohen's *d* values, comparing expression levels between complete and incomplete triads, are shown as dot plots, with significant cases highlighted in pink. *n* = 326 (A missing), 338 (C missing) and 280 (D missing).

that variation in DNA sequences, such as in promoter regions or regulatory elements, contributes to the gene-expression patterns we observed.

When analysing the stability of 60-lets not only across all lines but also across all six analysed tissues, only 13.5% of genes maintained stability. The significant reduction in overall stability indicates considerable tissue-specific differences in gene-expression patterns across genotypes. This set of very conserved genes was enriched for fundamental cellular processes such as protein biosynthesis, vesicle trafficking and RNA processing, among others (Supplementary Table 9). On average, 50.1% of the triads exhibited variability in their classification into one of the seven expression level categories across the 20 *A. sativa* lines, and these were termed 'variable triads' (Fig. 2b,c). Nearly all of these variable triads exhibited expression variability in both lines and tissues. Single-copy orthologues with expression variability were enriched for several families of transcription factors, including R2R3-MYB, ERF, NAC and MADS/AGL (Supplementary Table 7).

After focusing on 60-lets, we relaxed our criteria to include genes that occurred as 1:1:1 single-copies in some genomes, but lacked the copy of one subgenome in others (1:1:0, 1:0:1, 0:1:1). We compared the expression patterns of 'complete' and 'incomplete' triads between 20 *A. sativa* lines to see how gene loss in the hexaploid has affected the expression of the remaining homeologues. The set of incomplete triads comprised 944 members. Among these, 326, 338 and 280 lacked the A, C and D gene copies, respectively. Loss of either the A or the D genome copy was accompanied by a significant increase of expression in the remaining D or A copy in 13.3% and 23.2% of cases, respectively. Higher expression of the C genome compensated for the loss of A or D genes in only 6% and 5.9% of cases, respectively. Similarly, if a C-derived gene was lost, compensatory upregulation of its A or D homeologues occurred in just 4.3% and 4% of cases, respectively (Fig. 2f and Supplementary Table 10). The A and D genomes diverged less than 4.3 million years ago, and are thus more closely related to each other than to the C genome, which split from the A and D lineage 8 million years ago[7]. This shorter

evolutionary distance suggests that A and D genes can more frequently compensate for each other than for a lost C-subgenome copy.

## A map of structural variation

Our chromosome-scale reference sequences made it possible to study the prevalence and impact of large structural variants (SVs) in domesticated oats. The mosaic genome of hexaploid oat is the outcome of rare chromosomal rearrangements that have accumulated over evolutionary timescales[4,7]. Three chromosomal rearrangements that are polymorphic in cultivated oat and its immediate wild progenitor *A. sterilis* were previously discovered using cytological techniques[10]. Not unexpectedly, many of these known events were identified in our genome assemblies (Supplementary Table 11), including (i) a translocation from chromosome 1C to 1A that has been implicated in phenological shifts associated with winter types; (ii) a translocation from 6C to 1D (ref. 30); and (iii) a large pericentric inversion on chromosome 3C (Fig. 1c and Extended Data Fig. 5a). In addition, the pangenome revealed four previously unknown major structural rearrangements: an alternate reciprocal version of the 1A/1C translocation found in the *A. occidentalis* CN25955 assembly; a 2A/2C translocation; a pericentric inversion on chromosome 4C in Amagalon (Extended Data Fig. 5c); and a large 420-Mb pericentric inversion on chromosome 7D (Fig. 3a). The non-reciprocal 1C to 1A translocation polymorphism is present in *A. sterilis* and thus must have arisen before domestication[10]. The same applies to the 3C inversion and the pericentric inversion on chromosome 7D, the allelic state of which differs between the two *A. sterilis* accessions (TN1 and TN4) in the PanOat panel (Fig. 1c and Extended Data Fig. 5a,b). These rearrangements lead to localized gene-expression changes near certain translocation break points—especially on chromosomes 1A/1C, 3C and 7D—affecting metabolic and regulatory genes, which could hint at functional consequences of structural variation (Supplementary Table 12 and Supplementary Fig. 1).

In the following four sections, we highlight case studies from this structural variation map that illustrate its functional importance and applied relevance. These include examples already introduced—the pericentric chromosomes 7D and the 2A/2C translocation, with clear links to flowering time and mutation breeding, respectively—as well as two additional cases: chromosome changes in synthetic-derived lines and widespread effects of SVs on recombination and segregation in breeding populations. Notably, many of these rearrangements are not confined to historical or wild germplasm—they persist in modern cultivars and continue to shape the genomic landscape of contemporary oat breeding.

## An inversion linked to early heading

Tinker et al.[31] observed a lack of recombination in a cross that was polymorphic on most of chromosome 7D, and postulated the presence of an inversion. This prediction was borne out by our data. Although we were not able to design diagnostic markers spanning the inversion break points, owing to high sequence divergence in these regions, we observed two long pericentromeric haplotypes among PanOat lines that were diagnostic of the inversion. These haplotypes were identified in short-read resequencing data (fivefold coverage) of 295 oat varieties from the Collaborative Oat Research Enterprise (CORE) panel[32], which is composed mainly of North American varieties (Supplementary Table 13, Fig. 3b, Extended Data Fig. 6a and Supplementary Fig. 7). Those PanOat genomes that had the ancestral state of the 7D inversion (also seen in *A. insularis*) had the less common of the two haplotypes (1), whereas those with the derived allele, all had the more frequent haplotype (2). A genome-wide association scan (kmerGWAS) for heading date yielded two prominent peaks on chromosomes 7A and 7D, consistent with previous genetic mapping[32] (Fig. 3c). Carriers of haplotype 1 on chromosome 7D flowered on average 3.7 days earlier

than carriers of haplotype 2 (Fig. 3d and Extended Data Fig. 7b,c). Three important quantitative trait loci (QTLs) are mapped to chromosome 7D, involving flowering time (CO and VRN3/FT)[31] and daylength insensitivity (Di)[33]. We further examined the genes within the significant association interval on 7D (468.7–469.8 Mb) and identified an *FT1/VRN3* homologue, a key regulator of flowering time. Its paralogue on chromosome 7A was also analysed for comparison. Expression profiling showed increased transcript levels of both genes in internode tissues, with *FT1*-7D exhibiting markedly higher expression in inverted lines (Fig. 3e). We also detected structural variation: an 18-bp deletion in *FT1*-7D specific to inverted lines, and a 12-bp deletion in *FT1*-7A introducing a premature stop codon in the same genotypes (Supplementary Figs. 5 and 6). Further research is required to find out whether the inversion is associated with flowering owing to suppression of recombination between multiple genes affecting early- and late-flowering haplotypes or, alternatively, whether it influences heading date directly; for instance, by leading to altered expression patterns of flowering time regulators or by affecting the proximity of genes and regulatory factors.

## The hidden legacy of mutation breeding

One of the previously unknown translocations discovered in this study was a homeologous exchange of the short arms of chromosomes 2A and 2C in the Australian variety Bannister (Figs. 1c and 4a). This is unlikely to be an assembly artefact because an intense interchromosomal signal was observed in the Hi-C data for Bannister mapped to the genome of GMI423, an accession with the standard karyotype (Extended Data Fig. 7b). Sequence data from three recombinant inbred lines (RILs) derived from a cross between Bannister and non-translocated Williams revealed whole chromosome arm deletions and compensating duplications on chromosomes 2A and 2C, which are most readily explained by anomalous meiosis in heterozygotes (Extended Data Fig. 7d). Furthermore, we designed diagnostic PCR marker assays spanning the translocation break points to confirm the presence of the 2A and 2C translocations (Extended Data Fig. 7c). Finally, C-banding in Bannister confirmed the presence of recombined 2A and 2C chromosomes with pronounced differences in heterochromatin content between long and short arms (Fig. 4a).

To measure the frequency of this reciprocal translocation in a broader set of oat germplasm, we used the translocation diagnostic markers to genotype 564 accessions from 41 countries around the world and found 17 (3%) that carried the translocation (Supplementary Table 14 and Supplementary Fig. 8). Carriers were mainly of Australian origin and included cultivars released since the last quarter of the twentieth century. Among them were the Dolphin, Echidna, Kalgan and Hay varieties, as well as current highly successful varieties Bannister, Possum and Wombat. These varieties share a common parent in OT207, a mutation breeding line created in 1970 using fast neutron irradiation[34]. OT207 carries the 2A/2C translocation and the *Dw6* mutant allele on chromosome 6D (ref. 35), conferring the semi-dwarf growth habit, but none of its parental varieties do (Fig. 4b). As expected, *Dw6* is not linked to the 2A/2C translocation and occurs also in the absence of the latter (for example, in Bilby). It is thus most likely that the translocation is an unintended side effect of the mutagenesis. However, it is intriguing that the 2A/2C translocation has become so frequent in Australian varieties. We speculated that it has adaptive advantages in Australian environments. To test this hypothesis, we obtained grain yield data from the Australian National Variety Trials (NVT) for the years between 2017 and 2022, which represented 19 to 31 trials per year across all of Australia (158 field trials). Varieties carrying this translocation are among the highest-yielding varieties in Australia and, as a group, significantly ($t$-test, $P = 0.03$) out-yielded their non-translocated counterparts in 131 field trials[32] (Fig. 4c; mean yields of 3,991 kg ha$^{-1}$ versus 3,667 kg ha$^{-1}$). However, given the complexity of yield, with low heritability and strong

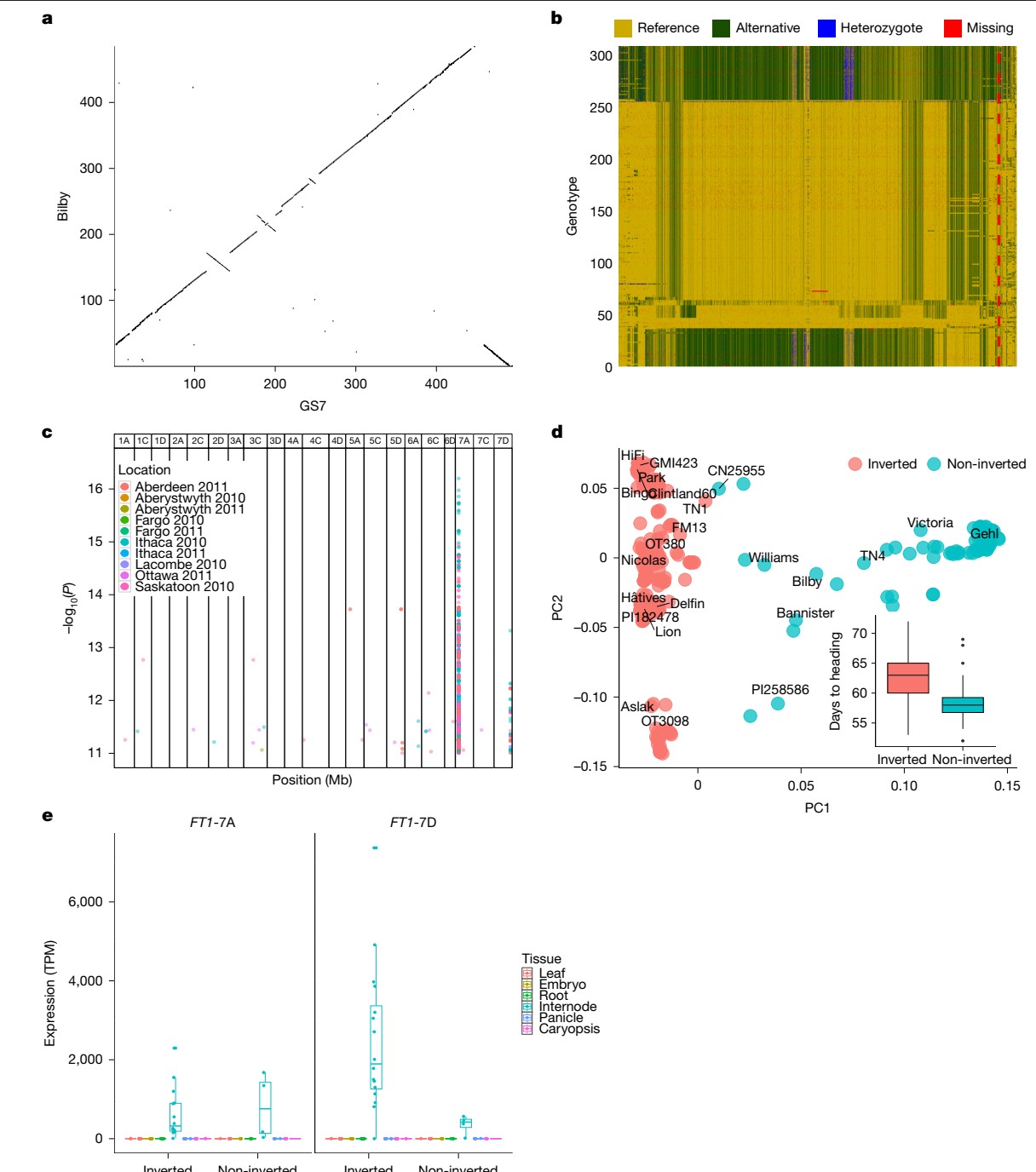

**Fig. 3 | A chromosomal inversion linked to early heading. a**, Genome alignment of the ancestral chromosome 7D in the Australian cultivar Bilby and the inverted chromosome 7D in cultivar GS7 shows a large 450-Mb pericentric inversion and a small 50-Mb distal segment that are the same in both forms. The three small inversions at around 120 Mb, 200 Mb and 240 Mb are unique to the reference genome GS7. **b**, Haplotype plot of 307 CORE and hexaploid PanOat lines sorted by predicted inversion state (top, green, ancestral; bottom, yellow, inverted), on the basis of SNP calling against the GS7 genome. The red dashed line marks the inversion position. **c**, Significant kmerGWAS results for ten locations in 2010 and 2011. Two significant peaks are visible on chromosomes 7A and 7D. **d**, PCA plot of CORE[32] lines ($n = 292$) using SNPs at the distal end of

chromosome 7D (400–495 Mb) overlaid with PanOat assemblies ($n = 24$) (Supplementary Table 1). The box plot shows the difference in heading time in the Ithaca 2010 field trial between inverted ($n = 240$) and non-inverted ($n = 52$) lines. Box plots show the 25th (bottom edges) to 75th (top edges) percentiles, with median lines; whiskers extend to $1.5 \times IQR$; outlier points are observations beyond $1.5 \times IQR$ ($n = 292$). TN1, *A. sterilis* TN1; TN4, *A. sterilis* TN4. **e**, Expression pattern (TPM, transcripts per million) across tissues of the *FT1* gene on chromosome 7D (*FT1*-7D) and chromosome 7A (*FT1*-7A) in 21 annotated hexaploid PanOat accessions. Box plots show the 25th (bottom edges) to 75th (top edges) percentiles, with median lines; whiskers extend to $1.5 \times IQR$; outlier points are observations beyond $1.5 \times IQR$ ($n = 108$ (*FT1*-7A) and $n = 120$ (*FT1*-7D)).

G × E effects, this association should be viewed as exploratory, and warrants targeted studies for confirmation.

To study the effect of the translocation, we analysed a population of RILs derived from a cross between Bannister, which carries

the translocation, and non-translocated Williams, which does not. Among 285 RILs, we identified 10 with chromosomal arm deletions; these showed harmful agronomic traits such as reduced fertility and grain shattering. Sequencing the genomes of three of the RILs from

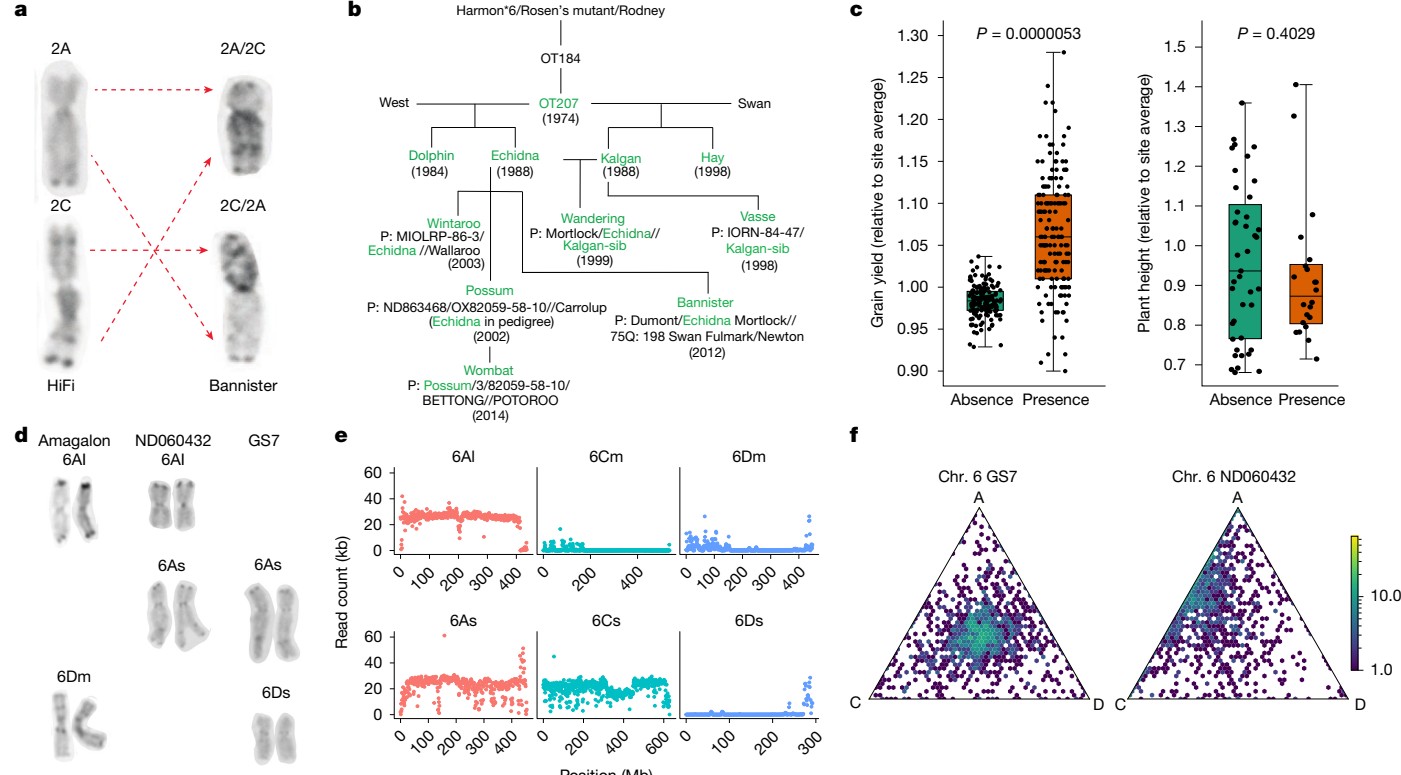

**Fig. 4 | Chromosomal translocations and homeologous sequence exchanges.**
**a**, Karyotype analysis comparing chromosomes 2A and 2C between a non-translocated variety (HiFi) and a translocated one (Bannister). **b**, Origin of the 2A/2C translocation and inheritance in Canadian and Australian oat varieties (green text highlights all the lines with the 2A/2C translocation that had OT207 as a common parent in their pedigree). **c**, Yield advantage of the 6 Australian varieties carrying the 2A/2C translocation in Australia's NVT, in comparison with 11 varieties without the translocation. Two-sided *t*-test, $n = 158$ (grain yield for both translocation presence and absence), $n = 22$ (height, translocation presence) and $n = 41$ (height, translocation absence). Box plot shows the median; the IQR from the 25th to 75th percentile (box limits); and the minimum

and maximum (whiskers). **d**, Karyotypes of Amagalon, *A. sativa* ND060432 and *A. sativa* GS7. The karyotypes show the lack of chromosome 6D in ND060432 and its substitution with chromosome 6A from Amagalon. **e**, Read mapping counts of whole-genome resequencing data from ND060432, showing the substitution of *A. sativa* 6D with Amagalon 6A. Reads were mapped to a hybrid reference combining Amagalon (top) and *A. sativa* GS7 (bottom). **f**, Ternary plot of the expression of triads on chromosome 6 in GS7 (left) and ND060432 (right) for leaf tissue. Each point represents the relative contribution of the three subgenomes (A, C and D) to the expression of a homeologous triad. Colour intensity in the hexbin plots reflects data-point density on a logarithmic scale.

the above ten lines confirmed whole chromosome arm deletions and compensating duplications on chromosomes 2A and 2C, which are most readily explained by aberrant meiosis as heterozygotes.

Using the Bannister × Williams population, previous mapping results[35] and the genome sequences of both parents to fine-map this locus, we narrowed down the QTL for plant height to a 554-kb interval on chromosome 6D. Three members of the PanOat panel—Bilby, Bannister and OT3098—are semi-dwarfs and share a common haplotype in the *Dw6* interval. This haplotype comprises a 41-kb large insertion containing nine genes, one of which encodes a fatty acid hydroxylase (Supplementary Table 15.8). Further functional studies are required to test whether this gene is causal to the *Dw6* phenotype.

## Genomic changes in synthetic-derived oat

Our pangenome panel includes Amagalon, a synthetic hexaploid oat that was derived from a cross between diploid *A. longiglumis* ($A_lA_l$; CW-57) and tetraploid *Avena magna* ($C_mC_mD_mD_m$; CI8330), a close relative of *A. insularis* ($C_iC_iD_iD_i$)—the putative donor of the CD genome in hexaploid oat[36]. A source of disease-resistance genes[37], Amagalon and a derived cultivar HiFi have been widely used in US and Canadian oat breeding. The Amagalon assembly revealed several chromosomal translocations relative to other oat varieties (Extended Data Fig. 5c). To determine whether these SVs might have affected inheritance in

Amagalon-derived breeding lines, we analysed whole-genome sequencing (WGS) and genotyping-by-sequencing (GBS) data and identified numerous regions in which read depth in 1-Mb windows deviated from the genome-wide average (Extended Data Figs. 8 and 9). The sizes of these regions varied from several Mb to whole chromosomes, in which the read depth halved, doubled or declined to zero. We found four lines from the North Dakota State University (ND) breeding program in our WGS panel (Supplementary Table 13) that showed a drop in read depth on chromosome 6D and a concomitant increase on chromosome 6A. The pedigree of these lines suggested that some of them had Amagalon in their pedigree (https://oat.triticeaetoolbox.org/). We analysed the karyotypes of Amagalon, ND060432 (chromosome $6D_s$ missing) and GS7 (chromosome $6D_s$ present), and confirmed the presence of two 6A chromosomes in ND060432, $6A_s$ and $6A_l$ (Fig. 4d). When the sequence data of the four lines were aligned to a reference combining Amagalon and the variety GS7, few reads mapped to either of the $6D_s$ reference chromosomes; however, reads abundantly mapped to both $6A_s$ and $6A_l$ chromosomes coming from GS7 and Amagalon, respectively (Fig. 4e). This suggests that the original cross involving Amagalon resulted in the *A. sativa* $6D_s$ chromosome being replaced with a $6A_l$ chromosome, which is ultimately derived from *A. longiglumis*. Owing to the low resolution of C-banding, it is unlikely that this event could have been discovered without genome sequencing and pangenome analysis.

We investigated the functional consequences of this SV by analysing gene expression in leaf tissue of ND060432 and compared it to the *A. sativa* line GS7. Specifically, the average expression across 20 bins on chromosome 6D was 2.22 TPM, compared to 14.81 TPM on 6A. This difference was highly significant (Wilcoxon rank-sum test, false discovery rate (FDR)-adjusted $P = 2.8 \times 10^{-6}$; Extended Data Fig. 4c–e and Supplementary Table 16). Subgenome-level categorization further confirmed that the D-suppressed expression pattern was significantly overrepresented in chromosome group 6 (chi-squared test, $P < 0.001$; standardized residual = 32.8; Fig. 4f, Extended Data Fig. 4f and Supplementary Table 16), indicating a marked shift in transcriptional dynamics resulting from the 6D replacement. The fact that the ND lines were selected for inclusion in the CORE panel owing to their strong agronomic performance suggests that such exchanges between homeologous chromosomes—and the resulting shifts in gene expression—are well tolerated and could be harnessed in future breeding efforts.

## SVs affect recombination

To gauge the extent to which SVs affect oat breeding, we genotyped and analysed recombination and segregation patterns in 13 biparental populations to assess the effect of SVs in an active oat breeding program. Numerous crosses exhibited segregation distortion as well as non-linear or suppressed recombination within or between chromosomes (Extended Data Fig. 10a). These include regions on chromosomes 1A, 1C, 3C, 4C and 7D, where inversions and translocations identified in this study and the companion global oat genomic diversity analysis[14] were confirmed. We analysed and karyotyped two half-sib crosses in greater detail (Extended Data Fig. 10b–f). One cross exhibited pseudo-linkage between chromosomes 1A and 1C associated with a 1A/1C reciprocal translocation heterozygote, as well as recombination suppression on chromosome 7D associated with inversion heterozygotes. The other showed recombination suppression on chromosome 3C, also associated with known chromosome inversion. Signatures of other SV heterozygotes were observed, including a potential translocation between chromosomes 1D and 7C. The companion study[14] introduces a SNP-based in-silico karyotyping method that could help breeders to avoid crosses with segregation irregularities, to design genomic selection strategies to reduce linkage drag or to use suppressed recombination to preserve blocks of adaptive alleles.

## Discussion

The oat pangenome and its detailed analysis reported here should accelerate the adoption of genomic methods in oat research and breeding. Multiple reference genome sequences, an extensive gene-expression atlas and resequencing data of a well-characterized diversity panel greatly expand the genomic resources available to oat researchers and will have an immediate effect on genomics-assisted breeding. As in other crops, the full extent of structural variation in elite varieties of oat has been unknown. In tracing the origin and effect of a recent homeologous recombination event, the 2A/2C translocation, we encountered a situation that is analogous to one described previously in barley[38]. These authors discovered a 141-Mb paracentric inversion, which was probably induced by irradiation and spread owing to a founder effect because the mutant variety had a desired semi-dwarf growth habit. This supports the notion that mutation breeding in the twentieth century resulted in positive changes to agronomically relevant traits, but also led to cryptic chromosomal aberrations that continue to influence crop improvement in surprising ways. Cytological evidence[39] leads us to expect that more such events will be discovered in oat germplasm.

A pangenome is an indispensable resource for applications in marker discovery, genetic mapping and molecular breeding. A map of large SVs will help breeders to interpret segregation patterns in crosses and guide the selection of parents. We also hope that the resources we have built will stimulate oat research beyond translational applications, and motivate more scientists to make oat a major research focus. There are still many open research questions in wild and domesticated oat that remain unanswered, including the genetic basis of the domestication syndrome in oat. Its key components are loss of grain shattering, reduction of the geniculate awn, loss of lemma pubescence and husk adherence[40], as well as loss of dormancy and changes in photoperiod or vernalization response. Some encouraging progress in the latter trait has been made thanks to the genome sequence of a hulless oat[7]. The next steps in oat genomics should be (i) expanding the pangenome to include all populations of cultivated oat and its wild progenitor[14]; and (ii) working towards a genus-wide pangenome of *Avena* comprising all of the approximately 30 species in the genus. The latter effort would underpin inquiries into structural genome evolution, which have so far been limited to the hexaploid and its immediate progenitors.

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

[1]Leibniz Institute of Plant Genetics and Crop Plant Research (IPK) Gatersleben, Seeland, Germany. [2]School of Life Sciences, Technical University of Munich, Freising, Germany. [3]Plant Genome and Systems Biology (PGSB), Helmholtz Center Munich–German Research Center for Environmental Health, Neuherberg, Germany. [4]Natural Resources Institute Finland (LUKE), Helsinki, Finland. [5]Plant and Wildlife Sciences, Brigham Young University, Provo, UT, USA. [6]Ottawa Research and Development Centre, Agriculture and Agri-Food Canada, Ottawa, Ontario, Canada. [7]Western Crop Genetics Alliance, Food Futures Institute and School of Agriculture, Murdoch University, Perth, Western Australia, Australia. [8]Institute of Biotechnology, University of Helsinki, Helsinki, Finland. [9]Institut de Biologie Intégrative et des Systèmes, Université Laval, Québec City, Quebec, Canada. [10]Institute for Sustainable Agriculture, CSIC, Córdoba, Spain. [11]Cereal Crops Improvement Research Unit, USDA-ARS, Edward T. Schafer Agricultural Research Center, Fargo, ND, USA. [12]Department of Plant Science, University of Manitoba, Winnipeg, Manitoba, Canada. [13]Molecular Plant Breeding, Institute of Agricultural Sciences, ETH Zurich, Zurich, Switzerland. [14]Arizona Genomics Institute, University of Arizona, Tucson, AZ, USA. [15]Agriculture and Food Solutions, General Mills, Minneapolis, MN, USA. [16]Small Grains and Potato Germplasm Research Unit, USDA-ARS, Aberdeen, ID, USA. [17]Plant Gene Resources of Canada, Saskatoon Research and Development Centre, Agriculture and Agri-Food Canada, Saskatoon, Saskatchewan, Canada. [18]Julius Kuehn Institute (JKI), Federal Research Centre for Cultivated Plants, Institute for Breeding Research on Agricultural Crops, Sanitz, Germany. [19]IBERS Gogerddan, Aberystwyth University, Aberystwyth, UK. [20]Centro de Biotecnología y Genómica de Plantas (CBGP), Universidad Politécnica de Madrid (UPM) & Instituto Nacional de Investigación y Tecnología Agraria y Alimentaria (INIA), Madrid, Spain. [21]R. W. Holley Center for Agriculture and Health, USDA-ARS, Ithaca, NY, USA. [22]School of Integrative Plant Science, Cornell University, Ithaca, NY, USA. [23]Department of Plant Science, McGill University, Sainte-Anne-de-Bellevue, Quebec, Canada. [24]ScanOats Industrial Research Center, Department of Chemistry, Division of Pure and Applied Biochemistry, Lund University, Lund, Sweden. [25]Crop Improvement and Genetics Research Unit, USDA-ARS, Albany, CA, USA. [26]Agriculture and Agri-Food Canada, Brandon, Manitoba, Canada. [27]Institute of Plant Genetics, Breeding and Biotechnology, University of Life Sciences in Lublin, Lublin, Poland. [28]Department of Cell and Systems Biology and Centre for the Analysis of Genome Evolution and Function, University of Toronto, Toronto, Ontario, Canada. [29]Department of Bioinformatics and Genomics, University of North Carolina at Charlotte, Charlotte, NC, USA. [30]Viikki Plant Science Centre, Helsinki, Finland. [31]Department of Bioengineering, University of California, Berkeley, Berkeley, CA, USA. [32]Institute of Agricultural and Nutritional Sciences, Martin Luther University Halle-Wittenberg, Halle, Germany. [33]Canadian Grain Commission, Winnipeg, Manitoba, Canada. [34]School of Medical, Molecular and Forensic Sciences, Murdoch University, Perth, Western Australia, Australia. [35]Research and Development Division, PepsiCo, Valhalla, NY, USA. [36]Centre for Crop and Food Innovation, Food Futures Institute, Murdoch University, Perth, Western Australia, Australia. [37]Department of Primary Industry and Regional Development, Government of Western Australia, Perth, Western Australia, Australia. [38]College of Agronomy, Shandong Agricultural University, Tai'An, China. [39]German Centre for Integrative Biodiversity Research (iDiv) Halle-Jena-Leipzig, Leipzig, Germany. [40]Present address: Global Oatly Science and Innovation Centre, Lund, Sweden. [41]These authors contributed equally: Raz Avni, Nadia Kamal, Lidija Bitz, Eric N. Jellen, Wubishet A. Bekele. ✉e-mail: nick.tinker@agr.gc.ca; jason.fiedler@usda.gov; c.li@murdoch.edu.au; jeff_maughan@byu.edu; manuel.spannagl@helmholtz-muenchen.de; mascher@ipk-gatersleben.de

## Methods

### DNA extraction

**Plant growth and isolation of high-molecular-weight DNA.** High-molecular-weight (HMW) DNA was extracted from young leaf tissue from a single two-week-old seedling grown in an isolated growth chamber under a 12-h photoperiod. The growing temperatures ranged from 18 °C (night) to 20 °C (day). The hydroponic growth solution was prepared using MaxiBloom Hydroponics Plant Food (General Hydroponics) at a concentration of 1.7 g l⁻¹. In preparation for PacBio HiFi sequencing, HMW DNA was extracted from 72-h dark-treated leaf samples using a CTAB-Qiagen Genomic-tip protocol as described previously[42]. DNA quantification and purity were checked using a Qubit dsDNA HS assay and a NanoDrop spectrophotometer, respectively.

### Short-read sequencing

Six assemblies included in PanOat have already been published: *A. insularis* (BYU209), *A. longiglumis* (CN58138) and Sang[4]; *A atlantica* and *A. eriantha*[15]; and OT3098, which was made available as a free resource by PepsiCo in 2020 and was later improved to pseudomolecules (https://wheat.pw.usda.gov/GG3/graingenes-downloads/pepsico-oat-ot3098-v2-files-2021, PRJEB76239 and PRJEB46951).

The *A. byzantina* PI258586 contig assembly was assembled using the TRITEX[43] pipeline and Hi-C data from the Dovetail Omni-C platform. Genome assemblies for Gehl and AC Nicolas were scaffolded using TRITEX and Hi-C to guide pseudomolecule assembly[43].

### PacBio HiFi sequencing

**DNA library preparation and PacBio HiFi sequencing.** HMW genomic DNA was sheared to 17 kb on a Diagenode Megaruptor and then converted into SMRTbell adapted libraries using SMRTbell Express Template Prep Kit 2.0. Size selection was performed using a Sage Blue Pippin to select fragments greater than 10 kb. These were then sequenced at the Brigham Young University (BYU) DNA Sequencing Center, except for the 'Aslak' accession, which was sequenced at the DNA Sequencing and Genomics Laboratory, Institute of Biotechnology, University of Helsinki, using the Sequel II Sequencing Kit 2.0 with Sequencing Primer v.5 and Sequel Binding kit 2.2. Run times were 30 h with adaptive loading, following PacBio SMRT Link recommendations.

### Hi-C sequencing

In situ Hi-C libraries were prepared from young seedlings according to the previously published protocol, using DpnII for the digestion of cross-linked chromatin[2] or with a Phase Genomics multi-enzyme mix. Sequencing and Hi-C raw data processing were performed as previously described[44].

### Genome sequence assembly and validation

**Chromosome-scale assembly. TRITEX + Dovetail Omni-C.** Chromosome-scale sequence assembly proceeded in three steps: (i) scaffold assembly using the TRITEX pipeline[45]; (ii) super-scaffolding with the Dovetail HiRise pipeline[46] (v.2.0.5) using Omni-C data; and (iii) arranging super-scaffolds into chromosomal pseudomolecules using TRITEX (https://tritexassembly.bitbucket.io). PE450 reads were merged with BBmerge[47], error-corrected with BFC[48] and assembled with Minia3[49]. Scaffolding and gap filling were done with SOAPDenovo2[50] using MP6 and MP9 data. Super-scaffolds were generated with the Dovetail HiRise pipeline (v.2.0.5) from alignments of Omni-C data to scaffolds. Omni-C reads were aligned to the HiRise super-scaffolds with Minimap2[51]. Alignment records were converted to binary Sequence Alignment/Map format using SAMtools[52] and sorted with Novosort (http://www.novocraft.com/products/novosort/). A list of Omni-C links was extracted from Hi-C alignments using TRITEX scripts. Omni-C links and guide-map alignments were imported to the R statistical environment[53] and analysed further using TRITEX scripts. An initial Hi-C map was generated using the minimum spanning tree algorithm described previously[54]. The assembly and Hi-C map were iteratively corrected by inspecting Hi-C contact matrices, guide-map alignments and physical-coverage Hi-C reads. Sequence files in FASTA format and AGP tables for pseudomolecules were compiled using TRITEX scripts. The pseudomolecules of *A. byzantina* were aligned against the pseudomolecules constructed from a long-read sequence assembly of cv. OT3098. The OT3098 pseudomolecules (v.2)[8] were downloaded from GrainGenes[55].

**PacBio HiFi.** PacBio HiFi reads were assembled using hifiasm (v.0.14.1)[56] and the TRITEX pipeline[45] was used for pseudomolecule construction. Chimeric contigs and orientation errors were identified through manual inspection of Hi-C contact matrices[43]. GMI423 was used as the reference to map HiFi contigs, using a reduced single-copy genome.

Note on chromosome 7D: while assembling chromosome 7D, we noticed that, when aligning several of our genotypes to GMI423, there was a large, approximately 450-Mb inversion and a small approximately 50-Mb sequence with the same orientation as GMI423. We decided to flip the long sequence to the same orientation as GMI423 and flip the small sequence to the inverted orientation, thinking that the small segment was translocated from one end of the chromosome to the other. Retrospectively, this was a mistake. The more plausible explanation would be an inversion of the large sequence, as supported by several genetic studies[57] showing a distinct lack of recombination in this region.

**Multidimensional scaling (MDS) of Global Oat Diversity GBS data.** A total of 9,111 lines from the Global Oat Diversity (G.O.D.) collection[14] were analysed together with the PanOat assemblies. All of the GBS sequencing data from the G.O.D. lines were aligned to a single reference genome (GS7) using BWA followed by sorting using NovoSort and indexing with SAMtools. The PanOat assemblies were aligned to the GS7 genome. We simulated short reads (tenfold coverage) using the fastq generator (https://github.com/johanzi/fastq_generator) and mapped these to GS7 using Minimap2[51], followed by sorting using NovoSort and indexing with SAMtools. The resulting mapping files from the PanOat assemblies and the G.O.D. were merged into a VCF file using bcftools mpileup[58] with filtering for Q40 or larger and a minimum of 50% missing data per position or SNP.

MDS analysis was preformed using PLINK[59] (www.cog-genomics.org/plink/1.9/) with –maf=0.05 and a maximum of 70% missing data. Results were plotted in R using the ggplot2 package.

### Single-copy pangenome

A single-copy pangenome was constructed as described previously[60] (https://bitbucket.org/ipk_dg_public/barley_pangenome/), with one modification. MMSeq2[61] was used with the option ' --cluster-mode' instead of BLAST for all-versus-all alignment. A minimum sequence identity of 95% was required to accept matches. To estimate the pangenome size, the lengths of the largest sequences in each cluster were summed up.

### PanOat transcriptome sequencing

**Plant materials, growing conditions and tissue dissection.** A subset of 23 PanOat genotypes was selected for transcriptome sequencing (Supplementary Table 2). RNA was extracted from six tissues (Extended Data Fig. 2a–c and Fig. 2a). The 23 genotypes were grown in 6 sets for sampling each tissue separately. Each set comprised at least nine biological replicates (different plants) per oat genotype. Every set with replicates was grown in a separate unit of the growth facility and allocated randomly using the 'sample' function in the R statistical environment[62]. Sampled tissues from 3 different plants (3 technical replicates) were pooled into one tube, making one biological replicate, and this process was repeated twice more to collect a total of 3 biological replicates for each of the 23 PanOat genotypes chosen and the 6 selected tissues (Extended Data Fig. 2a–c).

**Embryonic tissues.** Seeds were sterilized in ethanol (70%) and sodium hypochlorite (50%), then rinsed five times in sterile water, followed

by germination of dehulled seeds in Petri dishes (50 mm; covered in two layers of aluminium foil to maintain darkness) in a growth chamber under constant temperature (about 18 °C), humidity (about 75% relative humidity) and 16-h days. Parts of the coleoptile, mesocotyl and seminal roots were dissected from germinating seeds starting from four days after germination (Extended Data Fig. 2b). These were promptly frozen in liquid nitrogen and stored at −80 °C and thawed before RNA extraction.

**Leaf tissue.** Seedlings were germinated from sterilized seeds as above, but larger Petri dishes (120 mm) were used. Seedlings were grown until two leaves had emerged. Then, the middle part of the leaf blade was dissected for RNA extraction.

**Root tissue.** Seedlings were grown in a small pot on a perlite substrate until three leaves had emerged (Extended Data Fig. 2b). Roots were then separated from the perlite and rinsed in sterile water. Cleaned roots were dissected from the top parts of the plants and stored at −80 °C until RNA extraction.

**Stem tissue.** Plants were grown in pots (one seed per pot) on a soil substrate in a greenhouse chamber with constant temperature (20 °C) and semi-controlled light conditions (16-h light period) until the main stem and four tillers had developed (Extended Data Fig. 2b). Two-millimetre-wide stem discs were dissected from the internode elongating below the flag leaf.

**Panicle tissue.** Plants were grown as above until the main stem and five tillers had developed (Extended Data Fig. 2b). A developing panicle with a size not longer than 15 mm was dissected from the main tiller.

**Caryopsis tissue.** Plants were grown as above until the phenophase in between kernel water ripe with no starch and early milk (Extended Data Fig. 2a,b). This phenophase is recognized to happen four days after anthesis[63].

**RNA extraction.** Total RNA from embryo tissues, leaves, roots, stem and developing panicle was extracted using the RNeasy Plant Mini Kit (QIAGEN). Total RNA from developing caryopsis tissues was extracted using the RNeasy PowerPlant Kit (QIAGEN), according to the manufacturer's instructions. Before RNA extraction, all samples were digested using RNase-free DNase (QIAGEN). Tissue samples were thawed and processed in a random order. Extracted RNA was diluted in 100 µl of buffer and checked for degradation, quantity and purity. RNA integrity was checked using an Agilent Bioanalyzer. Purity (absence of contaminating proteins) was checked by measuring the fluorescence absorbance of nucleic acids at 260 nm and 280 nm using a NanoDrop spectrophotometer. RNA amounts were determined using a Qubit fluorometer (Thermo Fisher Scientific). Average RNA integrity numbers (RINs) varied from 7.62 in leaf tissues to 9.50 in developing panicles and stem tissues. RIN was, on average, lower in leaf tissues but varied little between samples. Only pure RNA samples with high RIN scores (greater than 8.5, except leaves) and sufficient concentration were used for further processing.

**Illumina RNA-seq.** Sequencing libraries were prepared for 432 high quality total RNA samples (RIN > 7.62). First, 500–1,000 ng of total RNA were Poly(A+) enriched, then RNA-seq libraries were produced using the CORALL mRNASeq V1 kit according to the manufacturer's instructions (Lexogen) For each library, barcoding was done using unique dual indices (UDI). To avoid any experimenter's bias, the preparation of the libraries was done randomly. Sequencing was done in 8 pools, with each pool containing 54 randomized single libraries in equimolar amounts. Before sending the pools to the sequencing facility, each pool was sequenced on the iSeq 100 benchtop sequencer at the Natural Resources Institute Finland (LUKE) for quality control. Paired-end sequencing (2 × 150 bp) was done on a Novaseq 6000 device (Illumina) distributed on two full S4 flow cells at the Finnish Functional Genomics Centre in Turku, Finland. Sequencing (2 × 150 bp) of nine libraries was repeated on a NextSeq 550 device (Illumina) in the genomics laboratory at LUKE.

The total number of raw reads per sample and the BioSample IDs are provided in Supplementary Table 17.

**PacBio Iso-Seq.** For each genotype, total RNAs from all tissues and replicates from the respective genotypes were pooled, with between 1,623 ng and 2,001 ng of pooled RNA used for each library. In total, 24 full-length cDNAs were prepared using the TeloPrime Full-Length cDNA kit (Lexogen). Different from the manufacturer's protocol, the purification of 100 µl of cDNA was done with 86 µl ProNex beads (Promega), the standard size selection was done according to the Iso-SeqTM Express Template preparation protocol (PacBio) and no enrichment for shorter or longer transcripts was used. Owing to the 5′ cap specificity of this method, only full-length, double stranded cDNA was obtained. The cDNAs ranged in size from 1,000 to 5,800 bp, with mean peak values between 1,845 bp in HiFi and 2,832 bp in GMI423. After purification, the cDNAs were quantified with Qubit (Thermo Fisher Scientific). According to the Iso-Seq Express Template preparation protocol (PacBio) the amount of cDNA should be in the range of 160–500 ng for Sequel II systems. Libraries with a lower amount of cDNA were re-amplified following the PacBio guidelines. After DNA damage repair, end repair, A-tailing, overhang adapter ligation and clean up, the concentrations were checked using Qubit. The quality was verified using a Bioanalyzer (Agilent). Twenty-four cDNA SMRTbell libraries with a mean fragment length distribution between 2,155 bp and 3,557 bp were transferred to the sequencing facility. The Iso-Seq SMRTbell libraries were sequenced at BYU, each library in a separate Sequel II run. Numbers of reads and total read lengths are provided in Supplementary Table 1.

### Annotation of protein-coding genes

For the 23 oat lines with native transcriptome data generated in this study (Supplementary Table 2), we performed de novo structural gene prediction, confidence classification and functional annotation, following a protocol described previously[17]. The strategy applied in this study differs only in the use of TE soft-masked genome sequences instead of TE hints (see 'Repeat-masking for gene detection'). We applied the same gene prediction procedure for the de novo annotation of the lines A. sterilis TN4 and A. byzantina PI258586 using transcriptome data as evidence. Gene predictions for the lines OT380, A. sterilis TN1, A. fatua CN25955, A. eriantha BYU132/CN19328 and A. atlantica Cc7277, which had no native transcriptome data, were done using a gene consolidation approach that has been described previously[16]. Here, the gene predictions for all 30 oat lines described above were cross-mapped with the genome sequences of one another to identify and correct for missed gene models and to annotate genomes without native transcriptome data.

Finally, for the three lines Leggett, Williams and AC Morgan, we predicted their gene content using the projections of the aforementioned evidence-based gene models to their genomic sequences. The principle of the projection method is described in https://github.com/GeorgHaberer/gene_projection; applied parameters of the workflow and code have been deposited in the directory 'panoat' in the parent directory.

### Repeat-masking for gene detection

To minimize the inclusion of transposon-related gene models, the genome assemblies were soft-masked for transposons (TEs) before gene detection. The TE library used, developed for the oat reference genome[4], masked approximately 60% of the assembly for each line. Soft-masking was performed using vmatch (anaconda.org/bioconda/vmatch, v2.3.0) with the following parameters: '-l 75 -identity 70 -seedlength 12 -exdrop 5 -d -p -qmaskmatch tolower'.

### Construction of the oat core, shell and cloud genomes

Phylogenetic HOGs based on the primary protein sequences from 30 oat lines with consolidated gene predictions were calculated using

OrthoFinder v.2.5.5[20] with standard parameters (see 'Annotation of protein-coding genes' for details; Leggett, Williams and AC Morgan were not part of this orthologous framework, because their gene content was not consolidated). Before the analysis, input sequences were filtered for transposon- and plastid-related proteins and proteins encoded on unanchored contigs were discarded for this analysis.

Depending on the focus of the analyses, we treated each of the subgenomes of hexaploid and tetraploid oat lines either as individual entities or, for our analysis of core, shell and cloud genomes, as parts of the single lines.

The scripts for calculating core, shell and cloud genes have been deposited in the repository https://github.com/PGSB-HMGU/BPGv2.

Core HOGs contain at least one gene model from all 30 compared oat lines. Shell HOGs contain gene models from at least 2 oat lines and at most 29 oat lines. Genes that were not included in any HOG (singletons), or were clustered with genes only from the same line, were defined as cloud genes.

GENESPACE[64] was used to determine syntenic relationships between the chromosomes of all 30 oat lines.

## Protein functional annotation and gene-set enrichment analysis

For functional enrichment analysis in the identified expression level categories, Mercator4[65] (v.6.0) protein functional annotation was performed for the identified 5,291 single-copy HOGs across the 20 *A. sativa* lines, which yielded 4,682 protein annotations (Supplementary Table 18). These annotations were used to test enrichment using over-representation analysis (ORA) of sets of genes associated with expression level categories with the R package clusterProfiler[66] (v.4.6) and a Benjamini–Hochberg FDR correction $P$ value cut-off of 0.05.

Similarly, for ORAs across genes classified into core, shell and cloud categories, all of the 31 oat lines' proteomes (2,869,876 proteins and 131,729 HOGs) were functionally annotated with Mercator4 (v.6.0). This resulted in a total of 53,018 annotated HOGs for the core (8,325 annotated HOGs), shell (32,108 annotated HOGs) and cloud (12,585 annotated HOGs) categories, applied as universe in the enrichment analyses. ORAs showed enrichment across multiple Mercator4 hierarchical categories (labelled as levels 1–7) (https://hmgubox2.helmholtz-muenchen.de/index.php/s/Y3wWa7bn2rayEqw).

## Gene-expression analyses

For the analysis of gene expression, RNA-seq data from 23 oat varieties (Supplementary Table 2) were processed using Fastp[67] (v.0.24.1) for trimming, followed by quality assessment and outlier detection. The data for each line were aligned to the relevant reference genome using Kallisto[68] (v.0.48; Supplementary Figs. 2 and 3) and normalized to transcripts per million (TPMs) using Deseq2's tximport function[69,70]. All RNA-seq data were also aligned to the GS7 reference genome for specific analyses.

To compare the expression levels across different subgenomes (A, C and D), gene-expression data from six tissues (leaf, embryo, root, internode, panicle and caryopsis) were examined.

We focused on high-confidence genes and for each gene, the expression value was normalized using a log transformation (log(value + 1)) to stabilize variance and ensure that the data were suitable for statistical comparisons. The log-transformed expression values were aggregated by calculating the mean expression level for each line, tissue and subgenome combination. To determine whether the expression levels between different subgenomes were significantly different, Mann–Whitney $U$ tests were performed for each tissue type. Comparisons were made between each pair of subgenomes (A versus C; A versus D; and C versus D).

For the purpose of identifying genes with stable versus variable expression, the analysis was limited to 20 *A. sativa* varieties and 5,965 HOGs. These HOGs were characterized as single-copy orthologues with an A:C:D ratio of 1:1:1 across all 20 varieties, providing a standardized basis for comparison. HOGs were deemed stable if 90% of the varieties exhibited the same expression category; otherwise, they were classified as variable.

In the set of 5,965 '60-lets', expression levels were categorized into one of 7 categories on the basis of the Euclidean distance to 7 ideal expression level profiles: A-, C- or D-dominant or A-, C- or D-suppressed, in which one gene is predominantly expressed or suppressed, and a balanced category, in which A, C, and D genes are equally expressed, as outlined previously[4].

## Analysis of differences in gene expression between diads and triads

We focused on the 20 *A. sativa* lines and selected genes that had either one single-copy homeologue in each of the subgenomes (so forming complete triads: A:C:D 1:1:1 constitution) or a constitution in which one triad member was missing (so forming diads: A:C:D 1:0:1, 1:1:0 or 0:1:1). The lines were then categorized into groups with complete triads and diads, while ensuring uniformity in the missing pattern across the lines. This approach allowed for a controlled comparison across genetic backgrounds. To ensure robust statistical analysis, specific filtering criteria were applied. Each group analysed was required to consist of at least five lines, allowing us to achieve sufficient statistical power.

We used an unpaired $t$-test to assess the significance of expression differences between groups with a missing homeologue and those with complete triads. Furthermore, Cohen's $d$ was used to determine the directionality of these differences. A chi-squared test was used to compare the frequency of significant compensatory expressions across the different subgenomes.

## Analysis of changes in gene expression at translocation break points

To assess gene-expression changes associated with translocation break points, we first identified the break points on the basis of the GS7 genome by aligning the chromosomes of each oat line to GS7 using Minimap2[51]. We then extracted 100 syntelogues on either side of each break point using GENESPACE[64].

For gene-expression profiling, RNA-seq data from all oat lines were mapped to the GS7 transcriptome using Kallisto[68], and differential gene-expression analysis was performed with DESeq2[69]. Gene expression was compared across multiple tissues between lines carrying each translocation and those without.

To determine whether the regions surrounding translocation break points were significantly enriched for differentially expressed genes relative to the rest of the chromosome, we applied a hypergeometric test. This statistical test assesses whether the observed number of DEGs in the translocation regions exceeds what would be expected by chance, given the total number of genes and DEGs on the chromosome. Statistical significance was determined using an FDR-adjusted $P$ value (Benjamini–Hochberg correction, $\alpha = 0.05$).

## Comparative gene-expression analysis in ND060432

Total RNA was extracted from young leaf blades of ND060432 in four biological replicates and sequenced using the same protocol as was used for the other PanOat leaf samples. RNA-seq reads from the leaf samples of GS7 and ND060432 were processed with Fastp[67] (v.0.24.1) for trimming and quality assessment. Processed reads were aligned to the GS7 reference transcriptome using Kallisto[68] (v.0.51.1) and normalized to TPMs. Expression values of biological replicates were summarized to their median value. To investigate whether the substitution of chromosome 6D with 6A affects gene expression, we compared the expression ratio of A- to D-subgenome genes within each chromosome group, with a particular focus on changes on chromosome 6D. Each chromosome was divided into 20 bins that were proportional in size to the size of the chromosomes. The expression was then summarized to the average value per bin. Statistical comparisons were performed using the

Wilcoxon rank-sum test with FDR correction for multiple comparisons in the seven chromosome pairs. Significance levels were denoted as: ***$P < 0.001$, **$P < 0.01$, *$P < 0.05$, NS, not significant. Effect size was calculated as $r = |Z|/\sqrt{N}$, where $Z$ is the standardized test statistic and $N$ is the total sample size ($n = 40$ per comparison: 20 bins per chromosome × 2 chromosomes; Supplementary Table 16).

To further study the effect of the replacement of chromosome 6D with chromosome 6A in ND060432, subgenome expression bias categories were quantified. Therefore, triads were identified in each of GS7 and ND060432. Expression levels of triads overall and in each chromosome group were categorized into one of seven categories on the basis of the Euclidean distance to seven ideal expression level profiles: A-, C- or D-dominant or A-, C- or D- suppressed, in which one gene is predominantly expressed or suppressed, and a balanced category, in which A, C and D genes are equally expressed, as outlined in a previous study[4]. Because oat has many translocations and triad genes can be located on chromosomes of different chromosome groups, triad membership to a certain chromosome group was determined according to the location of the D-subgenome gene to investigate the effect of the replacement of chromosome D. In addition, ternary plots were generated to visualize these subgenome expression patterns across all triads and within chromosome groups 1–7. These plots illustrate the distribution of expression bias among the three subgenomes, where points near the vertices indicate subgenome-specific dominance and points near the centre represent balanced expression. Statistical significance of the bias in the patterns was assessed using chi-squared tests for each chromosome group (Supplementary Table 16).

### WGS of the North American spring oat collection

**Sequencing, mapping and GWAS of CORE samples.** A total of 295 North American spring oat accessions from the CORE population[32] (Supplementary Table 13) were sequenced using an Illumina Novaseq 6000 (paired-end 150 bp) with a mean read depth of 4.58 per accession.

The genome assembly of GS7 was chosen as a reference because it is a North American variety with a long-read assembly. The adapter sequence ('AGATCGGAAGAGC') was removed using cutadapt[71] and then all reads were aligned to the GS7 reference genome using Minimap2[51], sorted with NovoSort (https://www.novocraft.com/products/novosort/) and converted to a compressed reference-oriented alignment map (CRAM[72]) file using SAMtools[58]. A VCF file was created using bcftools 'mpileup'[58], including all variations with mapping quality higher than 40. Read-depth variation was determined by counting how many reads were aligned in 1-Mb windows along the genome.

For GWAS, a $k$-mer-based reference-free pipeline was used (kmerGWAS)[73]. The phenotypes used included heading date, plant height and grain yield, collected from ten locations in two consecutive years (2010 and 2011 (ref. 32); Supplementary Fig. 4). $k$-mers with significant association ($-\log_{10}$ threshold for 10% family-wise error rate) were mapped to the GS7 genome.

**PanOat assemblies.** To align the PanOat assemblies to the GS7 genome, we simulated short reads (tenfold coverage) using fastq generator (https://github.com/johanzi/fastq_generator) and mapped these to GS7 using Minimap2[51]. The resulting mapping files were merged into a VCF file together with all 295 CORE genotypes using bcftools merge[58].

**Genome alignments.** Whole-genome alignments of complete pseudomolecule assemblies were performed using Minimap2[74] with the -f 0.05 option to filter out repetitive minimizers and speed the alignment process. Visualization was done using NGenomeSyn[41] as shown in Fig. 1c.

**PCA of WGS data.** Focusing only on SNPs on chromosome 7D, PCA analysis was done with all 295 genotypes and PanOat assemblies using PLINK with −maf=0.05 and a maximum of 70% missing data.

SNP haplotypes were analysed using a custom Perl script[75] (https://github.com/guoyu-meng/barley-haplotype-script/tree/main/05.other/SNP_haplotype_plot), then sorted by the predicted inversion state from the PCA analysis and plotted in R.

### Reciprocal translocation 2A/2C in Australian and Canadian oat
**Plant materials.** The oat varieties used in this study were part of a worldwide oat germplasm collection from the Western Crop Genetics Alliance at Murdoch University (Supplementary Table 14), and included 564 lines from 41 countries. To track down the pedigree of the chromosome 2A/2C translocation, potential parental lines for the varieties with the translocation were collected for the second-round test with 32 varieties (Supplementary Table 15.2). Seeds of the oats were grown in pots in a glasshouse, with natural lighting cycles and regular watering. Leaves from three-week-old seedlings were collected for DNA extraction.

A RIL population was derived from Bannister × Williams crossing. An $F_5$ population was grown in the greenhouse at InterGrain. A total of 188 lines, together with their parents, were used for molecular map construction using DArTseq technology, following the online instruction (Diversity Arrays Technology). Three to five centimetres of leaf sections were collected for DNA extraction from seedlings at the three-week stage.

**Chromosome-specific molecular markers for 2A/2C translocation.** Genomic DNA was extracted from the leaves of each oat line using the cetyl trimethyl ammonium bromide (CTAB) method[76]. DNA quality was assessed on 1% agarose gels and quantified using a NanoDrop spectrophotometer (Thermo Fisher Scientific). DNA was diluted to 50 ng μl$^{-1}$ for PCR.

Two DNA samples (OT207 and Kanota) used in this study were obtained from Agriculture & Agri-Food Canada, owing to the unavailability of these varieties in Australia. Specific primers for PCR (Supplementary Table 15.1) were based on sequences from the Australian oat Bannister and the Spanish oat FM13. Primers Ban2A_F3 and Ban2A_R3 are specific to the Bannister chromosome 2A break point, producing a PCR amplicon of 397 bp. Another pair of primers, Ban2C_F3 and Ban2C_R3, is specific to the Bannister chromosome 2C break point, producing a PCR amplicon of 595 bp. PCR was done in 10-μl reactions in 384-well PCR plates (Axygen) containing 50 μM of each of the three primers, 200 μM dNTPs, 1.5 mM MgCl$_2$ and BIOTAQ (Bioline Australia) in a Veriti Thermo cycling machine. The PCR products were separated and visualized on 2% agarose gels stained with GelGreen (Biotium). Oat lines containing the 2A/2C translocation produced bands and were scored as 'present', whereas normal oat lines did not produce bands and were scored as 'absent'.

**Genetic map construction.** The RIL lines were genotyped with DArTseq markers. The genotypes were filtered with the following parameters: call rate higher than 90%, polymorphic information content higher than 0.2 and heterozygous frequency less than 0.6. MSTmap[77] (https://github.com/ucrbioinfo/MSTmap) was used to construct the genetic map[77]. Several rounds of calculations were performed to correct and impute genotype calls. After the first round of genetic map construction, the markers were sorted on the basis of the genetic map and considering the physical orders. Missing data and noisy markers were corrected if the physical and genetic orders were consistent. The heterozygous regions were fixed first, and then the nearby markers were corrected in subsequent rounds of calculation. The final genetic map was calculated after three to four rounds of corrections. The chromosomal 2A/2C translocation-specific molecular markers were manually integrated into the molecular linkage map.

**Validation of chromosome 2A/2C fragment deletion in the RIL population of Bannister/Williams.** The whole-genome DarT seq method identified ten RILs with potential chromosome fragment deletions

(Supplementary Table 15.3), but only three RILs had set seeds. On the basis of DArT marker sequences, PCR primers for different locations on chromosome 2A or 2C were designed to validate the truncations seen using agarose-gel-based methods. The primer sequences and locations (aligned FM13 genome sequences) are listed in Supplementary Table 15.4. Each pair of primer sequences or a marker is unique and only has one specific amplicon.

PCR and gel analysis was performed as described above. With this set of primers, a score of 'present' indicated that the DNA sequence at a particular locus was normal. Those scored as 'absent' indicated mismatches with the primers, or that a sequence was missing. When a few consecutive loci on the same chromosome are all scored as absent, it is most likely that the section of the chromosome has been replaced or is truncated.

Sixteen markers scattered across chromosomes 2A and 2C were tested in three lines (BW041, BW080 and BW123). The three or four plants from each pot were verified to have the same genetic background. BW041 and BW080 are likely to be truncated on 2C from 412,337 kb. The tip of chromosome 2A from BW123 is missing; the truncation point is likely to be between 2A 26,716 kb and 344,891 kb. The deleted fragments were further validated by 10x whole-genome shotgun sequencing.

**Yield evaluation.** Grain yield data were obtained from the Australian NVT from 2018 to 2022. Each year included 19 to 31 trials across Australia, with 158 trials designed in 3 replications. There were six 2A/2C translocated varieties and 11 non-translocated varieties.

NVT is an Australian national program that provides field-collected information comparing yield performance and grain quality on commercially available grain varieties, including barley, wheat and oat. Detailed trial locations are in the oat-growing areas across the whole continent (Grains Research and Development Corporation, https://nvt.grdc.com.au). Trials follow a standard protocol to facilitate yield evaluation and comparison (https://nvt.grdc.com.au/_data/assets/pdf_file/0023/613166/NVT-PROTOCOLS-v2.0.1_FINAL.pdf).

Trial designs use statistical methodology and allow for site and subsequent across-site analysis. Trial sites were located throughout the Australian grain-growing region. Each trial had four to eight repeated cultivar entries yearly to allow connectivity among trials. Each cultivar entry was replicated in three randomly placed plots. Plot size was standard at 1.2 m by 6 m. Seed sowing and trial management followed the best local agronomic practice. Each trial was harvested at the earliest opportunity after physiological maturity to minimize grain losses through wind, insect, rain or pest damage. The field plot yield was calculated from the harvested grain. Grain yield was adjusted with linear mixed models, in which variance parameters in the mixed models were estimated using the residual (restricted) maximum likelihood (REML) procedure with ASReml-R (v.4.1.0; https://asreml.kb.vsni.co.uk/wp-content/uploads/sites/3/2018/07/ASReml-Package.pdf). Spatial variations were examined, including local autocorrelations, global trends and extraneous variations. The blocking structure of the experiments was fitted as random effects. Spatial trends and residual variances with two-dimensional auto-regressive correlation at first order for rows and columns were examined and fitted when they were statistically significant. Statistical tests were used to examine the levels of significance, including likelihood ratio tests for random effects, and conditional Wald tests for fixed effects. Residual diagnostics were performed to examine the validity of the model assumption of independence, normality and homogeneity of variance. Both the empirical best linear unbiased predictions (eBLUPs) estimated means for varieties, with variety fitted as random effects, and the empirical best linear unbiased estimations (eBLUEs) estimated means for varieties, with variety fitted as fixed effects, were produced from respective fitted models. Trials with an average grain yield of less than 1,000 kg ha$^{-1}$ were deemed as abnormal and removed from further analysis.

To account for variability between trials and years, we recalculated the variety yield as the ratio to the average yield of all varieties (17) within the trial, following:

Variety yield (relative to site average) = Variety trial yield/average yield within the trial

The average variety yield (relative to site average) within a trial of translocated and non-translocated varieties across all trials was compared using a two-sided $t$-test with SPSS v.29 (IBM). Significance was taken as $P < 0.05$.

Because yield advantages are known to be associated with reduced plant height, we also examined the height of varieties with or without the 2A/2C translocation. In a common-garden experiment, the oat varieties were grown in South Perth (Western Australia) for three years. The height was measured from the tip of the panicle to the ground (detailed experiment and results are to be published in a separate paper). To account for variability between years, we calculated the height as the ratio to the average height of all (17) varieties. The height (relative to the site average) of translocated and non-translocated varieties across all trials was then compared using a two-sided $t$-test with SPSS v.29 (IBM). Significance was taken as $P < 0.05$.

**QTL mapping.** Plant height was measured before grain harvest. The genotypic and quantitative trait data were formatted for use with MapQTL5.0. A permutation test was performed to calculate the logarithm of the odds (LOD) value threshold for positive QTL detection. Internal mapping was first performed to identify the markers with the highest LOD value above 3.2. The markers with the highest LOD values from different QTLs were selected for multiple QTL mapping analysis.

**Large-scale chromosomal rearrangements shaped segregation and recombination patterns in progenies of 13 crosses from a working oat breeding program**

Thirteen $F_1$ crosses were made in 2019 at the oat breeding program at the Ottawa Research and Development Centre, Agriculture and Agri-Food Canada (AAFC) (Supplementary Table 19.1), among oat lines that were selected for their excellent trait profiles and adaptation to Canadian environments. Progenies were advanced by a modified single-seed descent method to the $F_6$ generation. Parents and progeny were genotyped using a targeted GBS method[78]. Progenies were filtered to remove those with more than 90% similarity to a parent and to eliminate progeny that showed more than 98.5% similarity to another progeny. The position of tag-level haplotype markers on the Sang reference genome[4] was used to compute recombination fractions ($r$) between all pairs of markers. Values of $r$ were averaged within a sliding window of 20 Mb at 10-Mb increments in two directions of a complete genome matrix, as described previously[31], such that recombination fractions were scaled to physical distance. Recombination matrices across the full genome were displayed as heat maps, coloured from yellow ($r = 0$) to cyan ($r = 0.2$) to purple ($r >= 0.4$) (also in Supplementary Table 19.2). Chromosomes with inadequate marker coverage to estimate recombination were coloured grey.

**C-banding**

All karyotypes of lines throughout the manuscript were determined by C-banding as described previously[79], except that 0.1% colchicine at 20 °C for three to five hours was used to arrest microtubule assembly.

**Reporting summary**

Further information on research design is available in the Nature Portfolio Reporting Summary linked to this article.

**Data availability**

The data generated by the PanOat Consortium have been made freely available and publicly accessible through deposition in public

databases. Sequence data were deposited in the European Nucleotide Archive under project IDs PRJEB56828 (genome assembly raw data), PRJEB57570 (transcriptome sequencing) and PRJEB62778 (WGS resequencing data). Project IDs for individual assemblies and BioSample IDs for individual CORE genotypes are listed in Supplementary Tables 20 and 13, respectively. The annotation datasets are available for download from the USDA-ARS GrainGenes database[55] at https://graingenes.org//GG3/content/panoat-data-download-page. This page also serves as a landing page for access not only to data but also to genome browser tools and BLAST services[80]. Thirty-three genome browsers for each PanOat accession were created in GrainGenes. The links to these genome browsers are available from the data download landing page mentioned above. They are also available from the main GrainGenes Genome Browser landing page at https://graingenes.org/GG3/genome_browser. Each genome browser contains datasets as tracks, which include pseudomolecule sequences, as well as high-confidence and low-confidence gene models. The gene models have external links to the eFP Browser at the University of Toronto (https://bar.utoronto.ca/eFP-Seq_Browser/). BLAST services in GrainGenes include databases that have pseudomolecules from 35 accessions, as well as scaffold sequences from a subset of 6 accessions: BYU960, *A. byzantina* PI258586, Leggett, AC Morgan, OT3098 v2 and Williams. Note that when a BLAST query sequence hits a region of a genome assembly that has a genome browser in GrainGenes, a clickable link to the hit region on that genome browser is made available through the JBrowse Connect API[81], as are other details, such as hit scores, statistics and sequence alignments.

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

**Acknowledgements** We thank J.-P. Schnitzler and G. Gerl for the pictures of oat plants; S. Moisander and S. Kankanpää for their assistance with RNA and DNA extractions; A. Kedonperä, M. Segerstedt, M. Jalli and A. Nissinen for assistance with growing oats under greenhouse conditions and for photo documentation; E. Wilcox for support and expertise with PacBio sequencing; L. Paulin and the DNA Sequencing and Genomics Laboratory, Institute of Biotechnology, University of Helsinki, for WGS of 'Aslak'; and the Natural Resources Institute Finland (LUKE) for financial support. M.M., M. Spannagl and M.H.H. were supported by a grant from the German Ministry of Food and Agriculture (BMEL, FUGE, FKZ: 28AIN02C20). R.A. was supported by the Alexander von Humboldt foundation. Y.C. was supported by the European Union's Horizon 2020 research and innovation program under the Marie Skłodowska-Curie grant agreement no. 847585–RESPONSE. W.A.B., N.A.T. and W.Y. were supported by Agriculture and Agri-Food Canada (AAFC) and by partnership funding contributed by the Canadian Field Crops Research Alliance and the Prairie Oat Breeding Consortium. W.A.B., N.A.T., W.Y., Y.-B.F. and J.S. acknowledge co-funding by Genome Quebec, AAFC and McGill University through a project entitled 'Targeted and useful genomics for Barley and Oat' (Tugboat), and are grateful for sequencing assistance from the Staff at the Centre d'expertise et de services, Genome Quebec. N.K. was supported by the European Research Council (ERC) under the European Union's Horizon 2020 Research and Innovation programme (grant agreement no. 101116452, ERC starting grant project 'RESIST'). E.P. and F.J.C. were supported by grant PID2022-142574OB-I00 funded by MICIU/AEI/10.13039/501100011033, FEDER, UE and Junta de Andalucia (QUAL21_023 IAS). S. Marmon and N. Sirijovski acknowledge funding from the Swedish Foundation for Strategic Research (ScanOats: IRC15-0068) and the two industrial partners of the ScanOats Research Center (Lantmännen and Oatly AB); support from the Swedish National Genomics Infrastructure funded by the Science for Life Laboratory, the Knut and Alice Wallenberg Foundation and the Swedish Research Council; the SNIC/Uppsala Multidisciplinary Center for Advanced Computational Science for providing assistance with massively parallel sequencing and access to the UPPMAX computational infrastructure; and J. Bentzer at Lund University for data management support. C.J.H. and T. Langdon were supported by BBSRC grants BBS/E/IB/230001, BBS/E/W/0012843 and BB/S008195/1. J.I.S. was supported by the Beatriz Galindo Program (BEAGAL18/00115), grant PID2021-123718OB-I00 funded by MCIN/AEI/10.13039/501100011033 and by the ERDF: 'A way of making Europe', CEX2020-000999-S and the Severo Ochoa Program for Centres of Excellence in R&D from the Agencia Estatal de Investigación of Spain, grant SEV-2016-0672. K.T.N. acknowledges technical support from S. Pandurangan and funding from the Prairie Oat Breeding Consortium. A.M.F. was funded by the German Research Foundation (DFG) TRR 356/1 2023 – 491090170. We acknowledge the Finnish Functional Genomics Centre core facility, Turku Bioscience for RNA-seq; the Functional Genomics Center Zurich (Switzerland) for its contribution to WGS of the Hâtives des Alpes accession; the Australian Grains Research and Development Corporation (GRDC) (project code UMU2003-002RTX, under management of M. Groszmann) and the Western Australia Oat Industry Partnership and Department of Primary Industry and Regional Development (POIGP/01), which provided funds for C.L. A. Rattey provided the Bannister/Williams RIL population. The GRDC-supported NVT programme (https://nvt.grdc.com.au/) provided long-term yield data for Australian oat varieties. Funding for the sequencing of *A. sativa* cv. GS7 came from NSF award ABR-PG 1444575. This research was supported by US Department of Agriculture (USDA) Agricultural Research Service project numbers 3060-21000-046-000D, 2030-21000-056-000D, 2050-21000-038-000D and 8062-21000-053-000D. Mention of trade names or commercial products in this publication is solely for the purpose of providing specific information and does not imply recommendation or endorsement by the USDA. USDA is an equal opportunity provider and employer.

**Author contributions** Selection of genotypes: W.A.B., L.B., B.B., F.J.C., S.R.E., K.E.K., Y.-B.F., M.H.H., C.J.H., J.I.S., J.-L.J., E.N.J., T. Langdon, C.L., M.M., P.J.M., K.T.N., E.P.-G., E.P., A.R., J.S., N. Sirijovski, B.S., N.A.T., S. Viitala, S.W. and W.Y. Genome sequencing: P.A., W.A.B., L.B., O.B., H.S.C., Y.C., D.C., V.D., S.R.E., K.E.K., Y.-B.F., R.G., A.H., C.J.H., A.I., J.-L.J., R.K., M.K., T. Langdon, C.L., S. Marmon, P.J.M., K.T.N., E.P.-G., A.R., R.W.R., J.A.S., A.H.S., J.S., M. Singh, N. Sirijovski, N. Stein, B.S., N.A.T., S.W., P.W., A.J.W., X.-Q.Z. and Z.Z. Sequence assembly: R.A., W.A.B., H.S.C., Y.C., H.H., M.M., P.J.M., K.T.N. and B.S. Transcriptome sequencing and analysis: R.A., L.B., O.B., A.M.F., J.D.F., R.G., N.K., T. Langdon, T. Lux, V.M., M.M., P.J.M., R.S.N., A.H.S., S.C.L. and C.H.C. Annotation: H.G., G.H., N.K., T. Lux, V.M., K.F.X.M. and M. Spannagl. Analysis and interpretation of SVs: T.T.A., R.A., W.A.B., B.C., V.D., T.H., E.N.J., Y.J., C.L., R.S.N., N.A.T., P.W., C.P.W., X.-Q.Z. and G.Z. Data management and submission: R.A., L.B., A.F., M.M., S. Michel, R.S.N., A.P., N.J.P., T.Z.S., J.S., M. Singh, S. Viitala, S. Vronces, E.Y. and Z.Z. Writing: R.A., W.A.B., L.B., O.B., V.D., Y.-B.F., J.-L.J., N.K., C.L., T. Lux, M.M., A.H.S., T.Z.S., M. Spannagl, N.A.T., X.-Q.Z. and G.Z. Coordination: W.A.B., L.B., O.B., J.D.F., R.G., N.K., C.L., M.M., P.J.M., A.H.S., T.Z.S., J.S., M. Spannagl and N.A.T. All authors read and commented on the manuscript.

**Funding** Open access funding provided by Leibniz-Institut für Pflanzengenetik und Kulturpflanzenforschung (IPK).

**Competing interests** S.R.E. is an employee of General Mills. A.J.W. is an employee of PepsiCo. N. Sirijovski is an employee of Oatly. The remaining authors declare no competing interests.

**Additional information**
**Correspondence and requests for materials** should be addressed to Nicholas A. Tinker, Jason D. Fiedler, Chengdao Li, Peter J. Maughan, Manuel Spannagl or Martin Mascher.

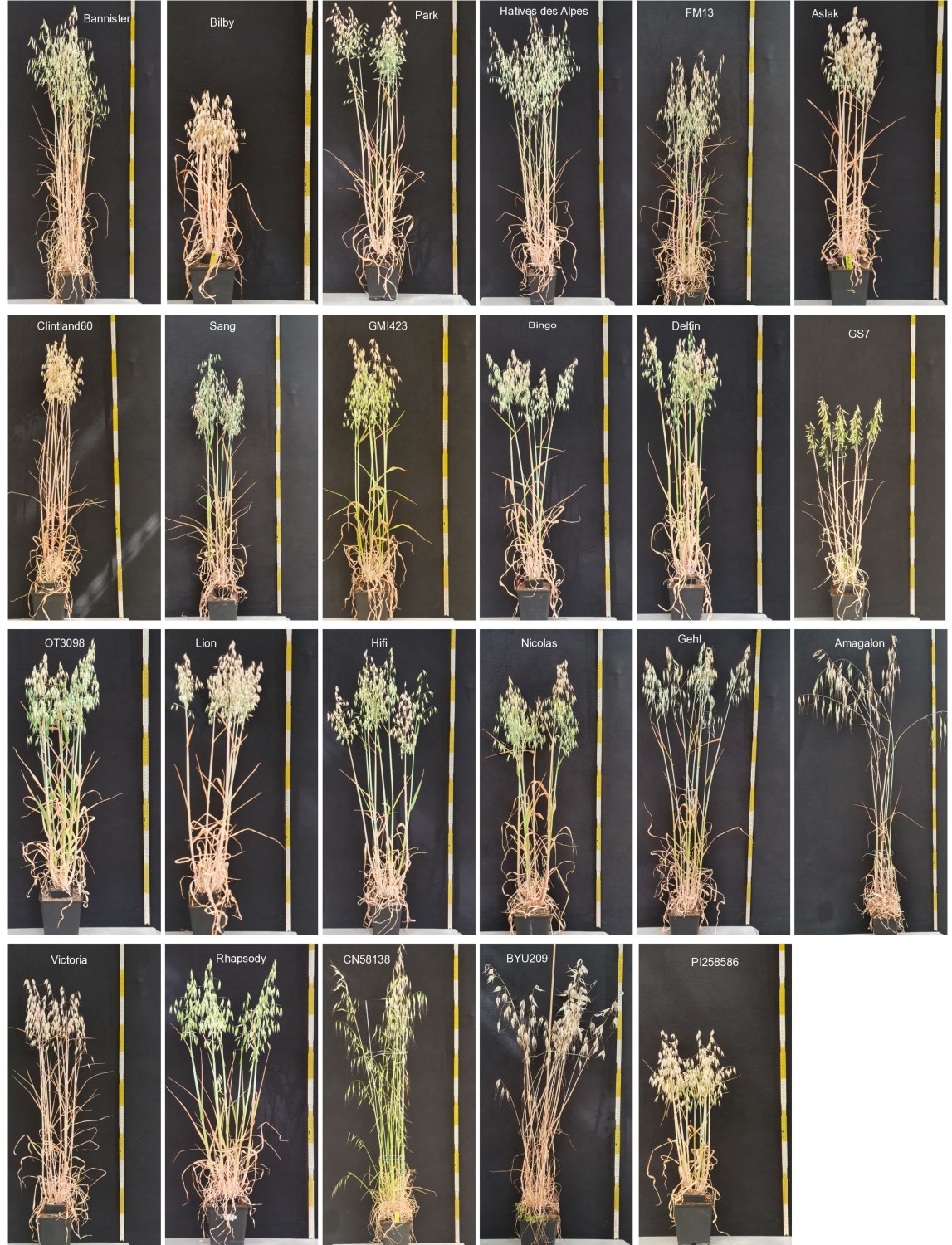

**Extended Data Fig. 1 | Images of PanOat accessions in maturity.** Images showing one representative individual of each of the 23 oat lines for which RNA-seq data were generated. Accessions showing left to right and top to bottom (1) Bannister (2) Bilby (3) Park (4) Hâtives des Alpes (5) FM13 (6) Aslak (7) Clintland60 (8) Sang (9) GMI423 (10) Bingo (11) Delfin (12) GS7 (13) OT3098 (14) Lion (15) HiFi (16) Nicolas (17) Gehl (18) Amagalon (19) Victoria (20) Rhapsody (21) CN58138 (22) BYU209 (23) PI258586. Pictures courtesy of Jörg-Peter Schnitzler and Georg Gerl from Helmholtz Munich.

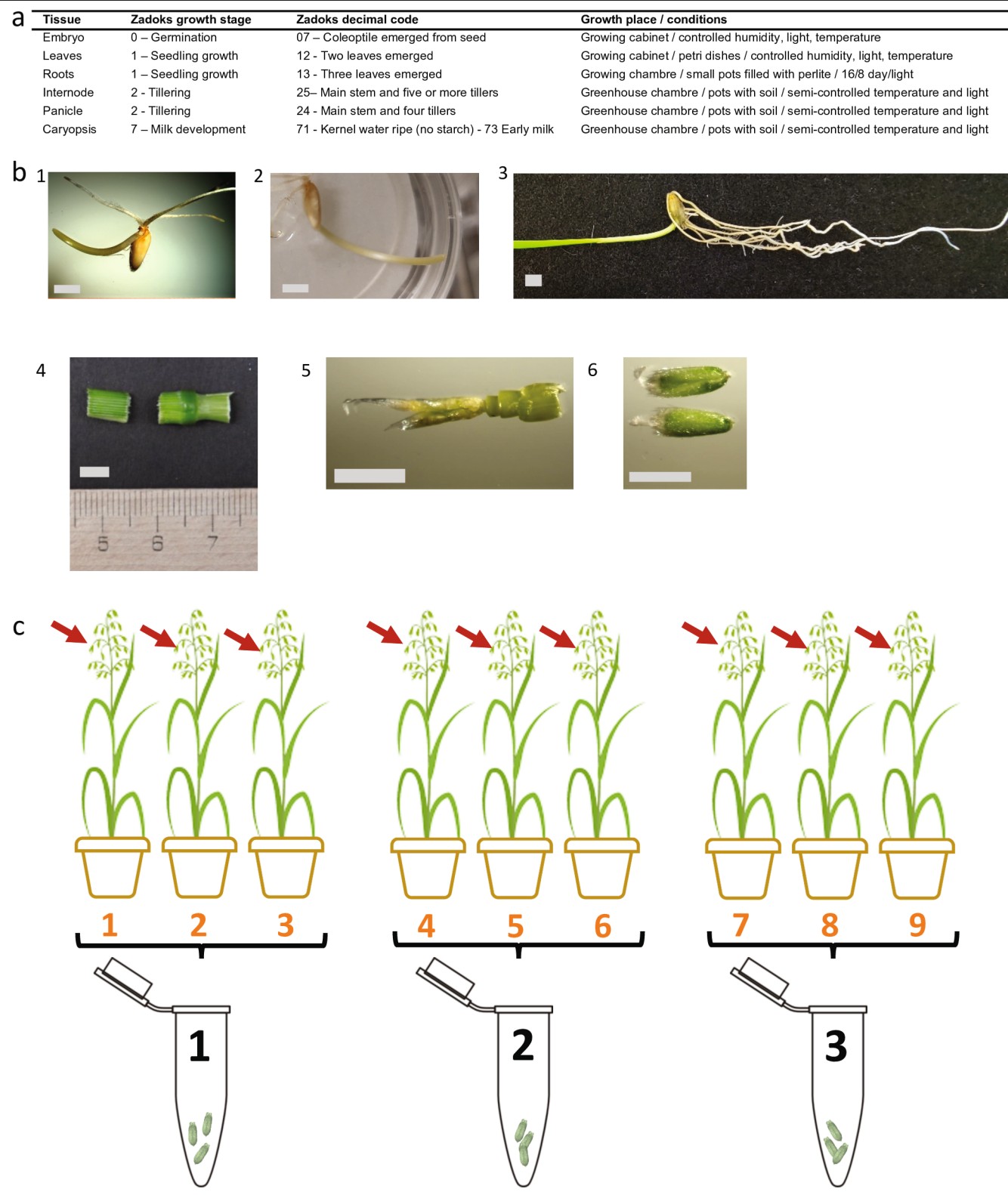

| Tissue | Zadoks growth stage | Zadoks decimal code | Growth place / conditions |
|---|---|---|---|
| Embryo | 0 – Germination | 07 – Coleoptile emerged from seed | Growing cabinet / controlled humidity, light, temperature |
| Leaves | 1 – Seedling growth | 12 - Two leaves emerged | Growing cabinet / petri dishes / controlled humidity, light, temperature |
| Roots | 1 – Seedling growth | 13 - Three leaves emerged | Growing chambre / small pots filled with perlite / 16/8 day/light |
| Internode | 2 - Tillering | 25– Main stem and five or more tillers | Greenhouse chambre / pots with soil / semi-controlled temperature and light |
| Panicle | 2 - Tillering | 24 - Main stem and four tillers | Greenhouse chambre / pots with soil / semi-controlled temperature and light |
| Caryopsis | 7 – Milk development | 71 - Kernel water ripe (no starch) - 73 Early milk | Greenhouse chambre / pots with soil / semi-controlled temperature and light |

**Extended Data Fig. 2 | Tissue sampling and RNA extraction for transcriptome sequencing. a**, List of oat tissues dissected from 24 members of the PanO at panel according to defined growth stages in cereals (Zadoks scale[82]). **b**, Representative photos of sampled tissues in selected PanOat lines: 1. germinating seed (Delfin), 2. seeding leaf (Aslak) 3. seedling root (OT3098), 4. stem (Bilby), 5. developing panicle (GMI423) and 6. immature seed (Aslak). Grey bars = 0.5 cm. **c**, Sampling scheme used to collect all tissues for RNA-Seq and Iso-Seq sequencing.

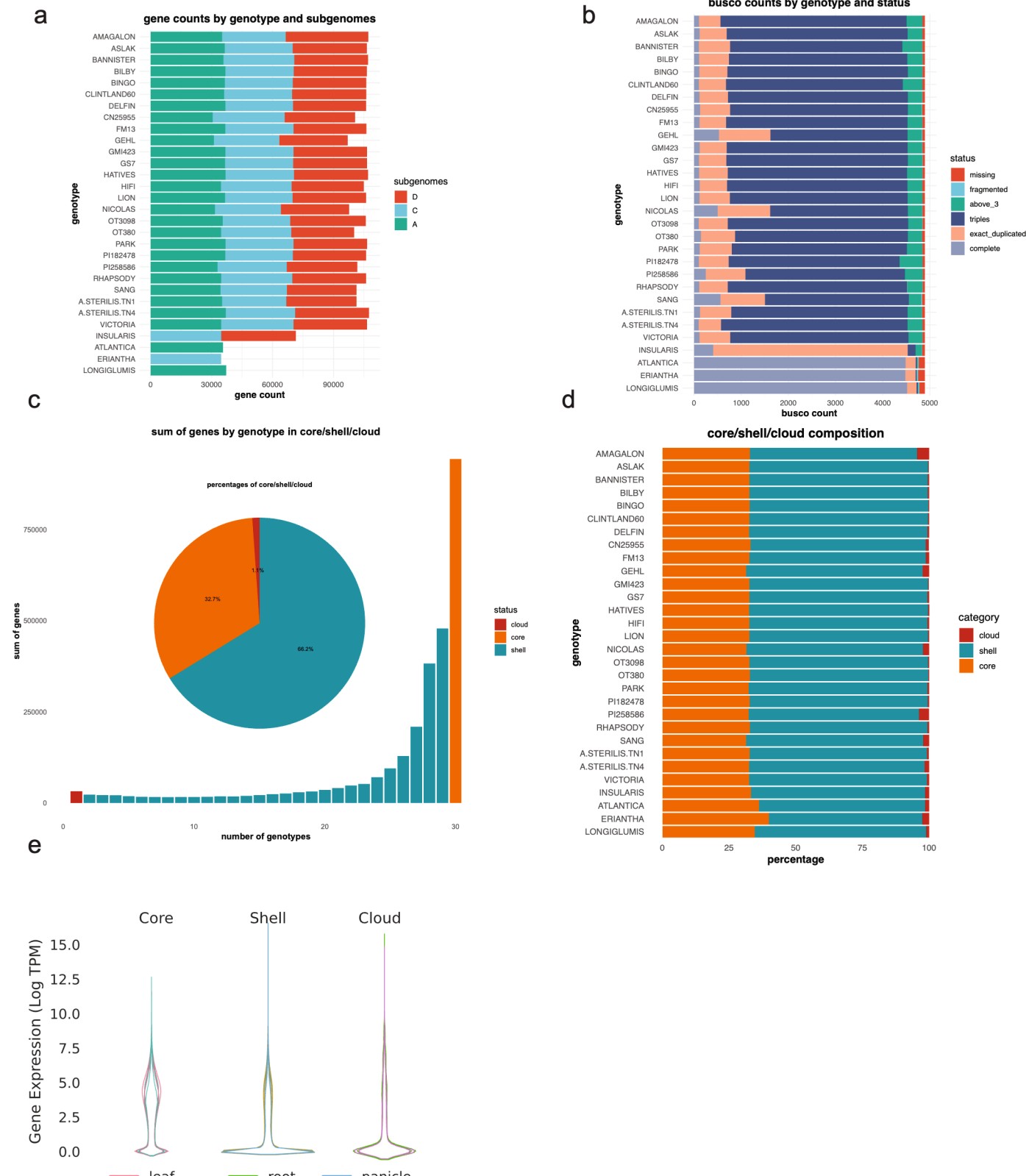

**Extended Data Fig. 3 | Compartments of the oat pangenome. a**, Number of genes predicted for the respective oat line, detailed by subgenome. **b**, Completeness of gene predictions assessed by BUSCO counts. **c**, The bar chart shows the number of genes (y-axis) found in orthologous groups consisting of one oat line (cloud; red), between two and twenty-nine oat lines (shell; turquoise) or thirty oat lines (core; orange) (x-axis). The pie chart provides the overall ratios of genes in the core-, shell- and cloud-genome categories. **d**, Ratios of genes in the core-, shell- and cloud-genome categories detailed for each oat line. **e**, Gene-expression counts as Log TPM values for the core-, shell- and cloud- genes and six different tissues.

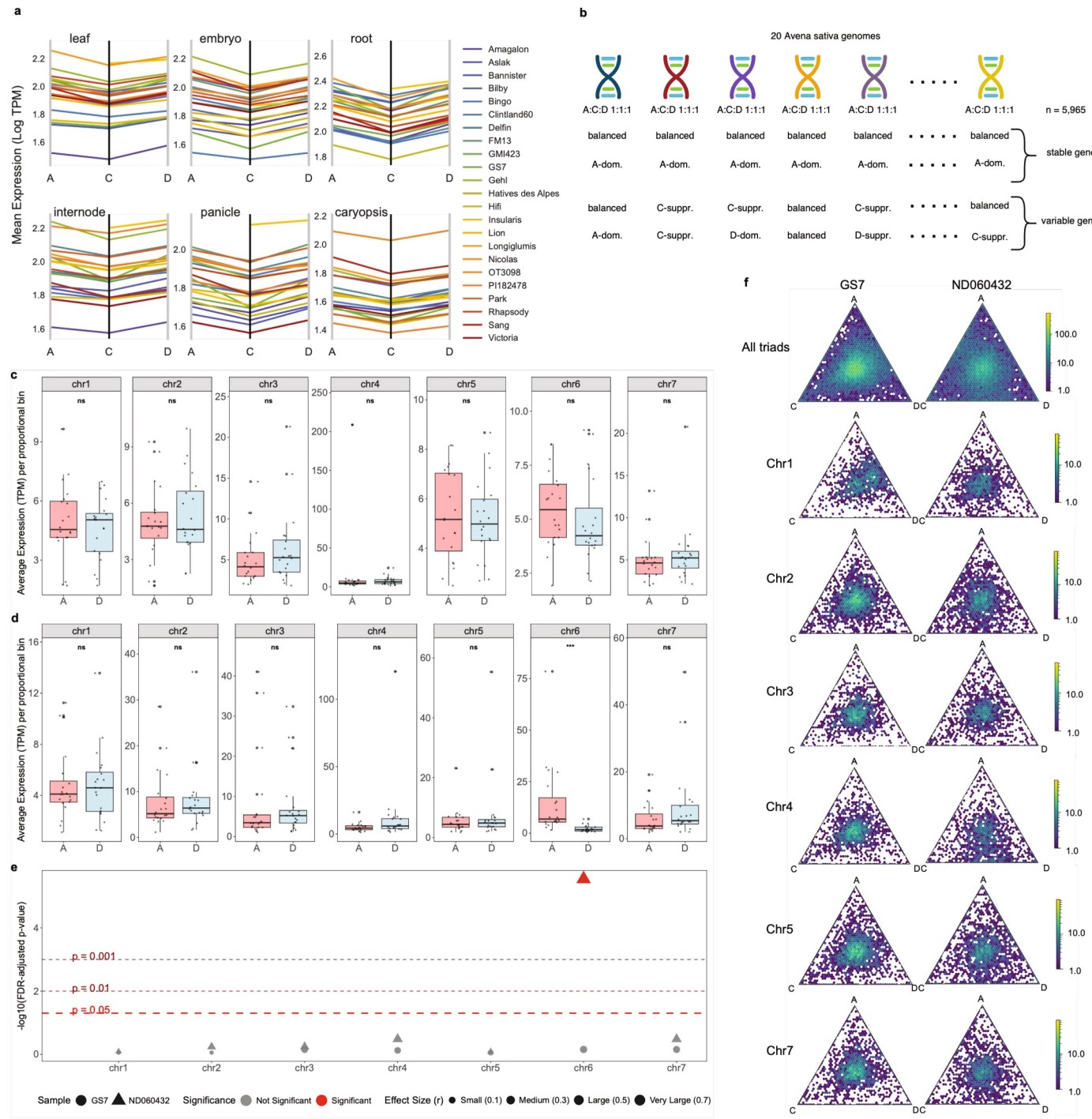

**Extended Data Fig. 4 | Expression diversity revealed by the oat pangenome.**
**a**, Parallel coordinates plot showing the mean expression of each of the A, C, and D subgenome in six tissues of 22 oat genomes, including 20 *A. sativa* genomes, synthetic Amagalon, and tetraploid BYU209 (*A. insularis*). **b**, Expression level categories were identified in 20 *A. sativa* lines using 5,294 common single-copy orthologues. Triads were classified as stable or variable based on whether all pangenome lines shared the same expression level category. **c**,**d**, Average expression levels of the A- and D-chromosomes in each chromosome group in GS7 (**c**) and ND060432 (**d**); n = 40 bins per comparison (20 bins per chromosome in each subgenome). For each chromosome, the total number of genes was divided into 20 bins, with each bin representing average expression values of all genes in that genomic window based on transcripts per million. Number of genes per chromosome: 1A: 7,121; 2A: 6,090; 3A: 4,307; 4A: 7,971; 5A: 6,230; 6A: 6,273; 7A: 5,700; 1D: 6,941; 2D: 7,048; 3D: 4,957; 4D: 7,739; 5D: 6,615; 6D: 3,484; 7D: 6,194. Box plots indicate the median (centre line), 25th and 75th percentiles (bounds of box), whiskers extending to 1.5×IQR, and outliers shown as points.

Statistical significance was assessed using two-sided Wilcoxon rank-sum tests with FDR correction for multiple comparisons across seven chromosome pairs, significance levels: ***P < 0.001, **P < 0.01, *P < 0.05, ns = not significant. **e**, Summary of statistical significance and effect sizes for A vs. D expression differences across chromosome pairs in GS7 and ND060432. Statistical significance was assessed using two-sided Wilcoxon rank-sum tests with FDR correction for multiple comparisons. The y-axis shows −log$_{10}$(FDR-adjusted P-values) of the Wilcoxon rank-sum tests. Point size corresponds to effect size (r), calculated as |Z|/√N, where Z is the standardized test statistic and N = 40 per comparison (20 bins per subgenome chromosome) is total sample size (n = 40 per comparison: 20 bins per chromosome). Exact FDR-adjusted P-values for chromosome 1 to 7: for GS7: 0.8831, 0.8831, 0.7029, 0.7509, 0.8831, 0.7029, 0.7029; for ND060432: 0.8201, 0.5789, 0.5727, 0.3306, 0.8201, 0, 0.3306. **f**, Ternary plots of subgenome expression in all triads and the triads in each of chromosome groups 1,2,3,4,5, and 7 (the plot for chr6 is displayed in Fig. 4f) in GS7 and ND060432.

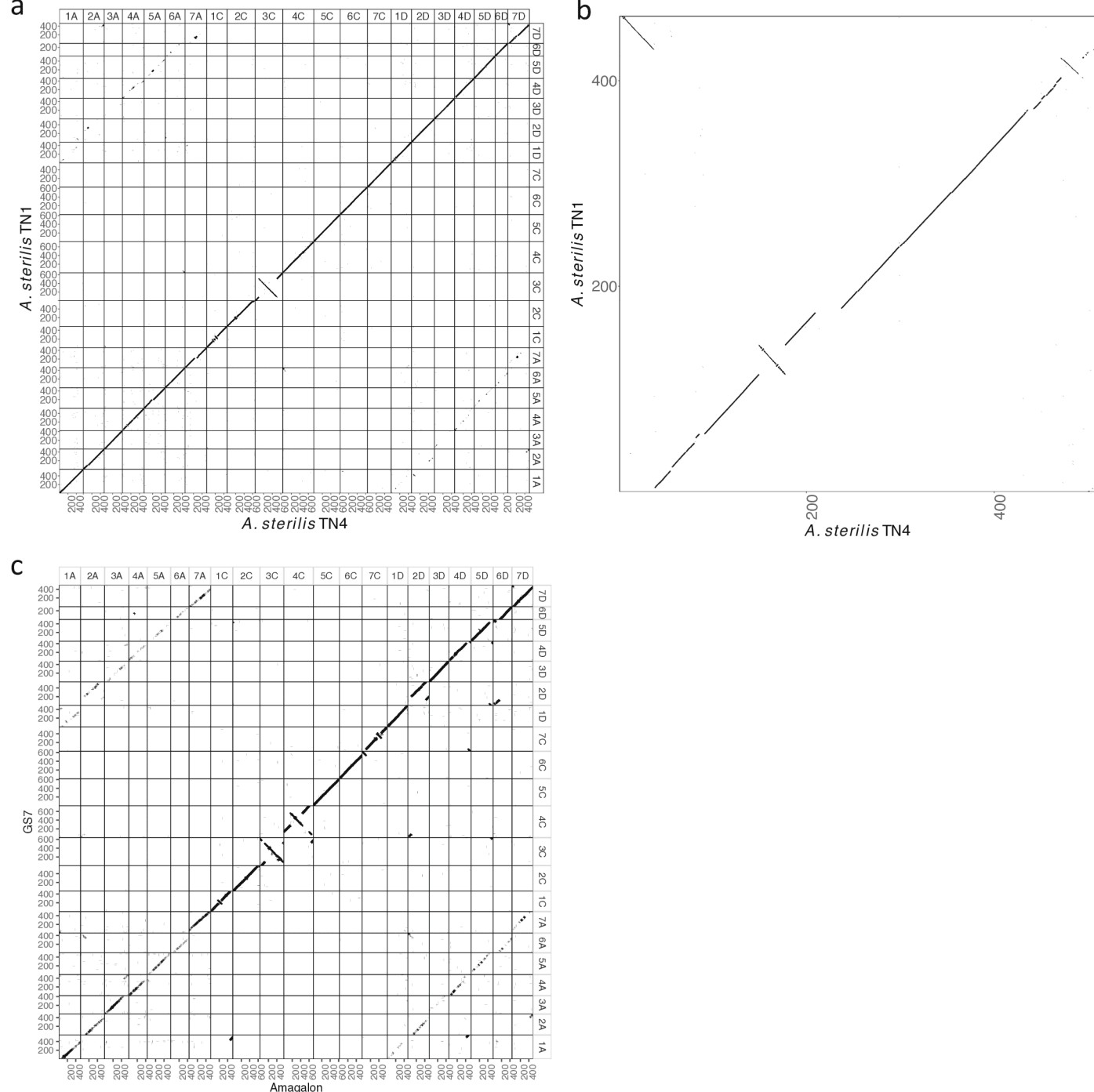

**Extended Data Fig. 5 | Genome alignment between *A. sterilis* TN1 and *A. sterilis* TN4. a**, Alignment of all chromosomes showing two large inversions on chromosomes 3 C and 7D. **b**, Alignment of chromosome 7D shows the inversion in more detail. **c**, Alignment of all chromosomes between GS7 and Amagalon.

**a**

| Reads | Mean | Median | Minimum | Maximum |
|---|---|---|---|---|
| All | 475,034,770 | 449,656,668 | 193,608,828 | 1,276,622,548 |
| Trimmed | 474,763,660 | 449,656,586 | 193,367,074 | 1,276,221,378 |
| Mapped | 465,236,133 | 437,976,338 | 189,131,003 | 1,254,042,969 |
| duplicated | 105,895,495 | 86,854,230 | 2,352,404 | 307,127,844 |
| >q10 | 256,308,499 | 258,649,374 | 126,588,032 | 688,270,256 |
| >q20 | 243,434,873 | 245,814,824 | 120,313,807 | 654,848,094 |
| >q30 | 230,530,475 | 232,100,688 | 114,146,180 | 621,257,464 |

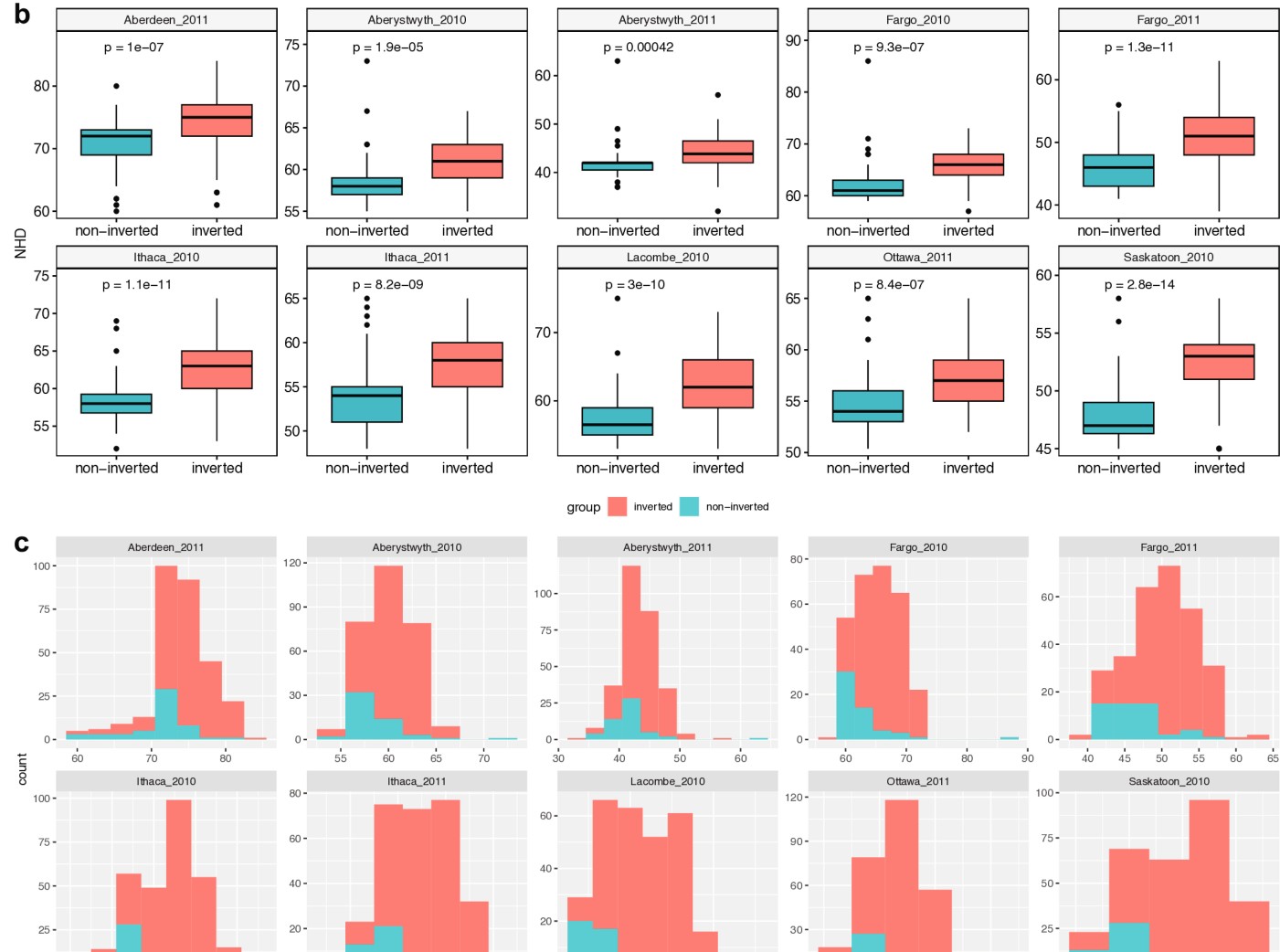

**Extended Data Fig. 6 | Distribution of heading date phenotype in the CORE panel in ten environments. a**, WGS sequencing and mapping statistics for 295 CORE genotypes. **b,c**, Box plots (**b**) and histograms (**c**) of heading date measurements (NHD) grouped by the allelic state of the inversion on chromosome 7D as inferred by PCA analysis. A two-sided t-test was used to calculate p-values. Box plots show the 25th (lower edges) to 75th (upper edges) percentiles with median lines, and whiskers extending to 1.5× the interquartile range (IQR). Outlier points are observations beyond 1.5×IQR. Sample sizes for each location/year combination are as follows: Aberdeen_2011; n = 240 (inverted), n = 53 (non-inverted), Aberystwyth_2010; n = 241 (inverted), n = 53 (non-inverted), Aberystwyth_2011; n = 240 (inverted), n = 54 (non-inverted), Fargo_2010; n = 240 (inverted), n = 53 (non-inverted), Fargo_2011; n = 240 (inverted), n = 52 (non-inverted), Ithaca_2010; n = 240 (inverted), n = 52 (non-inverted), Ithaca_2011; n = 239 (inverted), n = 53 (non-inverted), Lacombe_2010; n = 241 (inverted), n = 52 (non-inverted), Ottawa_2011; n = 235 (inverted), n = 53 (non-inverted), Saskatoon_2010; n = 239 (inverted), n = 52 (non-inverted).

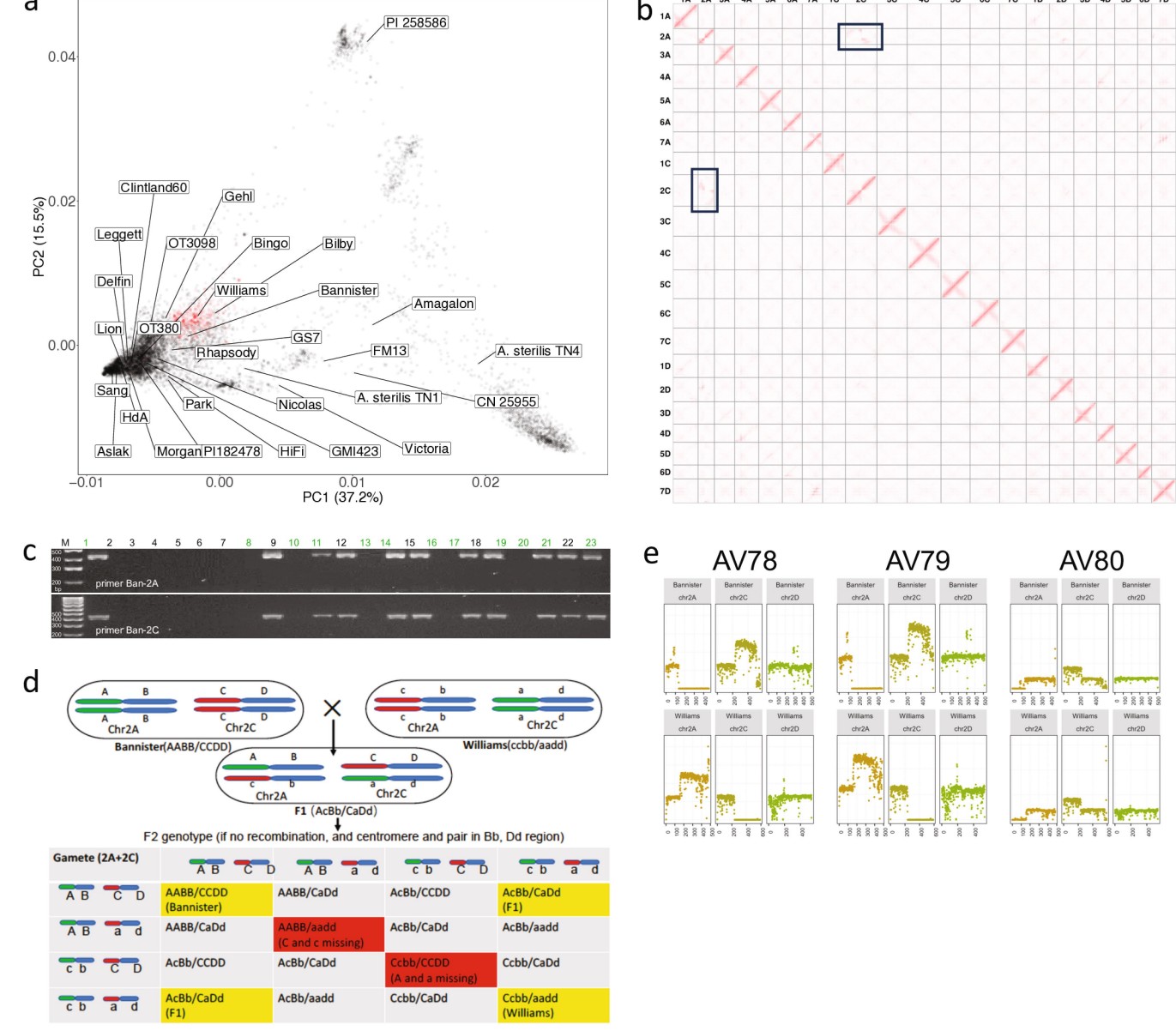

**Extended Data Fig. 7 | Legacy of mutation breeding in Australian oat. a**, PCA plot of 9,111 hexaploid oats including *A. sativa, A. byzantina* and *A. sterilis* overlaid with the positions of the 29 hexaploid PanOat lines. Australian accessions are highlighted in red. *A. byzantina* = *A. byzantina* PI258586, HdA = Hâtives des Alpes and CN25955 = *A. occidentalis* CN25955. **b**, Contact matrix of the Bannister Hi-C data when aligned to the non-translocated reference GMI423. Off-diagonal signals on chromosomes 2A and 2C are due to the reciprocal translocation between these chromosomes in Bannister. **c**, Diagnostic PCR

assay to detect both translocation breakpoints on chromosomes 2A and 2C. **d**, Types of meiosis and synapsis in $F_1$ plants and their $F_2$ progenies. **e**, Deletion and compensatory duplication of whole chromosome arms in three Bannister x Williams RILs. WGS reads were mapped to the Bannister and Williams reference genomes and reads were counted in 1 Mb bins. *In **a**, PI258586 = *A. byzantina* PI258586, HdA = Hâtives des Alpes, Nicolas = AAC Nicolas, Morgan = AC Morgan, and CN25955 = *A. occidentalis* CN25955.

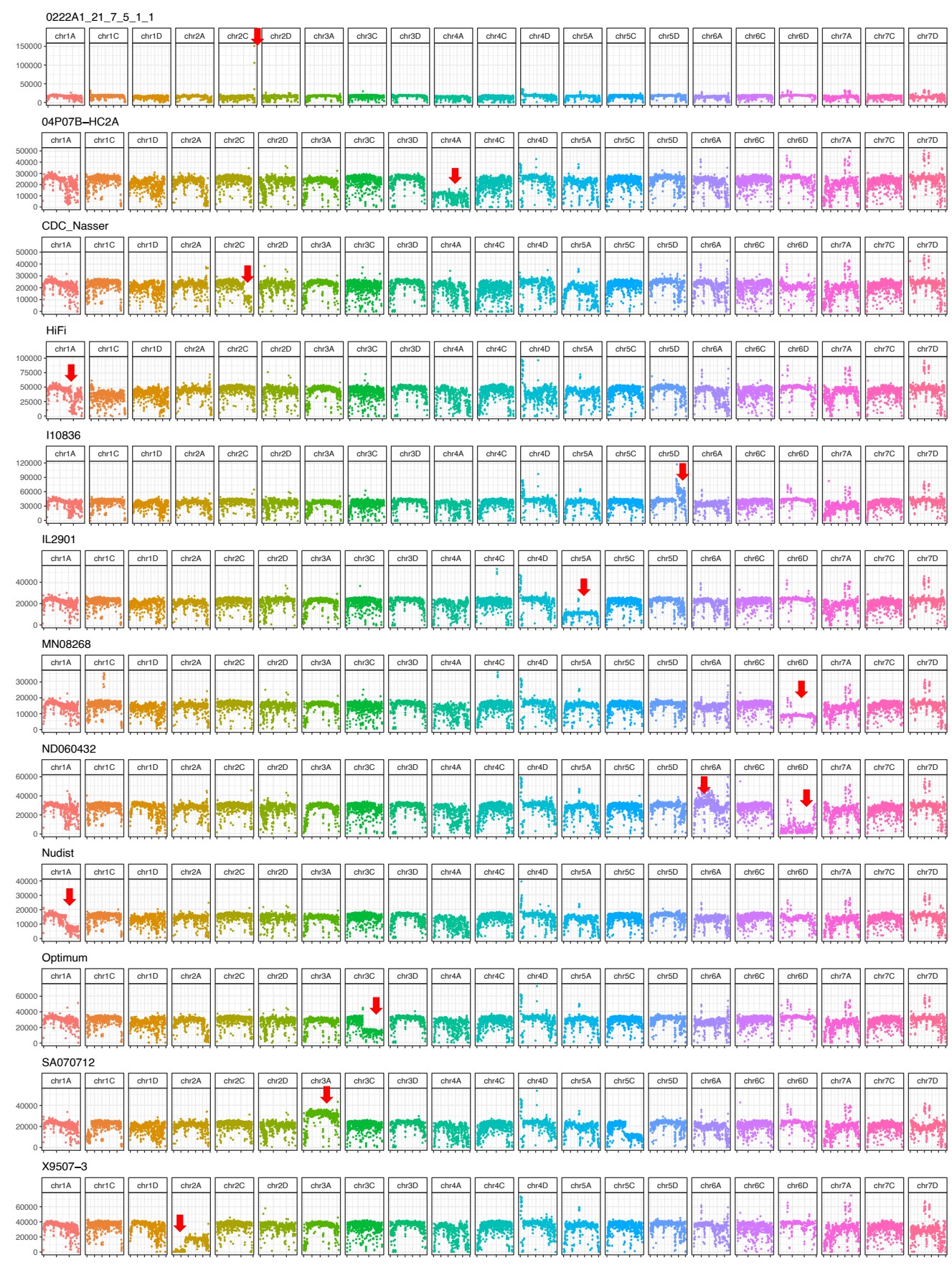

**Extended Data Fig. 8 | Chromosomal anomalies in representative CORE lines.** WGS reads were aligned to the GS7 reference genome and read counts were aggregated in 1 Mb windows. Each row shows one genotype (Supplementary Table 13). Red arrows mark SVs. At least one chromosome in each genotype is affected by large SVs, which are most likely to be deletions, duplications or homeologous exchanges. A detailed example is described in Fig. 3.

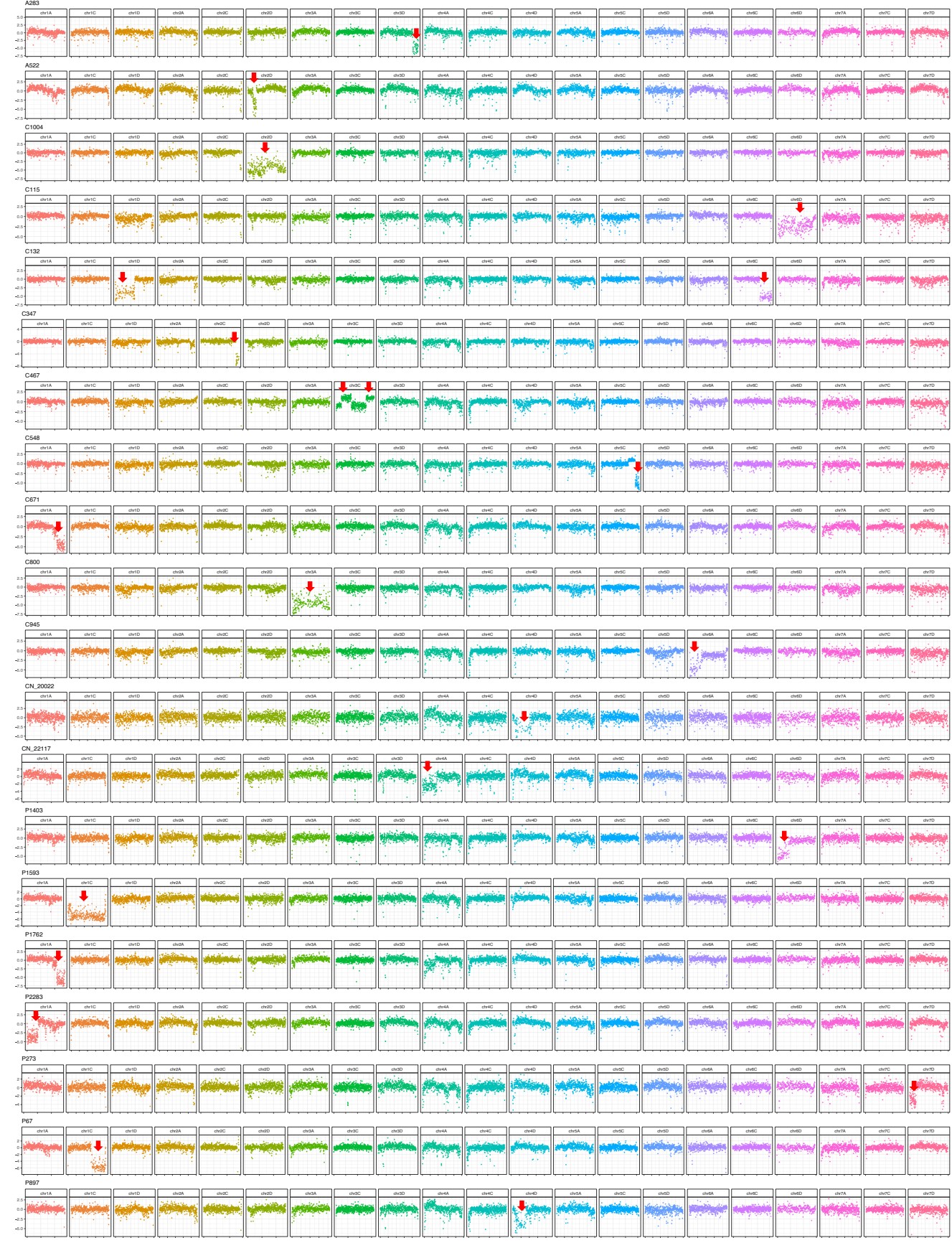

**Extended Data Fig. 9** | See next page for caption.

**Extended Data Fig. 9 | Chromosomal anomalies in representative lines from the Global Oat Diversity panel.** G.O.D. GBS data were aligned to the GS7 reference genome and normalized to GBS data from GS7; reads were counted in 1 Mb windows. Each row shows one genotype (Bekele et al.[14]). Red arrows mark SVs. At least one chromosome in each genotype is affected by large SVs, which are most likely to be deletions, duplications or homeologous exchanges as in the example elaborated on in Fig. 3. A283 is an *A. sterilis* accession from Morocco. The read depth variant on 3D is shared with A284, also an *A. sterilis* accession from Morocco.

# a

| Cross | 1A | 1C | 1D | 2A | 2C | 2D | 3A | 3C | 3D | 4A | 4C | 4D | 5A | 5C | 5D | 6A | 6C | 6D | 7A | 7C | 7D |
|---|---|---|---|---|---|---|---|---|---|---|---|---|---|---|---|---|---|---|---|---|---|
| 19S32 | 52:2:46 0.527 | 54:2:44 0.499 | 49:1:50 0.502 | 40:1:59 0.078 | 44:1:55 0.229 | 48:2:50 0.664 | 54:1:46 0.338 | 47:1:52 0.300 | 41:2:58 0.164 | 50:1:49 0.441 | 55:1:44 0.327 | 40:2:59 0.100 | 57:1:42 0.132 | 43:1:56 0.182 | 49:1:50 0.465 | 39:1:60 0.061 | 46:1:53 0.383 | 50:1:49 0.436 | 61:1:38 0.025 | 42:1:57 0.120 | 51:2:47 0.644 |
| 19S43 | 41:1:58 0.107 | 48:1:50 0.529 | 34:1:65 0.005 | 24:1:75 0.000 | 31:1:68 0.001 | 24:1:75 0.000 | 42:1:57 0.198 | 41:1:58 0.091 | 51:2:47 0.689 | 34:2:64 0.009 | 54:1:46 0.289 | 49:1:49 0.577 | 55:1:44 0.297 | 43:2:55 0.308 | 43:1:56 0.270 | 43:1:56 0.270 | 43:1:56 0.270 | 43:1:56 0.270 | 43:1:56 0.270 | 43:1:56 0.270 | 43:1:56 0.270 |
| 19S10 | 49:1:50 0.465 | 38:1:61 0.031 | 56:1:43 0.244 | 42:2:56 0.296 | 40:2:58 0.115 | 59:1:40 0.058 | 47:1:52 0.394 | 44:1:55 0.240 | 63:1:36 0.011 | 54:2:44 0.494 | 64:1:35 0.006 | 36:2:62 0.032 | 51:1:48 0.439 | 51:1:48 0.559 | 42:1:57 0.136 | 57:1:42 0.180 | 43:1:56 0.160 | 31:1:68 0.001 | 44:1:55 0.320 | 24:1:75 0.000 | 64:1:35 0.004 |
| 19S16 | 48:1:52 0.316 | 46:1:53 0.451 | 55:1:43 0.298 | 53:1:46 0.475 | 48:2:50 0.747 | 57:1:42 0.145 | 44:1:55 0.209 | 38:1:61 0.031 | 40:1:59 0.076 | 56:1:43 0.165 | 50:1:49 0.504 | 63:1:36 0.011 | 41:2:58 0.164 | 48:1:51 0.556 | 49:1:50 0.465 | 43:1:56 0.264 | 43:1:56 0.212 | 43:1:56 0.196 | 57:1:42 0.192 | 41:1:57 0.159 | 49:2:49 0.680 |
| 19S21 | 39:1:60 0.037 | 56:1:42 0.207 | 60:1:39 0.039 | 24:1:75 0.000 | 31:1:68 0.005 | 34:1:65 0.005 | 41:1:58 0.075 | 31:1:68 0.001 | 41:2:58 0.170 | 51:1:48 0.439 | 37:1:62 0.022 | 31:1:68 0.001 | 48:1:51 0.365 | 41:1:58 0.084 | 34:1:65 0.005 | 66:1:33 0.002 | 50:1:49 0.610 | 58:2:41 0.155 | 63:2:35 0.017 | 63:1:36 0.010 | 46:2:52 0.727 |
| 19S24 | 49:1:50 0.609 | 52:1:47 0.481 | 34:1:65 0.005 | 57:2:41 0.189 | 61:1:38 0.035 | 43:1:56 0.236 | 45:1:54 0.301 | 63:1:36 0.011 | 63:1:36 0.011 | 49:1:50 0.611 | 44:1:54 0.359 | 51:1:48 0.420 | 45:1:54 0.238 | 58:2:40 0.153 | 51:1:48 0.389 | 62:1:37 0.020 | 62:1:37 0.023 | 24:1:75 0.029 | 62:1:37 0.000 | 24:1:75 0.000 | 47:1:52 0.364 |
| 19S28 | 44:1:54 0.348 | 59:2:39 0.105 | 60:2:39 0.069 | 54:1:45 0.304 | 49:1:49 0.577 | 48:2:51 0.743 | 46:1:53 0.399 | 47:2:51 0.696 | 60:2:38 0.054 | 47:1:52 0.319 | 24:1:75 0.000 | 55:1:44 0.271 | 57:1:42 0.180 | 46:1:53 0.355 | 41:1:58 0.098 | 42:1:57 0.162 | 56:3:42 0.363 | 41:1:58 0.091 | 41:2:58 0.170 | 58:1:41 0.123 |  |
| 19S29 | 24:1:75 0.000 | 66:1:33 0.002 | 24:1:75 0.000 | 24:1:75 0.000 | 24:1:75 0.000 | 58:0:41 0.066 | 24:1:75 0.000 | 24:1:75 0.000 | 24:1:75 0.000 | 42:1:57 0.113 | 24:1:75 0.000 | 24:1:75 0.000 | 24:1:75 0.000 | 51:1:48 0.483 | 24:1:75 0.000 | 63:1:36 0.011 | 24:1:75 0.000 | 24:1:75 0.000 | 58:0:41 0.066 | 38:1:61 0.027 | 24:1:75 0.000 |
| 19S36 | 45:2:54 0.448 | 44:1:55 0.294 | 62:2:37 0.026 | 54:1:46 0.338 | 43:1:56 0.204 | 56:1:43 0.252 | 54:2:44 0.398 | 45:1:54 0.372 | 50:2:48 0.629 | 64:1:35 0.007 | 59:1:40 0.073 | 62:1:37 0.023 | 31:1:68 0.001 | 60:1:39 0.044 | 45:2:53 0.464 | 36:1:63 0.013 | 45:1:54 0.313 | 42:1:56 0.216 | 49:1:50 0.406 | 50:2:48 0.732 | 24:1:75 0.000 |
| 19S4 | 56:2:42 0.232 | 42:2:56 0.279 | 62:2:37 0.026 | 38:2:61 0.041 | 24:1:75 0.000 | 24:1:75 0.000 | 39:1:60 0.049 | 24:1:75 0.000 | 46:2:51 0.824 | 48:2:50 0.730 | 52:1:47 0.481 | 62:1:37 0.029 | 54:1:45 0.370 | 50:1:48 0.561 | 60:2:38 0.060 | 38:1:61 0.036 | 60:1:39 0.039 | 42:1:57 0.198 | 54:2:45 0.466 | 24:1:75 0.000 | 49:2:49 0.647 |
| 19S5 | 39:1:60 0.063 | 65:1:34 0.003 | 55:1:44 0.305 | 39:2:59 0.094 | 54:1:45 0.272 | 24:1:75 0.000 | 62:1:37 0.029 | 58:1:41 0.131 | 45:1:54 0.369 | 50:1:49 0.507 | 31:1:68 0.001 | 64:1:35 0.004 | 24:1:75 0.000 | 57:1:42 0.192 | 47:1:52 0.462 | 38:1:61 0.027 | 46:1:52 0.445 | 56:1:43 0.267 | 45:2:53 0.468 | 48:2:50 0.672 | 60:1:40 0.051 |
| 19S7 | 58:1:41 0.123 | 45:1:54 0.377 | 55:2:43 0.326 | 24:1:75 0.000 | 49:2:50 0.709 | 44:1:55 0.320 | 42:1:57 0.154 | 48:2:50 0.756 | 48:1:51 0.576 | 55:1:44 0.275 | 38:1:61 0.043 | 56:1:43 0.267 | 49:1:50 0.502 | 52:1:46 0.456 | 43:1:56 0.270 | 48:1:51 0.540 | 46:1:52 0.511 | 60:1:39 0.039 | 53:2:45 0.535 | 40:1:58 0.102 | 63:2:35 0.012 |
| 19S8 | 42:1:57 0.147 | 31:1:68 0.001 | 41:1:58 0.084 | 63:2:35 0.017 | 49:2:50 0.777 | 66:1:33 0.002 | 60:1:39 0.044 | 38:1:61 0.036 | 24:1:75 0.000 | 39:1:60 0.049 | 39:2:59 0.087 | 24:1:75 0.000 | 59:1:40 0.058 | 24:1:75 0.000 | 47:1:52 0.462 | 44:1:55 0.209 | 48:1:51 0.540 | 52:2:47 0.658 | 52:1:47 0.497 | 50:1:49 0.509 | 54:1:45 0.404 |

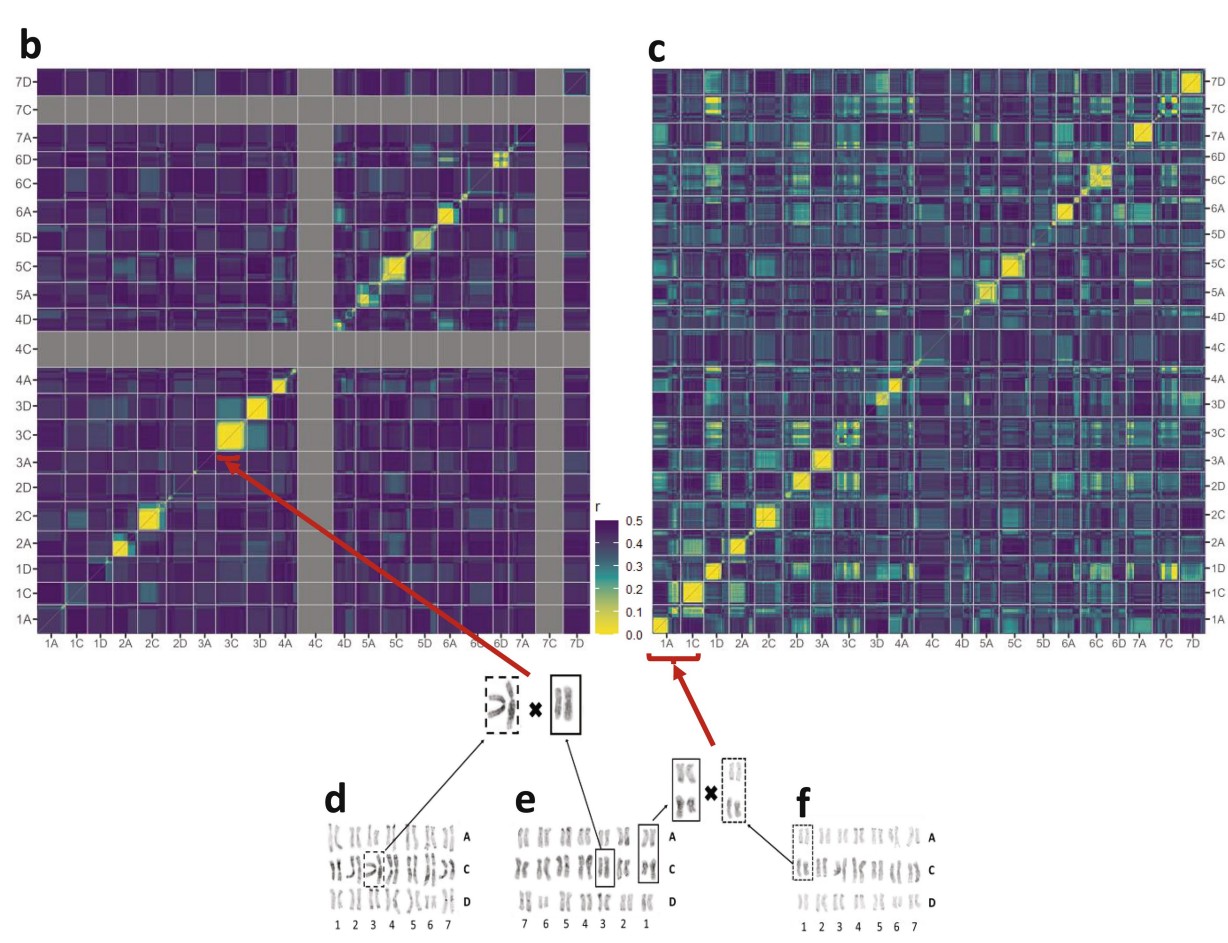

**Extended Data Fig. 10** | See next page for caption.

**Extended Data Fig. 10 | Large-scale chromosomal rearrangements shaped the segregation and recombination patterns in progenies of 13 crosses from a working oat breeding program.** (**a**) Summary of 13 crosses (rows) by chromosome (columns). The top row in each cell shows the segregation ratios (AA:AB:BB). The bottom row shows the p-value for the chi-square goodness-of-fit tests of the expected $F_6$ ratio (31:2:31). Cell colors highlight significant P values to reject the fit (pink = p < 0.01; blue = p < 0.05). Boxes around cells represent suppressed recombination within or between chromosomes. Most crosses demonstrated expected segregation ratios and recombination patterns across the majority of chromosomes. Cross 19S29 showed segregation distortions across most chromosomes. All crosses had at least one chromosome that showed distorted segregation. The recombination patterns of half-sib crosses 19S32 and 19S43, which share the common parent OA1613-5, were investigated in further detail using recombination heatmaps. (**b**) The 19S32 progenies (n = 147) exhibited some suppressed recombination, but only on chromosomes 2C and 3C. There were insufficient markers to generate accurate scaled r values on chromosomes 4C and 7C. (**c**) In contrast, the 19S43 progenies (n = 122) exhibited pseudo linkage between chromosomes 1A and 1C, as well as between chromosomes 1D and 7C. Additional chromosomes that exhibited some recombination suppression include chromosomes 2C, 3A, 3C, and 4D.

Karyotypes of the three parental lines confirmed that they have a complete set of 21 chromosome pairs. (**d**) The karyotype of OA1623-2, the female parent of 19S32, confirmed the presence of a heterozygous inversion on chromosome 3C as well as a homozygous chromosome 1A/1C translocation. (**e**) The karyotype of OA1613-5, the pollen donor for the two crosses, shows a homozygous 3C inversion (non-ancestral) and a homozygous 1A/C translocation. This is manifested in cross 19S32 by a partially suppressed recombination pattern on chromosome 3C and by the expected recombination patterns in 1A and 1C. (**f**) The karyotype of OA1568-6, the female parent of 19S43, shows a pair of ancestral 1A chromosomes (without the 1A/1C translocation), confirming the reason for pseudo-linkage between chromosomes 1A and 1C in cross 19S43. These crosses were not made intentionally for this purpose; rather, they are part of a breeding program that uses both conventional and genomic selection. Even though these populations were small and were advanced by modified single-seed descent, they still show evidence of known SVs and suggest additional SVs in oats. The companion study14, which detected chromosome inversions using a population-based approach, identified large-scale chromosomal inversions in all affected chromosomes except 7C. A pseudo-linkages similar to 1A/1C between 1D and 7C in 19S43 (**c**) suggests the possibility of an additional translocations.

Nicholas A. Tinker
Jason D. Fiedler
Chengdao Li
Peter J. Maughan
Manuel Spannagl

## Reporting Summary

## Statistics

For all statistical analyses, confirm that the following items are present in the figure legend, table legend, main text, or Methods section.

| n/a | Confirmed | |
|-----|-----------|---|
| ☐ | ☒ | The exact sample size (*n*) for each experimental group/condition, given as a discrete number and unit of measurement |
| ☐ | ☒ | A statement on whether measurements were taken from distinct samples or whether the same sample was measured repeatedly |
| ☐ | ☒ | The statistical test(s) used AND whether they are one- or two-sided *Only common tests should be described solely by name; describe more complex techniques in the Methods section.* |
| ☒ | ☐ | A description of all covariates tested |
| ☐ | ☒ | A description of any assumptions or corrections, such as tests of normality and adjustment for multiple comparisons |
| ☐ | ☒ | A full description of the statistical parameters including central tendency (e.g. means) or other basic estimates (e.g. regression coefficient) AND variation (e.g. standard deviation) or associated estimates of uncertainty (e.g. confidence intervals) |
| ☐ | ☒ | For null hypothesis testing, the test statistic (e.g. *F*, *t*, *r*) with confidence intervals, effect sizes, degrees of freedom and *P* value noted *Give P values as exact values whenever suitable.* |
| ☒ | ☐ | For Bayesian analysis, information on the choice of priors and Markov chain Monte Carlo settings |
| ☒ | ☐ | For hierarchical and complex designs, identification of the appropriate level for tests and full reporting of outcomes |
| ☐ | ☒ | Estimates of effect sizes (e.g. Cohen's *d*, Pearson's *r*), indicating how they were calculated |

*Our web collection on statistics for biologists contains articles on many of the points above.*

## Software and code

Policy information about availability of computer code

| | |
|---|---|
| Data collection | No software used for data collection. |
| Data analysis | Multiple published software packages were used in the analysis including: BBMap v37.93, Bcftools v1.12, bedtools v2.31.0, BFC v1.0, blast v2.14.0, blast v2.12, clusterProfiler v4.6, CRAM v3.1, cutadapt v3.3, Deseq2 , diamond v2.1.8, EVidenceModeler v1.1.1, Fastp v.0.24.1, GENESPACE v1.2.2, GenomeThreader v1.7.3, genometools-genometools v1.6.5, gffread v0.12.7, gmap v2018.07.04, HiRise pipeline v2.0.5, Kallisto, kmerGWAS v2, LiftoffTools v0.4.4, Mercator4 v6.0, mikado v2.3.4, Minia3 v3.2.0, minimap2 v2.26, minimap2 v2.1, Minimap2 v2.24, miniprot v0.11, MMSeqs2 v13, MSTmap, NGenomeSyn v1.41, Novosort v4.03.01, Orthofinder v2.5.5, PASApipeline v2.5.3, pfam database release 34, PLINK v1.9, portcullis v1.2.4, prodigal v2.6.3, prot-scriber v0.1.5, R statistical environment v3.5.1, samtools v1.17, samtools v1.13, samtools v1.16.1, SOAPDenovo2 v2.04-r240, star v2.7.10b, stringtie 2.2.1, TransDecoder v5.2.0, vmatch v2.3.0 |

For manuscripts utilizing custom algorithms or software that are central to the research but not yet described in published literature, software must be made available to editors and reviewers. We strongly encourage code deposition in a community repository (e.g. GitHub). See the Nature Portfolio guidelines for submitting code & software for further information.

## Data

Policy information about availability of data

  All manuscripts must include a data availability statement. This statement should provide the following information, where applicable:
- - Accession codes, unique identifiers, or web links for publicly available datasets
- - A description of any restrictions on data availability
- - For clinical datasets or third party data, please ensure that the statement adheres to our policy

The data generated by the PanOat Consortium are made freely available and publicly accessible through deposition in public databases. Sequence data were deposited in the European Nucleotide Archive under project IDs PRJEB56828 (genome assembly raw data), PRJEB57570 (transcriptome sequencing) and PRJEB62778 (WGS resequencing data). Project IDs for individual assemblies and BioSample IDs for individual CORE genotypes are listed in Supplementary Tables 17 and 13, respectively. The annotation datasets are available for download from the USDA-ARS GrainGenes database17 at https://graingenes.org//GG3/content/panoat-data-download-page. This page also serves as a landing page for access not only to data but also to genome browser tools and BLAST services.

## Research involving human participants, their data, or biological material

Policy information about studies with human participants or human data. See also policy information about sex, gender (identity/presentation), and sexual orientation and race, ethnicity and racism.

| | |
|---|---|
| Reporting on sex and gender | Not applicable. |
| Reporting on race, ethnicity, or other socially relevant groupings | Not applicable. |
| Population characteristics | Not applicable. |
| Recruitment | Not applicable. |
| Ethics oversight | Not applicable. |

Note that full information on the approval of the study protocol must also be provided in the manuscript.

# Field-specific reporting

Please select the one below that is the best fit for your research. If you are not sure, read the appropriate sections before making your selection.

☒ Life sciences   ☐ Behavioural & social sciences   ☐ Ecological, evolutionary & environmental sciences

For a reference copy of the document with all sections, see nature.com/documents/nr-reporting-summary-flat.pdf

# Life sciences study design

All studies must disclose on these points even when the disclosure is negative.

| | |
|---|---|
| Sample size | Population structure of a diversity panel (CORE) was analyzed. Representative accessions were chosen. |
| Data exclusions | No data were excluded. |
| Replication | DNA/RNA was extracted from verified clones of the same genotypes. |
| Randomization | Genome assembly and analysis were conducted on a individual Avena ssp. plant, thus randomization is not necessary. |
| Blinding | Genome assembly and analysis were conducted on a individual Avena ssp. plant, thus randomization and/or blinding is not necessary. |

# Reporting for specific materials, systems and methods

We require information from authors about some types of materials, experimental systems and methods used in many studies. Here, indicate whether each material, system or method listed is relevant to your study. If you are not sure if a list item applies to your research, read the appropriate section before selecting a response.

## Materials & experimental systems

| n/a | Involved in the study |
|-----|----------------------|
| ☒ | Antibodies |
| ☒ | Eukaryotic cell lines |
| ☒ | Palaeontology and archaeology |
| ☒ | Animals and other organisms |
| ☒ | Clinical data |
| ☒ | Dual use research of concern |
| ☐ ☒ | Plants |

## Methods

| n/a | Involved in the study |
|-----|----------------------|
| ☒ | ChIP-seq |
| ☒ | Flow cytometry |
| ☒ | MRI-based neuroimaging |

# Dual use research of concern

Policy information about <u>dual use research of concern</u>

## Hazards

Could the accidental, deliberate or reckless misuse of agents or technologies generated in the work, or the application of information presented in the manuscript, pose a threat to:

| No | Yes | |
|----|-----|---|
| ☒ | ☐ | Public health |
| ☒ | ☐ | National security |
| ☒ | ☐ | Crops and/or livestock |
| ☒ | ☐ | Ecosystems |
| ☒ | ☐ | Any other significant area |

## Experiments of concern

Does the work involve any of these experiments of concern:

| No | Yes | |
|----|-----|---|
| ☒ | ☐ | Demonstrate how to render a vaccine ineffective |
| ☒ | ☐ | Confer resistance to therapeutically useful antibiotics or antiviral agents |
| ☒ | ☐ | Enhance the virulence of a pathogen or render a nonpathogen virulent |
| ☒ | ☐ | Increase transmissibility of a pathogen |
| ☒ | ☐ | Alter the host range of a pathogen |
| ☒ | ☐ | Enable evasion of diagnostic/detection modalities |
| ☒ | ☐ | Enable the weaponization of a biological agent or toxin |
| ☒ | ☐ | Any other potentially harmful combination of experiments and agents |

# Plants

| | |
|---|---|
| Seed stocks | Seeds of the of pangenome panel and CORE population are available from German federal ex situ genebank at IPK Gatersleben.de OR USDA-ARS genebank https://www.ars-grin.gov/. |
| Novel plant genotypes | *Describe the methods by which all novel plant genotypes were produced. This includes those generated by transgenic approaches, gene editing, chemical/radiation-based mutagenesis and hybridization. For transgenic lines, describe the transformation method, the number of independent lines analyzed and the generation upon which experiments were performed. For gene-edited lines, describe the editor used, the endogenous sequence targeted for editing, the targeting guide RNA sequence (if applicable) and how the editor was applied.* |
| Authentication | *Describe any authentication procedures for each seed stock used or novel genotype generated. Describe any experiments used to assess the effect of a mutation and, where applicable, how potential secondary effects (e.g. second site T-DNA insertions, mosiacism, off-target gene editing) were examined.* |

