## [Peer Review File · Nature]

A pangenome and pantranscriptome of hexaploid oat

Corresponding Author: Dr Martin Mascher

Version 0:

Reviewer comments:

Referee #1

(Remarks to the Author)

The manuscript "A pangenome and pantranscriptome of hexaploid oat" describes chromosome-level assemblies of 33 wild and domesticated oat accessions, along with an atlas of gene expression for six tissues in 23 of these accessions. This reveals presence-absence variation present in the pangenome, patterns of gene expression across different tissues and subgenomes, genetic diversity and gene expression in the cellulose synthase family of genes, and chromosomal rearrangements present in oat germplasm, which may have implications for breeding traits of interest in oat.

The manuscript represents the first pangenome in oat, which will be extremely valuable to the oat research and breeding communities, as it provides valuable new genomic resources and insight into oat evolution and adaptation.

Overall, the data and methodology provided are very good, although I have a few suggestions for improvements/questions in places, as follows:

Main Body

Line 119 - The start of the panel composition list is rather vague, and could include more specific information (ie. how many "elite" lines; what do you mean by "plant genetic resources with interesting properties" - are these the landraces?; has *A. occidentalis* been left out of the list?)

Line 124 - Perhaps I am misinterpreting the figure as it stands (see later comment), but do the PanOat lines cover the entire genetic diversity of oat as suggested in the text? It seem that there is at least one cluster of samples on the PCA (eg. under the label for Bannister) which is not represented?

Line 248-274 - As the rationale for exploring diversity in the cellulose synthase superfamily is its role in beta-glucan synthesis, I feel like the link between the results and beta-glucan could be strengthened. This could perhaps be done by comparing the expression data to whatever is known about the beta-glucan phenotype of these lines to look for patterns in copy number/gene expression in low/high beta-glucan lines. Alternatively, if there is insufficient information about beta-glucan levels in this collection of lines, the authors could make more links to other studies. For example, higher expression of the C homeologue of the *CsIF6* gene has previously been linked to lower beta-glucan levels in oat (10.1002/csc2.20015) - is this consistent with your data? Also in Line 262, what do you mean by "most important families"? *CsIH* is also important in beta-glucan synthesis in cereals (10.1073/pnas.0902019106), but is not mentioned much here.

Line 289 - Does the data in Fig 3a suggest several inversions on chromosome 7D (rather than a single inversion)? If so, this text (and that in the Fig 3a legend) should be updated.

Line 304 - Is the position of the GWAS peak consistent with the inversion position (or likely breakpoints) on 7D? This is not really clear from the text or Figure 3c.

Figures

- A number of figure panels are not mentioned in the text (Fig 1d, 2d, 2e). This should be amended in the text, or alternatively

these figure panels removed or moved to supplementary materials if they are not required.

- I find Figure 1a quite difficult to navigate. I suggest editing the figure to prevent crossing lines, and ensure all accessions have a line pointing to their location on the plot (A. sterilis does not appear to have one). The authors may also consider outlining boxes for those lines which are also included in Figure 1c in the same colour-coding (ie. Bannister in pink, HiFi in red etc.) to provide a link between these panels.

- For me, Figure 1b is also quite hard to read. Use of more contrasting colours and different colour-coding for missing data and the inversion position would make this figure clearer.

Referee #2

(Remarks to the Author)

This large pangenome and pantranscriptome consortium has delivered a much needed resource for oat breeding and research which is also of wide general interest as this important crop species is a polyploid with a complex but very interesting evolutionary history. Now, thanks this work we have 33 annotated sequence assemblies and transcript data from 23 of these accessions from six tissues. Without any analysis of any kind this a formidable and exciting resource which should be publicised widely via high profile publication in order to help realise the huge potential of the resource.

However, the manuscript attempts to do more than this and attempts to tell multiple stories of direct biological interest. Following the description of the genomic and transcriptomic results and signposting readers towards these datasets the authors use a set of case studies to exemplify the utilisation of the pan resources.

These are:

1. Analysis of the oat cellulose synthase gene family
2. 7D inversion associated with flowering time
3. 2A/2C translocation (associated with yield increase) and Dw6 semi dwarfing gene
4. 6Ds-6Ai substitution

Topic 1 (cellulose synthase) tries to shed light on B glucan content in oats which is a very important topic. But the authors do not show an association of any of these transcripts with B glucan content in oat. Nor do they show why B glucan levels are high in oat and barley but low in related species like wheat. As far as I can see, these results showed that increased copy number for some of the genes in this pathway was correlated with their expression level. I do not think that there is any novelty in that observation.

I suggest that the Cellulose synthase section is removed.

The structural variants found within oat are extremely interesting and topics 2-4 cover this. Each is an interesting story in its own right. However, the ms read as if each set of authors conducting the three areas of research had written a section for the manuscript and they had been added as modules without very much effort to unify commonalities between these stories.

For example:

Line 284 1C/1A

Line 286 6C/1D

Line 288 1A/1C

Line 289 2A/2C

Line 399 refers to 5 regions of inversion or translocation (and nicely links to Bekele et al)

Much better to describe these all in one place and link to one table or figure showing them in the wider genomic context.

The writing needs to be more integrated so that in one section we get an overview of the big picture for SVs and then clear highlighting of the SVs that the authors want to expand on.

The ms could then be improved even further by more closely and systematically linking the SV case studies to the pan genome. At the moment the gulf between the two major elements of the ms (genomics and cases studies) are just too wide. For example the diversity in gene expression section (beginning line 182) gives an interesting but very high level view on the expression dynamics of homoeologous transcripts. But the SV case studies do not seem to make explicit linkages with the transcriptome datasets and trends in expression balance. It was honestly very surprising to see this. The authors distantly speculate on what the molecular implications of translocations and inversions are with fascinating concepts such as compensation alluded to. But what is compensation? If it includes the restoration of required transcript levels to the euploid state then this team has the dataset to show that. Similarly for candidate genes and polymorphisms. If Dw6 is a fatty acid hydroxylase then the genomic variants variants/local haplotypes for that gene should be strongly associated with height. The same applies for CO or FT1 in the case of earliness on 7D.

The SV case studies are insufficiently connected to pangenome data and analysis and this should be rectified.

Minor:

Line 182 (title) Should “thin” be “in”?

Supplementary tables (eg 11 and 12) there is hardly any description of the file contents. For me this made the tables impenetrable.

Line 345 The association with yield could be due to pedigree structures. Is there any Near Isogenic Line data that could add support?

Abstract, Introduction

There were quite a few vague and unconnected statements that do not really relate to the substance of the results eg

Plant based milk (none of the paper is about this)

Genomic research in oat is still at an early stage (does this matter, I thought it was a waste of words in the abstract)

We describe the interplay of gene expression (again this just seems vague better to make concrete statements about how the work will help researchers to understand gene expression in oat)

Fig 1

The PCA plot is not well described. What genotypic data is it based on? I could not see this in methods. I assume that this is the GBS described in the companion paper? Please say so.

Fig 2

a5 says “panicle” but shows a spikelet

a6 shows what looks like a mature seed (brown) but the sample taken was an immature caryopsis

a1 says embryo but picture is a whole seedling?

f shades of colour are almost impossible to discriminate on my printout

Fig 3 a at least six rearrangements are shown in the dot. A 450 Mb and 50 Mb inversion are described in the legend. It would be helpful to specifically label them in the figure. The units on the axes are not labelled.

c In this case the units are labelled (Mb) but there are no values shown.

Referee #3

(Remarks to the Author)

The manuscript presents a pangenome of hexaploid oat. The pangenome is composed of 33 lines. Assemblies and annotations are generated using state of the art approaches. Contig level assemblies are generated using HiFi reads and then put into chromosome-level scaffold using Hi-C data. The manuscript roughly follows the outline of two recent barley pangenome studies (<https://www.nature.com/articles/s41586-020-2947-8>, <https://doi.org/10.1101/2024.02.14.580266>) including:

(1) Germplasm genotyping

(2) Identification of minimum diversity set

(3) Generation of corresponding assemblies

(4) Assessment of the low copy pangenome

(5) Identification of structural variants including inversions

(6) Linking of large scale variation to hidden legacy of mutation breeding

Overall, I don't have any major technical concerns regarding these analyses. The authors have used similar methods in other species before and they have been proven to work.

The second and perhaps more intriguing aspect of the manuscript is exploration of gene expression patterns among the three subgenomes across multiple accessions and tissues. However, the results rely on gene expression quantification using Kallisto. Gene expression variation among homeologous genes in polyploids has been an active area of research and specialist quantification tools (for example EAGLE-RC) have been developed. Kallisto has in turn been shown to suffer from high error rate in sub-genome read assignment (<https://doi.org/10.1093/bib/bby121>, <https://doi.org/10.1093/bib/bbaa035>). I think that at the very least the transcriptome analysis should be repeated with a more appropriate quantification method to verify the conclusions based on Kallisto results.

Considering the focus on transcription diversity it would also be good if some epigenomic (for example DNA methylation) data were presented to disentangle contribution of genomic and epigenomic variation to gene expression variation. I am not entirely convinced that the correlation between genomic variation and gene expression conclusively shows that ‘genetic factors underlie the expression patterns we observed’. This should be at least discussed in more detail. How good is the agreement? Are there exceptions? Also, from technical perspective it was shown that the more similar the gene sequence the more similar Kallisto quantification results will be (<https://doi.org/10.1093/bib/bbaa035>), where sub-genome ‘read partitioning errors lead to gains in correlation’.

Version 1:

Reviewer comments:

Referee #1

(Remarks to the Author)

I am satisfied that all of my comments and suggestions have been addressed, and I have no further suggestions for improvement.

Referee #2

(Remarks to the Author)

I appreciate the authors efforts to accommodate my suggestions. The beta glucan section has been removed and the “map of structural variation” section added. However, the original problem still exists and I apologise if I did not state this strongly enough. In the first review I said: the gulf between the two major elements of the ms (genomics and cases studies) are just too wide. I am sorry to say that I still find this to be the case. I could write a very long review now with many areas that I simply found insufficient. They are all initially intriguing scientific stories but they still seem only half finished experimentally and (still) almost completely unconnected to the oat pangenome resources developed here. I would implore the authors to take just one of these stories and make a proper job of it. Currently, I do not think that either the specialist or general reader would derive anything useful after line 243. To put it another way I would be highly critical and request major changes for each of the trait sub stories if they had been submitted to a specialist journal in the area.

Here are one or two examples for each:

A chromosomal inversion linked to early heading-

1. No flowering data is included in supplementary information or (as far as I can see) in linked data repositories. There is no meta data apart from institutional source of data an year, no sites, replication, which accessions carry the inversion.
2. I asked if the authors could show putative functional variants for the candidates they propose (CO etc). In response “Owing to long haplotype blocks associated with pericentric inversion on 7D, we cannot establish a strong link between sequence variants in these genes and phenotypic variation.” This was disappointing. One of the main uses eg a breeder might make of the oat pangenome would be to identify potential functional variants for candidate genes. If these genes are being proposed as candidates then the author is laying down a hypothesis that there is a functional variant contained within them that should be very strongly associated with flowering (regardless of LD around it). The response gave the impression that the authors are not clear about this (??) or they could not be bothered to look.
3. Although the authors went to the trouble of adding the map of structural variation section they have not linked this to the following sections which drill down on the actual structural variants. Eg there is a lack of linking sentence from the “420 Mb pericentric inversion” On line 260 to the whole section on this subject that you begin to wonder if they are really the same thing. This comes up again for the following sections and actually shows that this section has been inadequately written.
4. As far as I can see GWAS is inadequately described, maybe not at all. The authors say kmer GWAS was used. There is no description of the data quality (eg QQ plots) or the methods used to account for genetic structure. Moreover one of the main practical applications of the pan genome data will be conventional GWAS using the variant files derived from read mapping to the amazing new reference sequences developed here. Again, the resources described are not being used for the biological exemplars.

The hidden legacy of mutation breeding

1. Again the phenotypic data is not provided although it is at least mentioned in the Methods under “Yield evaluation” (I think). This is really important for as panel 4c is the reason that that this is an interesting question. The data is from the 2017-2022 Australian National Variety Tests. Each year included 19 to 31 trials across Australia, with a total of 158 trials designed in three replications. There were six 2A/2C translocated varieties and eleven non-translocated varieties. This means that some method of correction for year and site effect has been applied. This is not described in Methods but is crucial as it's correct implementation underpins the validity of this data. A p value is provided for yield ($p=0.0351$). How was this arrived at? Again, as far as I can see there are no statistics in Methods. It is not stated in the Fig 4 legend that $n=17$. For me this is the most important point. Yield is a complex, highly polygenic, with low heritability, high GxE. It is not statistically valid to associate a yield effect to a single locus based on 17 lines. If it was this simple why do we all bother with GWAS panels on a minimum $n=200$?

Chromosomal rearrangements in oat breeding germplasm

For me this section really summarises the problem with this manuscript. Looking through a set of very dry supplementary figures I was delighted to see S fig13. It showed a truly fascinating and highly dynamic landscape of major structural variants in oat. This is why do pan genomics! But these results are buried. Not mentioned in the map of structural variation section at all. No attempt is made to describe the transcriptional consequences of these major variants (deletions? Maybe introgressions?). Amagolon is not even shown in Fig1 although it carries the 4C inversion mentioned in text while the other chosen accession in fig 1 do not.

In summary I find that the members of the team responsible for responding to my suggested changes did not really embrace the spirit of the suggested changes. As I show above with several examples, the map of structural variation section stands alone and is not at all integrated with the following sections. In these sections there are several basic experimental problems that lead the reader to feel doubtful about the interpretation of results. Moreover, the team seem to lack some curiosity in terms of other interesting results. For example, because they were asked, they described the transcript levels at translocation junctions the results sound really interesting:

“We investigated gene expression changes at translocation breakpoints by performing 266 differential gene expression (DGE) analysis between translocated and non-translocated lines. 267 This revealed significant enrichment of differentially expressed genes near breakpoints on 268 chromosomes 1A, 1C, and 7D, whereas the reciprocal translocation between 2A and 2C 269 showed significantly fewer DEGs.”

But the next step chosen was based on annotation of functional gene classes (enrichment for carbohydrate metabolism.....). This seems highly speculative. It would be much more interesting eg to see a figure change with DEGs plotted along the

chromosome to graphically demonstrate the very interesting observation that is currently only given a few lines of text and a basic table in S12a.

Referee #3

(Remarks to the Author)

Overall, I am satisfied with the authors responses and the additional analyses performed. I have no further major concerns. I would suggest including the comparative analysis of different gene expression quantification methods found in the response letter (especially Figs 3 and 4) in the supplementary material to make it available to the interested readers. While overall trends remain the same across methods there are some differences, which could assist interpretation of the results. Also, in my opinion showing agreement across methods strengthens the message of the manuscript.

Version 2:

Reviewer comments:

Referee #2

(Remarks to the Author)

The authors have done a thorough job of bringing to life the functional integration of structural variants. This is achieved through added depth on ND060432 with 6D rearrangements from Amagalon. They have really shown how genome assemblies can be combined with re-sequencing and RNA-seq data to link structural variants with changes in transcriptome. The reanalysis of sequence variation and expression of FT1 demonstrates the usefulness of the genomic resources for candidate gene analysis. Chromosome-scale DEG plots show the beautiful expression changes at translocation breakpoints and shine a light for the general reader on the probable consequences of large-scale structural variants. I agree that the narrative improved and the gap between genomics and biology sufficiently well filled to demonstrate the utility of these important resources.

Small comments and corrections:

317-315

Three 318 important quantitative trait locus (QTLs) are mapped to chromosome 7D, involving flowering time (CO and VRN3/FT)30 and daylight insensitivity (Di)33 319 .

This should be "daylength" rather than "daylight".

324-325

an 18 bp deletion in FT1-7D specific to 325 inverted lines

This needs a little more detail. As far as I can see this N terminus coding in frame. Likely functional or not?

326-328

These results align with findings by Mehtab-Singh et al.^{32 327} , who reported delayed flowering in triple CRISPR/Cas mutants of FT1, underscoring 328 its likely role in heading date variation.

The role of FT1 in the control of flowering is already more than clear and I don't think that this sentence or reference is helpful.

Supplementary Fig 6. Multiple sequence alignment of FT1-7D CDS from the pangenome assemblies showing a 18bp deletion.

"deletion"

Supplementary Fig 5. Population structure among Australian oats. Australian oat varieties (n=169), including the 17 varieties with known presence or absence of the translocation, have been re-sequenced with whole-genome shotgun technology at 20x sequence depth, their reads were mapped to reference genome OT3098 and SNPs profiles were then derived (detailed methodology and results are to be published elsewhere). The population structure was examined with principal component analysis.

"the translocation" Presumably this is 7D translocation? This should be made clear.

391-393

fatty acid hydroxylase (Supplementary Table 15.8). Further functional studies are required 392 to test if this gene is casual to the Dw6 phenotype

"casual" to "causal"?

427-439 Changes are good

Sup table 14 column D, specify which translocation (2A/2C)

1424

Trails with average grain yield below 1000 kg/ha 1425 were deemed as abnormal and removed from further analysis "trails" to "trials"

1392 – 1440
Very helpful

Referee #1 (Remarks to the Author):

The manuscript "A pangenome and pantranscriptome of hexaploid oat" describes chromosome-level assemblies of 33 wild and domesticated oat accessions, along with an atlas of gene expression for six tissues in 23 of these accessions. This reveals presence-absence variation present in the pangenome, patterns of gene expression across different tissues and subgenomes, genetic diversity and gene expression in the cellulose synthase family of genes, and chromosomal rearrangements present in oat germplasm, which may have implications for breeding traits of interest in oat.

The manuscript represents the first pangenome in oat, which will be extremely valuable to the oat research and breeding communities, as it provides valuable new genomic resources and insight into oat evolution and adaptation.

Overall, the data and methodology provided are very good, although I have a few suggestions for improvements/questions in places, as follows:

Answer: We thank the reviewer for their encouraging assessment of our manuscript.

Main Body

Line 119 - The start of the panel composition list is rather vague, and could include more specific information (ie. how many "elite" lines; what do you mean by "plant genetic resources with interesting properties" - are these the landraces?; has *A. occidentalis* been left out of the list?)

Answer: The panel composition is provided in Supplementary Table 1, which describes the improvement status (cultivar, breeding line, landrace or wild) as well as key characteristics of each line. *A. occidentalis* was added to the description list.

Line 124 - Perhaps I am misinterpreting the figure as it stands (see later comment), but do the PanOat lines cover the entire genetic diversity of oat as suggested in the text? It seems that there is at least one cluster of samples on the PCA (eg. under the label for Bannister) which is not represented?

Answer: Our goal was for the PanOat panel to represent as much diversity of cultivated and wild oat as possible. Yet some of the diversity is still missing as described more extensively in our companion paper (Bekele et al., <https://doi.org/10.21203/rs.3.rs-5726397/v1>, Supplementary Discussion S3). This was because the number of *A. sterilis* lines examined in the companion paper was increased after the selection of PanOat lines was already committed. The primary omission was a representative from one of the four characterized *A. sterilis* populations. To clarify this point, we added 'most of' to the sentence in line 124:

'The PanOat lines cover most of the genetic diversity space'.

Furthermore, in the discussion we state that a genus-wide pangenome, which would aim for better representation of wild taxa, should be the next step in oat genomics.

Line 248-274 - As the rationale for exploring diversity in the cellulose synthase superfamily is its role in beta-glucan synthesis, I feel like the link between the results and beta-glucan could be strengthened. This could perhaps be done by comparing the expression data to whatever is known about the beta-glucan phenotype of these lines to look for patterns in copy number/gene expression in low/high beta-glucan lines. Alternatively, if there is insufficient information about beta-glucan levels in this collection of lines, the authors could make more links to other studies. For example, higher expression of the C homeologue of the CslF6 gene has previously been linked to lower beta-glucan levels in oat (10.1002/csc2.20015) - is this consistent with your data? Also in Line 262, what do you mean by "most important families"? CslH is also important in beta-glucan synthesis in cereals (10.1073/pnas.0902019106), but is not mentioned much here.

Answer: We agree with the reviewer that the link between beta-glucans, and structural diversity, and gene expression is preliminary. Further experiments are required to measure beta-glucan levels in a wider panel of accessions, either under controlled conditions with maternal plants, either grown under uniform conditions to minimize environmental effect, or in replicated field trials to properly account for environmental effect. We believe these experiments go beyond the scope of the current work. Following the suggestion of Reviewer #3, we removed this section.

Line 289 - Does the data in Fig 3a suggest several inversions on chromosome 7D (rather than a single inversion)? If so, this text (and that in the Fig 3a legend) should be updated.

Answer: These small inversions are unique to the reference we used, GS7. See the plot below, which shows that, compared to GS7, some PanOat lines do not have some or all of these inversions. We added an explanatory sentence to the legend of Fig. 3: "The three small inversions at ~120Mb, ~200Mb and ~240Mb are unique to the reference genome GS7. "

Figure 1. Chromosome 7D alignments. All PanOat lines with chromosome 7D were aligned to the GS7 line. The x-axis is the GS7 reference and the y-axis the Panoat accessions with positions in 1Mb scale. The plot shows that the small inversions between 100Mb to 300Mb are specific to some lines.

Line 304 - Is the position of the GWAS peak consistent with the inversion position (or likely breakpoints) on 7D? This is not really clear from the text or Figure 3c.

Answer: The GWAS peak is located on the small non-inverted part of chromosome 7D. Our analysis shows that non-inverted lines with an early flowering phenotype share a common haplotype, which is likely due to reduced recombination caused by the inversion (Extended Data Figure 10 and Tinker et al. 2022 [PMID: 35585176]).

Figures

- A number of figure panels are not mentioned in the text (Fig 1d, 2d, 2e). This should be amended in the text, or alternatively these figure panels removed or moved to supplementary materials if they are not required.

Answer: Thank you for pointing this out. We have now ensured that all figure panels, including Fig. 2d and 2e, are referenced in the text. Note that there is no Fig. 1d. The text has been amended accordingly.

- I find Figure 1a quite difficult to navigate. I suggest editing the figure to prevent crossing lines, and ensure all accessions have a line pointing to their location on the plot (A. sterilis does not appear to have one). The authors may also consider outlining boxes for those lines which are also included in Figure 1c in the same colour-coding (ie. Bannister in pink, HiFi in red etc.) to provide a link between these panels.

Answer: We updated Fig. 1a to make it clearer and connect it to the oat global diversity companion paper (Bekele et al. 2025). See also Supplementary Discussion S3 of that paper. We replaced the pangenome accession names with numbers and added colors to match the population assignments of Bekele et al. 2025.

- For me, Figure 1b is also quite hard to read. Use of more contrasting colours and different colour-coding for missing data and the inversion position would make this figure clearer.

Answer: We assume the reviewer meant Fig. 1c. We changed the colour of the connecting lines between genomes to a lighter shade of grey to make the translocations and inversions stand out more clearly.

Referee #2 (Remarks to the Author):

This large pangenome and pantranscriptome consortium has delivered a much needed resource for oat breeding and research which is also of wide general interest as this important crop species is a polyploid with a complex but very interesting evolutionary history. Now, thanks this work we have 33 annotated sequence assemblies and transcript data from 23 of these accessions from six tissues. Without any analysis of any kind this a formidable and exciting resource which should be publicised widely via high profile publication in order to help realise the huge potential of the resource.

However, the manuscript attempts to do more than this and attempts to tell multiple stories of direct biological interest. Following the description of the genomic and transcriptomic results and signposting readers towards these datasets the authors use a set of case studies to exemplify the utilisation of the pan resources.

Answer: We thank the reviewer for their encouraging comments.

These are:

1. Analysis of the oat cellulose synthase gene family
2. 7D inversion associated with flowering time
3. 2A/2C translocation (associated with yield increase) and Dw6 semi dwarfing gene

4. 6Ds-6Ai substitution

Topic 1 (cellulose synthase) tries to shed light on B glucan content in oats which is a very important topic. But the authors do not show an association of any of these transcripts with B glucan content in oat. Nor do they show why B glucan levels are high in oat and barley but low in related species like wheat. As far as I can see, these results showed that increased copy number for some of the genes in this pathway was correlated with their expression level. I do not think that there is any novelty in that observation.

I suggest that the Cellulose synthase section is removed.

Answer: We appreciate the reviewer's thoughtful feedback on the cellulose synthase section. As noted, cellulose synthase genes are not the sole contributors to cell wall biosynthesis and \$\beta\$ -glucan content. The elevated \$\beta\$ -glucan levels observed in barley and oats compared to related species like wheat remain a complex and open question, influenced by multiple factors. In light of the reviewer's suggestion and recognizing that our results do not demonstrate a direct association between specific transcripts and \$\beta\$ -glucan content, we have removed this section from the manuscript. We agree that this is a challenging and highly interesting topic. Future studies incorporating multi-omics approaches and co-expression network analyses will be valuable for unraveling the regulation of \$\beta\$ -glucan biosynthesis in oats.

The structural variants found within oat are extremely interesting and topics 2-4 cover this. Each is an interesting story in its own right. However, the ms read as if each set of authors conducting the three areas of research had written a section for the manuscript and they had been added as modules without very much effort to unify commonalities between these stories.

For example:

Line 284 1C/1A

Line 286 6C/1D

Line 288 1A/1C

Line 289 2A/2C

Line 399 refers to 5 regions of inversion or translocation (and nicely links to Bekele et al)

Much better to describe these all in one place and link to one table or figure showing them in the wider genomic context.

The writing needs to be more integrated so that in one section we get an overview of the big picture for SVs and then clear highlighting of the SVs that the authors want to expand on.

Answer: We reorganized the manuscript to have a separate section entitled "A map of structural variation". This section also includes an analysis of gene expression changes in the vicinity of structural variants (see our answer to the following comment).

The ms could then be improved even further by more closely and systematically linking the SV case studies to the pan genome. At the moment the gulf between the two major elements of the ms (genomics and cases studies) are just too wide. For example the

diversity in gene expression section (beginning line 182) gives an interesting but very high level view on the expression dynamics of homoelogenous transcripts. But the SV case studies do not seem to make explicit linkages with the transcriptome datasets and trends in expression balance. It was honestly very surprising to see this. The authors distantly speculate on what the molecular implications of translocations and inversions are with fascinating concepts such as compensation alluded to. But what is compensation? If it includes the restoration of required transcript levels to the euploid state then this team has the dataset to show that. Similarly for candidate genes and polymorphisms. If Dw6 is a fatty acid hydroxylase then the genomic variants variants/local haplotypes for that gene should be strongly associated with height. The same applies for CO or FT1 in the case of earliness on 7D.

The SV case studies are insufficiently connected to pangenome data and analysis and this should be rectified.

Answer: We appreciate the suggestion to more explicitly connect expression dynamics to structural variants (SVs), and this was indeed something we considered during our analysis. However, we did not initially identify a strong, cohesive narrative or case story, which is not unexpected given that the translocations and inversions are large, and global expression patterns often remain stable due to the influence of heterochromatin, transposons, and other genomic features. To better connect our gene expression analysis with the SVs we discovered, we investigated whether expression changes occur around translocation breakpoints and conducted a detailed analysis to address this question. Based on these findings, we have now incorporated the description and results of this analysis into the manuscript (line 269).

An association of DW6 with plant height has been reported in the literature (Yan et al. 2021, PMID: 34093626, our ref. 35). We discuss DW6 in the context of the 2A/2C translocation, which is not genetically linked to DW6. Carriers of the translocation are shorter on average (Fig. 2c), but we cannot rule the action of genetic factors. CO and FT1 have been proposed as candidate genes involved in oat flowering time regulation by Tinker et al. 2022 (PMID: 35585176,) based on the role of these genes in other cereals (see also Trevaskis et al. PMID: 36311119). Owing to long haplotype blocks associated with pericentric inversion on 7D, we cannot establish a strong link between sequence variants in these genes and phenotypic variation. Future work, including site-directed mutagenesis, will establish the role of these candidate genes in oat.

Minor:

Line 182 (title) Should “thin” be “in”?

Answer: This typo has been corrected.

Supplementary tables (eg 11 and 12) there is hardly any description of the file contents. For me this made the tables impenetrable.

Answer: Since we removed the section on cellulose synthase coding genes, the tables referenced there, including Supplementary Tables 11 and 12, have been removed as well.

Line 345 The association with yield could be due to pedigree structures. Is there any Near Isogenic Line data that could add support?

Answer: We do not have near-isogenic lines, but we have access to unpublished WGS data for 142 from the original 564 lines for which we have phenotypes. We conducted a PCA analysis using these data and did not observe genetic separation between carriers and non-carriers of the translocation.

Figure 2. PCA analysis of 142 Australian lines with and without the 2A/2C translocation event. The plot shows that the translocation does not affect the population structure.

Abstract, Introduction

There were quite a few vague and unconnected statements that do not really relate to the substance of the results eg

Plant based milk (none of the paper is about this)

Genomic research in oat is still at an early stage (does this matter, I thought it was a waste of words in the abstract)

We describe the interplay of gene expression (again this just seems vague better to make concrete statements about how the work will help researchers to understand gene expression in oat)

Answer: We have rephrased the introduction following the reviewer's suggestions.

Fig 1

The PCA plot is not well described. What genotypic data is it based on? I could not see this in methods. I assume that this is the GBS described in the companion paper? Please say so.

Answer: This issue was also raised by Reviewer #1. In response, we updated Figure 1a to make it clearer and connect it to population structure analysis reported in the oat global diversity

companion paper (Bekele et al. 2025, <https://doi.org/10.21203/rs.3.rs-5726397/v1>). We redid the plot using MDS analysis, replaced the pangenome accession name with numbers and added colors to match the population assignment as in Bekele et al. 2025. The plot combines two data sets (1) the pangenome accessions and (2) a global oat collection from Bekele et al 2025 (companion paper). The corresponding legend is:

“MDS plot of 9,111 hexaploid oats including *A. sativa*, *A. byzantina* and *A. sterilis* (light gray points, Bekele et al, 2025) overlaid with the position of PanOat assemblies. Light gray dots are the GBS data and numbering refers to the PanOat assemblies and colors refer to the population assignment (P1-21) from Bekele et al 2025. “

Fig 2

a5 says “panicle” but shows a spikelet

a6 shows what looks like a mature seed (brown) but the sample taken was an immature caryopsis

a1 says embryo but picture is a whole seedling?

f shades of colour are almost impossible to discriminate on my printout

Answer: Thank you for pointing this out. The reviewer is correct, and we have addressed these issues in the updated Fig. 2a.

Regarding “a1”: The image labeled “a1” shows a germinating grain from which embryonic tissues were harvested for RNA extraction, as detailed in the methodology section. Specifically, we dissected parts of the coleoptile, mesocotyl, and seminal roots from germinating seeds approximately four days after germination. Extended Data Fig. 2b-1 shows an actual image of the dissected tissues.

Regarding “f”: we understand that color variations may not print clearly, potentially causing difficulty in distinguishing between colors. We apologize for this. However, this figure has been removed entirely due to comments from this reviewer and reviewer 1 on the section on cellulose synthase encoding genes.

Fig 3 a at least six rearrangements are shown in the dot. A 450 Mb and 50 Mb inversion are described in the legend. It would be helpful to specifically label them in the figure. The units on the axes are not labelled.

c In this case the units are labelled (Mb) but there are no values shown.

Answer: The figure refers only to the large inversion. The small inversions are unique to the reference used, GS7. We added an explanatory sentence to the figure legend: “The three small inversions at ~120Mb, ~200Mb and ~240Mb are unique to the reference genome GS7. “

Referee #3 (Remarks to the Author):

The manuscript presents a pangenome of hexaploid oat. The pangenome is composed of 33 lines. Assemblies and annotations are generated using state of the art approaches. Contig level assemblies are generated using HiFi reads and then put into chromosome-level

scaffold using Hi-C data. The manuscript roughly follows the outline of two recent barley pangenome studies (<https://www.nature.com/articles/s41586-020-2947-8>, <https://doi.org/10.1101/2024.02.14.580266>) including:

- (1) Germplasm genotyping
- (2) Identification of minimum diversity set
- (3) Generation of corresponding assemblies
- (4) Assessment of the low copy pangenome
- (5) Identification of structural variants including inversions
- (6) Linking of large scale variation to hidden legacy of mutation breeding

Overall, I don't have any major technical concerns regarding these analyses. The authors have used similar methods in other species before and they have been proven to work.

The second and perhaps more intriguing aspect of the manuscript is exploration of gene expression patterns among the three subgenomes across multiple accessions and tissues. However, the results rely on gene expression quantification using Kallisto. Gene expression variation among homeologous genes in polyploids has been an active area of research and specialist quantification tools (for example EAGLE-RC) have been developed. Kallisto has in turn been shown to suffer from high error rate in sub-genome read assignment (<https://doi.org/10.1093/bib/bby121>, <https://doi.org/10.1093/bib/bbaa035>). I think that at the very least the transcriptome analysis should be repeated with a more appropriate quantification method to verify the conclusions based on Kallisto results.

Answer: We appreciate the reviewer's thoughtful feedback on the use of Kallisto for gene expression quantification and the suggestion to explore additional methods such as EAGLE-RC. We acknowledge that quantifying gene expression in polyploids is challenging due to the inherent complexity of homeologous genes and subgenomes. Thank you for highlighting EAGLE-RC as a tool of interest.

While EAGLE-RC appears to be a promising option for the wheat community, it is designed based on assumptions that are not fully applicable to the oat genome. Specifically, EAGLE-RC assumes that there have been no translocations or structural rearrangements involving different subgenomes, meaning that gene content remains fixed within specific subgenomes without reshuffling between them, or, in other words, that homeologous genes remain restricted to their respective subgenomes. As described in Kamal et al. 2022 (PMID: 35585233) and the present manuscript, these assumptions do not hold true for the oat genome due to its mosaic genome structure and recombination patterns. Consequently, EAGLE-RC is not well-suited for use with oat data.

That said, after modifying the source code of EAGLE_RC (since the tool was designed with wheat in mind and hard-coded for that species), we were able to execute EAGLE-RC on our data. We compared its results against those of Kallisto, STAR, and HiSAT2, which provided an additional layer of validation for our findings. Notably, all four tools produced similar overall subgenome expression patterns, supporting the robustness of our conclusions (Figure 3).

Our choice of Kallisto was informed by its suitability for quantifying gene expression in polyploid genomes with highly similar subgenomes (<https://www.science.org/doi/10.1126/science.aar6089>). Kallisto's pseudoalignment approach avoids the alignment artefacts often observed in full base-by-base aligners like STAR, particularly in regions of high homeologous similarity. By probabilistically assigning multi-mapping reads, Kallisto provides accurate abundance estimates for homeologous

genes, capturing expression patterns critical for understanding subgenome dynamics. Kallisto achieves this with far greater computational efficiency than traditional aligners, making it a practical choice for analyzing the large and complex oat genome. The additional benchmarking we performed with STAR, HiSAT2, and EAGLE-RC further validates our findings, supporting the robustness of Kallisto’s results in this study.

Figure 3: Boxplot of log-transformed Transcripts Per Million (TPM) values for the A, C, and D subgenomes, estimated using the three read mapping tools Kallisto, STAR, and HISAT2, and the classifier EAGLE-RC.

We used Spearman correlation to evaluate the similarity of TPM (transcripts per million) values generated by four different tools: Kallisto, STAR, HISAT2, and the classifier EAGLE_RC. The correlations were consistently high, exceeding 92%, demonstrating strong agreement across the tools (Figure 4). However, Kallisto displayed slightly lower correlations compared to the others, which can be attributed to differences in their underlying methodologies. Unlike STAR, HISAT2, and EAGLE_RC, which rely on full read alignments followed by quantification using additional tools such as FeatureCounts, Kallisto employs a pseudoalignment strategy. This approach maps reads directly to transcripts without performing traditional alignment, offering computational efficiency but inherently differing in how it handles read quantification. The strong correlation between the three other tools and the lower correlation of each of them to Kallisto, likely reflects these methodological differences, as the alignment-based tools share a more similar pipeline, potentially introducing alignment-specific biases that Kallisto avoids.

Figure 4: Spearman correlation heatmap of gene expression levels (TPM) across the three mapping tools Kallisto, STAR, and HISAT2, and the classifier EAGLE_RC.

Considering the focus on transcription diversity it would also be good if some epigenomic (for example DNA methylation) data were presented to disentangle contribution of genomic and epigenomic variation to gene expression variation.

Answer: We have access to unpublished bisulfite sequencing data for three of the PanOat accessions (GS7, OT380 and *A. sterilis* TN4; see the plots below). The global methylation patterns are consistent between genotypes and subgenomes, indicating that at least at the genome-wide level epigenome differences cannot account for altered expressions levels between subgenomes.

We believe that a meaningful analysis of the relationship between epigenomic states and gene expression needs to be done with matching datasets collected in the relevant tissues and developmental stages in controlled environmental conditions. We believe such experiments go beyond the scope of the current manuscript.

Figure 5: Methylation pattern across the oat genome. Bisulfite sequencing data for three of the PanOat pangenome lines (GS7, *A. sterilis* TN4 and OT380) was mapped to the respective genome and plotted in 1Mb windows. The x-axis shows the position along the chromosome, y-axis shows the percent of methylation, green lines show CHH methylation, red lines show CHG methylation and blue lines show CpG methylation.

I am not entirely convinced that the correlation between genomic variation and gene expression conclusively shows that ‘genetic factors underlie the expression patterns we observed’. This should be at least discussed in more detail. How good is the agreement? Are there exceptions? Also, from technical perspective it was shown that the more similar the gene sequence the more similar Kallisto quantification results will be (<https://doi.org/10.1093/bib/bbaa035>), where sub-genome ‘read partitioning errors lead to gains in correlation’.

Answer: We realize that our phrasing may have unintentionally led to a misunderstanding, and we appreciate the opportunity to clarify. By “genetic factors,” we were specifically referring to DNA sequence variation, such as differences in promoter regions, coding sequences, or other regulatory elements, which could potentially influence gene expression patterns. In our analysis, we computed correlation matrices (using Cramér’s V) from both SNP data and gene expression data and compared them using a Mantel test. This test revealed a statistically significant agreement between each of the gene expression matrices and the matrix based on SNP data, suggesting that variation in the underlying DNA sequence

contributes to the observed variation in gene expression patterns. This observation is biologically intuitive, as sequence elements such as promoter regions are known to regulate gene expression. However, we agree that this correlation should not be interpreted as absolute or universal. While there is broad alignment, there are likely exceptions where other factors, such as environmental influences or epigenetic modifications play a role.

We have revised the manuscript to clarify this distinction and to avoid ambiguous phrasing. The revised sentence now reads:

“The clustering analysis performed using these matrices showed a significant correlation with genomic distance measures (Extended Data Fig. 4d), suggesting that variation in DNA sequences, such as in promoter regions or regulatory elements, contributes to the gene expression patterns we observed.”

Editor:

You will see that two reviewers now endorse publication, but reviewer #2 notes that their comments in the first round of review were misunderstood. Given their comments here, we ask that in a revised manuscript you focus on improving the integration of the biological discoveries with the genomic resources to convincingly *demonstrate* the value of the resources generated here.

Response: We have focused this revision on more fully integrating the biological discoveries with the genomic resources to clearly demonstrate their value. We took several targeted steps to address these concerns:

- Functional integration of structural variants: We investigated the breeding line ND060432, which carries a large structural rearrangement involving chromosome 6D inherited from Amagalon. By combining genome assemblies, whole-genome re-sequencing and newly generated RNA-seq data, we demonstrated how this rearrangement leads to altered gene expression patterns and subgenome expression balance, directly linking the structural variant map to transcriptional consequences (Fig. 4f, Extended Data Fig. 4c,d,e,f).
- Candidate gene analysis for flowering time: We analysed sequence variation and expression of the FT1/VRN3 gene within the inversion on chromosome 7D associated with heading date, uncovering deletions and expression shifts that reinforce its role as a candidate gene—showing how the pangenome resources enable biologically meaningful variant discovery (Fig 3e, Supplementary Fig 6 and 7).
- Transcriptomic effects at translocation junctions: We generated new chromosome-scale DEG plots to visualize expression changes at translocation breakpoints, moving beyond summary tables to illustrate local transcriptional impacts of large-scale structural variants (Supplementary Fig. 1).
- Improved narrative integration: We revised the manuscript text to explicitly connect these functional case studies back to the structural variation map, ensuring that the value of the genomic resources is clearly demonstrated throughout.

Referee #1:

I am satisfied that all of my comments and suggestions have been addressed, and I have no further suggestions for improvement.

Response: We thank the reviewer for their positive assessment.

Referee #2:

I appreciate the authors efforts to accommodate my suggestions. The beta glucan section has been removed and the “map of structural variation” section added. However, the original problem still exists and I apologise if I did not state this strongly enough. In the first review I said: the gulf between the two major elements of the ms

(genomics and cases studies) are just too wide. I am sorry to say that I still find this to be the case. I could write a very long review now with many areas that I simply found insufficient. They are all initially intriguing scientific stories but they still seem only half finished experimentally and (still) almost completely unconnected to the oat pangenome resources developed here. I would implore the authors to take just one of these stories and make a proper job of it. Currently, I do not think that either the specialist or general reader would derive anything useful after line 243. To put it another way I would be highly critical and request major changes for each of the trait sub stories if they had been submitted to a specialist journal in the area.

Response: We recognise the reviewer's central concern about the manuscript: that the original version did not sufficiently integrate the genomic resources with the biological case studies, leaving the connection between these major elements too weak. In this revision, we have focused our efforts on addressing precisely this point. We have strengthened the manuscript by explicitly linking the structural variation map and the pangenome resources to functional and phenotypic outcomes, ensuring that the value of these resources is clearly demonstrated throughout. Below, we detail the specific steps taken to integrate the genomic and biological analyses more fully and to make each case study more directly connected to the core pangenome framework.

Here are one or two examples for each:

A chromosomal inversion linked to early heading-

1. No flowering data is included in supplementary information or (as far as I can see) in linked data repositories. There is no meta data apart from institutional source of data an year, no sites, replication, which accessions carry the inversion.

Response: We thank the reviewer for raising this point. The flowering time data (heading date) used in our GWAS analysis derives from the CORE population and was previously published by Klos et al. (2016) [<https://doi.org/10.3835/plantgenome2015.10.0103>], where detailed experimental design, site information, and replication are described. These data are publicly available via the Triticeae Toolbox (<https://oat.triticeaetoolbox.org/>). Additionally, we have included the distributions of the flowering time phenotypes for the genotypes sequenced in this study in Extended Data Fig. 6. Regarding the inversion state, as noted in the manuscript (lines 289-291), we cannot directly determine inversion genotypes from short-read WGS data without Hi-C or long-read evidence. Instead, we infer inversion states indirectly by analysing the two haplotypes near the inversion breakpoint, which serve as proxies for the structural state.

2. I asked if the authors could show putative functional variants for the candidates they propose (CO etc). In response "Owing to long haplotype blocks associated with pericentric inversion on 7D, we cannot establish a strong link between sequence variants in these genes and phenotypic variation." This was disappointing. One of the main uses eg a breeder might make of the oat pangenome would be to identify potential functional variants for candidate genes. If these genes are being proposed as candidates then the author is laying down a hypothesis that there is a functional variant contained within them that should be very strongly associated with flowering

(regardless of LD around it). The response gave the impression that the authors are not clear about this (??) or they could not be bothered to look.

Response 2: We thank the reviewer for this important comment. In response, we carried out additional sequence and transcriptome analyses to investigate potential functional variants in flowering-time candidate genes located in the chromosome 7D region associated with heading date.

These analyses helped us generate plausible hypotheses about causal polymorphisms. Specifically, we focused on *FT1* (also known as *VRN3*), a well-characterised regulator of flowering. We identified an 18 bp deletion in the *FT1* coding region on chromosome 7D in all inverted lines (except Rhapsody and FM13), and a 12 bp deletion in *FT1* on chromosome 7A that introduces a premature stop codon in most inverted lines. In our transcriptome dataset, *FT1-7D* showed markedly higher expression in inverted lines (mean TPM: 2,499) compared to non-inverted lines (355), while *FT1-7A* expression was similar across genotypes. These findings are now described in the revised manuscript (see Figure 3e and Supplementary Fig. 6 and 7).

We also examined *FT1-7D* expression in the non-inverted cultivar Victoria, which, despite having the same *FT1-7D* coding sequence as other non-inverted lines, showed expression levels and local haplotype structure resembling inverted lines, along with delayed heading. This suggests that factors beyond the *FT1-7D* coding region—potentially including regulatory variation or the influence of other flowering genes such as *VRN2*—may affect expression and flowering time in this background. While these observations do not establish a direct link, they nonetheless highlight *FT1-7D* as a candidate warranting further study.

While these results support the role of *FT1* as a key gene underlying the association signal, functional validation—e.g. through analysis of induced mutants or detailed gene expression studies under controlled photoperiod and vernalisation regimes—is beyond the scope of the present study. Our goal here is to define the genomic landscape, identify promising candidate variants, and provide a foundation for future hypothesis-driven research.

3. Although the authors went to the trouble of adding the map of structural variation section they have not linked this to the following sections which drill down on the actual structural variants. Eg there is a lack of linking sentence from the “420 Mb pericentric inversion” On line 260 to the whole section on this subject that you begin to wonder if they are really the same thing. This comes up again for the following sections and actually shows that this section has been inadequately written.

Response: We thank the reviewer for pointing out this lack of linkage between the overview of structural variation and the detailed case studies that follow. To address this, we have added a bridging paragraph at the end of the “Map of structural variation” section that explicitly introduces the next four sections as illustrative examples drawn from the structural variation map. This should clarify for the reader how the large-scale patterns relate to the subsequent analyses of specific variants and their consequences.

In the following four sections, we highlight case studies from this structural variation map that illustrate its functional importance and applied relevance. These include examples already introduced—structural variants on chromosomes 7D and 2A/2C with clear links to flowering time and mutation breeding—as well as two additional cases: chromosome changes in synthetic-derived lines and widespread impacts of structural variants on recombination and segregation in breeding populations. Importantly, many of these rearrangements are not confined to historical or wild germplasm—they persist in modern cultivars and continue to shape the genomic landscape of contemporary oat breeding.

4. As far as I can see GWAS is inadequately described, maybe not at all. The authors say kmer GWAS was used. There is no description of the data quality (eg QQ plots) or the methods used to account for genetic structure. Moreover one of the main practical applications of the pan genome data will be conventional GWAS using the variant files derived from read mapping to the amazing new reference sequences developed here. Again, the resources described are not being used for the biological exemplars.

Response:

Data and population structure: As noted in our response to point 1, the GWAS analysis was based on heading date data from the CORE population (Klos et al. 2016) and used a reduced k-mer matrix derived from our sequencing data. To account for genetic structure, we included a kinship matrix in the linear mixed model implemented in GEMMA, which is standard for controlling relatedness in GWAS.

Data quality: We now include a QQ plot (Supplementary Fig. 4) for the GWAS based on the Ithaca 2010 heading date phenotype, demonstrating appropriate control of type I error and no inflation of test statistics. We also added an MDS plot (Supplementary Fig. 8) based on SNP data from all 295 CORE lines, which shows no pronounced population structure that might confound the GWAS results.

Future applications: We agree that one of the key practical uses of the pangenome resource will be conventional GWAS based on variant calls from mapping reads to these new high-quality references. The VCF files from our CORE data read mapping will be deposited in the European Variation Archive (EVA), enabling precisely such analyses by the community. Please note that an EVA accession can only be registered once the underlying sequence data (PRJEB62778) have been released in the European Nucleotide Archive.

The hidden legacy of mutation breeding

1. Again the phenotypic data is not provided although it is at least mentioned in the Methods under “Yield evaluation” (I think). This is really important for as panel 4c is the reason that this is an interesting question. The data is from the 2017-2022 Australian National Variety Tests. Each year included 19 to 31 trials across Australia, with a total of 158 trials designed in three replications. There were six 2A/2C translocated varieties and eleven non-translocated varieties. This means that some method of correction for year and site effect has been applied. This is not described

in Methods but is crucial as it's correct implementation underpins the validity of this data. A p value is provided for yield ($p=0.0351$). How was this arrived at? Again, as far as I can see there are no statistics in Methods. It is not stated in the Fig 4 legend that $n=17$. For me this is the most important point. Yield is a complex, highly polygenic, with low heritability, high G×E. It is not statistically valid to associate a yield effect to a single locus based on 17 lines. If it was this simple why do we all bother with GWAS panels on a minimum $n=200$?

Response: We thank the reviewer for these thoughtful and important comments on the yield analysis, which we agree is the most critical point.

Data description and statistical model:

The yield data were drawn from the Australian National Variety Trials from 2018–2022, encompassing 158 trials across Australia with three replications per trial, as now detailed under “Yield evaluation” in the Methods. Trials followed standard NVT protocols, and the multi-environment design provides robust estimates across locations and years. Yield was analysed using linear mixed models fitted with ASReml-R, incorporating spatial trends and random effects for blocking structures. Residual diagnostics were performed to confirm model assumptions.

Reanalysis accounting for site-year variability:

In response to the reviewer's concerns, we re-analysed the yield data by calculating each variety's yield relative to the mean of all varieties within each trial. This normalises for environmental variation across trials and years. An ANOVA was then performed on these relative yields to compare varieties with ($n=6$) and without ($n=11$) the 2A/2C translocation. The observed yield difference remained statistically significant ($p=0.035$).

Context and limitations:

We fully acknowledge that yield is a complex, polygenic trait with high G×E interactions and typically low heritability. As the reviewer rightly notes, robust associations to single loci generally require much larger, purpose-built populations (e.g. GWAS panels of $n>200$) and multi-year trials specifically designed for genetic analysis. Our comparison is thus best interpreted as an exploratory observation, highlighting a possible yield benefit linked to the 2A/2C translocation that warrants future investigation in dedicated studies. We have made this limitation explicit in the relevant paragraph of the Results section:

However, given the complexity of yield, with low heritability and strong G×E effects, this association should be viewed as exploratory and warrants targeted studies for confirmation.

Revisions to the manuscript:

We have substantially expanded the Methods section to include a full description of the yield data, statistical models, and residual checks. We also updated Fig. 4c to present the re-analysed results, adjusted for site-year variability, and revised the figure legend to state the sample sizes ($n=6$ vs $n=11$).

We appreciate the reviewer's comments, which led to a clearer presentation of the yield analysis and its limitations.

Chromosomal rearrangements in oat breeding germplasm

For me this section really summarises the problem with this manuscript. Looking through a set of very dry supplementary figures I was delighted to see S fig13. It showed a truly fascinating and highly dynamic landscape of major structural variants in oat. This is why do pan genomics! But these results are buried. Not mentioned in the map of structural variation section at all. No attempt is made to describe the transcriptional consequences of these major variants (deletions? Maybe introgressions?). Amagolon is not even shown in Fig1 although it carries the 4C inversion mentioned in text while the other chosen accession in fig 1 do not.

Response: We thank the reviewer for highlighting this crucial point. We fully agree that the dynamics of major structural variants are a key motivation for pangenomics, and have taken steps to better integrate these findings into the main narrative.

Specifically, to address the functional consequences of large-scale chromosomal rearrangements, we investigated the breeding line ND060432, which carries Amagolon in its pedigree and exhibits an exchange of chromosome 6D with chromosome 6A from Amagolon. This line thus provides a valuable opportunity to investigate the transcriptional impact of structural variants such as those observed in Extended Data Fig. 10 (we believe the reviewer meant Extended Data Fig. 10, since Supplementary Figure 13 doesn't exist).

To examine the transcriptional consequences of the loss of chromosome 6D in ND060432, we sequenced RNA data from leaf tissue from four biological replicates. Seedlings were grown under controlled conditions, and RNA was extracted from dissected leaf blades, making these samples comparable to the sample we previously obtained for the PanOat lines.

Gene expression was quantified using Kallisto against the GS7 transcriptome reference. Transcripts per million (TPM) were calculated, and gene-level expression was summarized using the median of replicates. We then compared the expression profiles of ND060432 to those from GS7 to assess the effect of the 6D deletion on global expression patterns.

Ternary plot of the expression of triads that involve chromosome 6. The hexagonal density plot displays the relative expression levels (TPM values) of homoeologous gene triads, where each point represents the normalized expression contribution of genes from subgenomes A (top vertex), C (bottom left vertex), and D (bottom right vertex). The plot presents the distribution of expression bias patterns among the three subgenomes, with points closer to the vertices indicating subgenome-specific expression dominance, while points near the centre represent balanced expression across all three subgenomes. Only triads with non-zero expression values across all subgenomes were included in the analysis. Number of triads for each plot: 13,156 (top left) and 14,034 (top right), 2,785 (bottom left) and 2,978 (bottom right).

Moreover, gene expression from the A and D subgenomes was compared across chromosome groups in both GS7 and ND060432. While GS7 displayed balanced A:D expression overall (**Extended Data Figure 4 c-e**), ND060432 showed a significantly increased A:D expression ratio. This confirms the transcriptional loss consistent with the 6D deletion and provides statistical evidence (p -value < 0.001 Wilcoxon rank-sum tests, FDR-corrected, Supplementary Table 15a, **Extended Data Fig 4c-e**) for the altered expression pattern, with clear effect sizes visualised via forest plots.

In addition, we find a strong bias for D-subgenome suppression in ND060432. In GS7, we observe that 65% of the triads (genes with a 1:1:1 copy in the A-, C-, D-subgenomes) on chromosome 6D show a balanced expression, while only 7% of the triads are D-suppressed. In ND060432, on the other hand, only 19% of the triads show a balanced expression and 49% were found to be D-suppressed. Ternary plots were used to visualize these patterns for all triads and in each of the chromosome groups (**Fig. 4f, Extended Data Fig 4f**).

These results demonstrate the functional impact of large-scale chromosomal variation—specifically the 6D deletion—on gene expression patterns in oat and illustrate the value of integrating RNA-seq data with structural genome variation.

These results were included in the main manuscript as well as in Fig. 4 and Extended Data Fig 4. In addition, we added Amagalon to Fig. 1c to highlight its unique 4C inversion, addressing the reviewer's concern that it was previously omitted.

We also extended our investigation of functional consequences by exploring the heading date GWAS signals on chromosome 7D. We now present variation and expression data for a candidate FT1/VRN3 gene within this low-recombination region, linking the inversion on 7D more directly to phenotypic effects on flowering time.

In summary I find that the members of the team responsible for responding to my suggested changes did not really embrace the spirit of the suggested changes. As I show above with several examples, the map of structural variation section stands alone and is not at all integrated with the following sections. In these sections there are several basic experimental problems that lead the reader to feel doubtful about the interpretation of results. Moreover, the team seem to lack some curiosity in terms of other interesting results.

For example, because they were asked, they described the transcript levels at translocation junctions the results sound really interesting:

“We investigated gene expression changes at translocation breakpoints by performing 266 differential gene expression (DGE) analysis between translocated and non-translocated lines. 267 This revealed significant enrichment of differentially expressed genes near breakpoints on 268 chromosomes 1A, 1C, and 7D, whereas the reciprocal translocation between 2A and 2C 269 showed significantly fewer DEGs.”

But the next step chosen was based on annotation of functional gene classes (enrichment for carbohydrate metabolism.....). This seems highly speculative. It would be much more interesting eg to see a figure change with DEGs plotted along

the chromosome to graphically demonstrate the very interesting observation that is currently only given a few lines of text and a basic table in S12a.

Response: We share the reviewer's interest in exploring transcript changes at translocation junctions, and performed additional analyses to address this point. We generated chromosome-scale plots of differentially expressed genes (DEGs), now included as Supplementary Fig. 1, which visually illustrate patterns of up- and down-regulation along translocation borders. While these reveal some local shifts in expression, there is no consistent genome-wide trend that would support a specific mechanistic model at this stage. For this reason, we have refrained from strong conclusions. We see these observations as an important basis for future targeted studies to dissect the regulatory consequences of such structural variants.

Referee #3:

Overall, I am satisfied with the authors responses and the additional analyses performed. I have no further major concerns. I would suggest including the comparative analysis of different gene expression quantification methods found in the response letter (especially Figs 3 and 4) in the supplementary material to make it available to the interested readers. While overall trends remain the same across methods there are some differences, which could assist interpretation of the results. Also, in my opinion showing agreement across methods strengthens the message of the manuscript.

Response: We thank the reviewer for their positive assessment and agree that including the comparative analysis of gene expression quantification methods adds value for readers. We have therefore incorporated the benchmarking results, including Figures 3 and 4 from the response letter, as a new Supplementary Note in the revised manuscript.

Referees' comments:

Referee #2 (Remarks to the Author):

The authors have done a thorough job of bringing to life the functional integration of structural variants. This is achieved through added depth on ND060432 with 6D rearrangements from Amagalon. They have really shown how genome assemblies can be combined with re-sequencing and RNA-seq data to link structural variants with changes in transcriptome. The reanalysis of sequence variation and expression of FT1 demonstrates the usefulness of the genomic resources for candidate gene analysis. Chromosome-scale DEG plots show the beautiful expression changes at translocation breakpoints and shine a light for the general reader on the probable consequences of large-scale structural variants. I agree that the narrative improved and the gap between genomics and biology sufficiently well filled to demonstrate the utility of these important resources.

Answer: We thank the reviewer for his thoughtful comments and suggestions during the review process that pushed us to improve our manuscript.

Small comments and corrections:

317-315

Three 318 important quantitative trait locus (QTLs) are mapped to chromosome 7D, involving flowering time (CO and VRN3/FT)³⁰ and daylight insensitivity (Di)^{33 319}. This should be “daylength” rather than “daylight”.

Answer: We corrected the “daylight” to “daylength”

324-325

an 18 bp deletion in FT1-7D specific to 325 inverted lines

This needs a little more detail. As far as I can see this N terminus coding in frame. Likely functional or not?

Answer: We cannot make any solid conclusion based only on the sequence variation. There is only the causal evidence that all the accessions that have the non-inverted haplotype also have the 18 bp deletion and are associated with a mild-winter habitat. This requires further experimental data.

326-328

These results align with findings by Mehtab-Singh et al.^{32 327}, who reported delayed flowering in triple CRISPR/Cas mutants of FT1, underscoring 328 its likely role in heading date variation.

The role of FT1 in the control of flowering is already more than clear and I don't think that this sentence or reference is helpful.

Answer: We removed the sentence from the manuscript

Supplementary Fig 6. Multiple sequence alignment of FT1-7D CDS from the pangenome assemblies showing a 18bp deletion.
“deletion”

Answer: Corrected

Supplementary Fig 5. Population structure among Australian oats. Australian oat varieties (n=169), including the 17 varieties with known presence or absence of the translocation, have been re-sequenced with whole-genome shotgun technology at 20x sequence depth, their reads were mapped to reference genome OT3098 and SNPs profiles were then derived (detailed methodology and results are to be published elsewhere). The population structure was examined with principal component analysis.

“the translocation” Presumably this is 7D translocation? This should be made clear.

Answer: Yes. We added “7D” to the figure legend.

391-393

fatty acid hydroxylase (Supplementary Table 15.8). Further functional studies are required 392 to test if this gene is casual to the Dw6 phenotype
“casual” to “causal”?

Answer: Corrected

427-439 Changes are good

Sup table 14 column D, specify which translocation (2A/2C)

Answer: Yes it is the 2A/2C translocation. We added this to the table description and to the appropriate column header.

1424

Trails with average grain yield below 1000 kg/ha 1425 were deemed as abnormal and removed from further analysis
“trails” to “trials”

Answer: Corrected

1392 – 1440

Very helpful